# Subicular neurons encode concave and convex geometries

Yanjun Sun[1,2 ✉], Douglas A. Nitz[3], Xiangmin Xu[2,4] & Lisa M. Giocomo[1 ✉]

Animals in the natural world constantly encounter geometrically complex landscapes. Successful navigation requires that they understand geometric features of these landscapes, including boundaries, landmarks, corners and curved areas, all of which collectively define the geometry of the environment[1–12]. Crucial to the reconstruction of the geometric layout of natural environments are concave and convex features, such as corners and protrusions. However, the neural substrates that could underlie the perception of concavity and convexity in the environment remain elusive. Here we show that the dorsal subiculum contains neurons that encode corners across environmental geometries in an allocentric reference frame. Using longitudinal calcium imaging in freely behaving mice, we find that corner cells tune their activity to reflect the geometric properties of corners, including corner angles, wall height and the degree of wall intersection. A separate population of subicular neurons encode convex corners of both larger environments and discrete objects. Both corner cells are non-overlapping with the population of subicular neurons that encode environmental boundaries. Furthermore, corner cells that encode concave or convex corners generalize their activity such that they respond, respectively, to concave or convex curvatures within an environment. Together, our findings suggest that the subiculum contains the geometric information needed to reconstruct the shape and layout of naturalistic spatial environments.

Neurons that contribute to building a 'cognitive map' for an environment, including hippocampal place cells[13,14] and entorhinal grid cells[15], integrate information from geometric environmental features to shape their spatial representations[16–24]. To accomplish this integration, the brain needs to represent the explicit properties of geometric features in the environment, such as boundary distances and corner angles. However, it is still unclear which geometric properties are encoded in the brain at the single-cell level, outside of egocentric (self-centred) and allocentric (world-centred) boundary coding in the hippocampal formation and associated regions[25–30]. Unlike traditional laboratory conditions, which often have straight walls, natural environments are full of concave and convex shapes, from networks of tree branches to winding burrow tunnels. Given that the combination of straight lines and curves can give rise to any shape, we hypothesized that the brain encodes the concave and convex curvatures of an environment (for example, corners and curved protrusions), in addition to straight boundaries[24,25,29]. One brain region that could play a role in encoding concave and convex environmental features is the subiculum, a structure that receives highly convergent inputs from both the hippocampal subregion CA1 and the entorhinal cortex[31,32]. Earlier work has demonstrated that neurons in the subiculum encode the locations of environmental boundaries and objects in an allocentric reference frame, as well as the axis of travel in multi-path environments[25,33–35]. Here, we describe single-cell neural representations for concave and convex environmental corners and curvatures in the dorsal subiculum, which reside interspersed with single-cell neural representations for environmental boundaries.

## Subiculum neurons encode environmental corners

To record from large numbers of neurons in the subiculum, we performed in vivo calcium imaging using a single photon (1P) miniscope in freely behaving mice (Fig. 1a, b). We primarily used *Camk2a-Cre; Ai163* (ref. 36) transgenic mice, which exhibited stable GCaMP6s expression in subiculum pyramidal neurons (Extended Data Fig. 1a) and thus facilitated longitudinal tracking of individual neurons[37] (Fig. 1c). Calcium signals were extracted with CNMF[38] and OASIS[39] deconvolution, and subsequently binarized to estimate spikes for all cells (Fig. 1d and Extended Data Fig. 1b). We treated these deconvolved spikes as equivalent to electrophysiological spikes and for calculating spike rates in downstream analyses.

We placed animals in one of four open field arenas, including a circle, an equilateral triangle, a square and a hexagon. On each day, we recorded subiculum neurons from two of these four arenas (20 min per session) (Fig. 1e,f). Many subicular neurons exhibited place cell-like firing patterns that were spatially modulated but not geometry-specific across the different environments (Extended Data Fig. 1c), as previously reported[40,41]. However, we also observed a subset of subicular neurons

[1]Department of Neurobiology, Stanford University School of Medicine, Stanford, CA, USA. [2]Department of Anatomy and Neurobiology, School of Medicine, University of California, Irvine, Irvine, CA, USA. [3]Department of Cognitive Science, University of California, San Diego, La Jolla, CA, USA. [4]Center for Neural Circuit Mapping (CNCM), University of California, Irvine, Irvine, CA, USA. ✉e-mail: yanjuns@stanford.edu; giocomo@stanford.edu

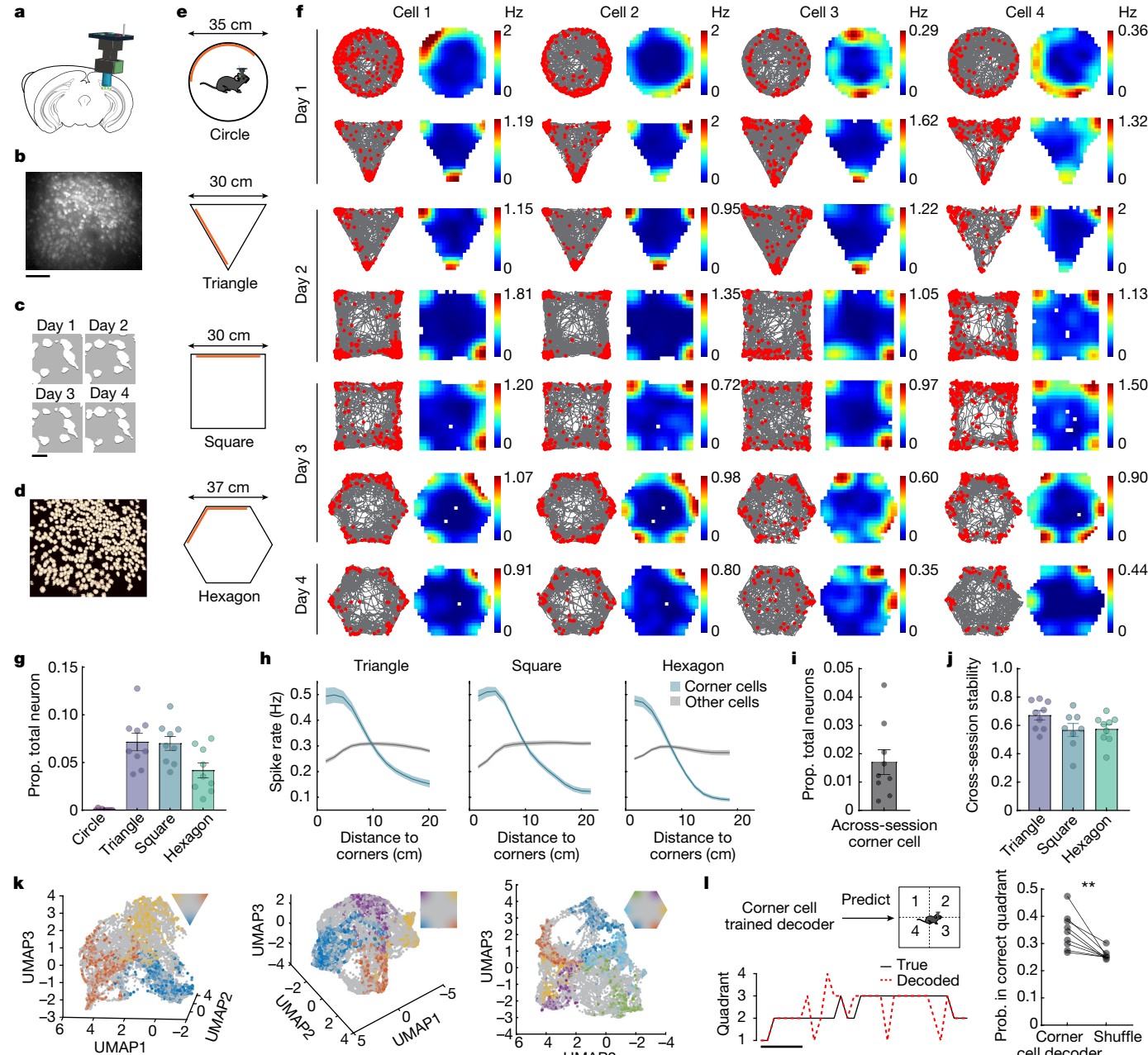

**Fig. 1 | The subiculum contains neurons that exhibit corner-associated activity. a**, Schematic of miniscope calcium imaging in the subiculum. **b**, Maximum intensity projections of subiculum imaging from a representative mouse. **c**, An enlarged region of interest from **b** across days. **d**, Extracted neurons from **b**. **e**, Open arena environment shapes. Orange bars indicate local visual cues. **f**, Four representative corner cells from three mice. Each column is a cell with its activity tracked across sessions. Raster plot (left) indicates extracted spikes (red dots) on top of the animal's running trajectory (grey lines) and the spatial rate map (right) is colour coded for maximum (red) and minimum (blue) values. **g**, Proportion of corner cells in each environment (arena shapes, *x*-axis). Each dot represents a mouse, with a maximum of two sessions averaged within each mouse (mean ± standard error of the mean (s.e.m.); $n = 9$ mice). **h**, Positional spike rates plotted relative to the distance to the nearest corner ($n = 9$ mice). Solid line, mean; shaded area, s.e.m. **i**, Proportion (prop.) of corner cells across sessions (mean ± s.e.m.; two-tailed Wilcoxon signed-rank test against zero: $P = 0.0039$; $n = 9$ mice). **j**, Cross-session stability (Pearson's correlation) of across-session corner cells in **i** for each environment. **k**, Three-dimensional (3D) embedding of the subiculum population activity in the triangle, square and hexagon from a representative mouse. Uniform manifold approximation and projection (UMAP) plots shown. Each dot is the population state at one time point. Time points within 5 cm of the corners are colour coded as shown in the inset. **l**, Left, an example of decoding the animal's quadrant location over time using a decoder trained on corner cell activity. Black line, true quadrant location; red dotted line, decoded quadrant location. Right, quadrant decoding accuracy versus shuffle (mean ± s.e.m.: decoder versus shuffle, 0.35 ± 0.02 versus 0.26 ± 0.006; two-tailed Wilcoxon signed-rank test: $P = 0.0039$; $n = 9$ mice). *y*-axis indicates the probability (prob.) that the animal's location was decoded in the correct quadrant. Scale bars, 100 μm (**b**), 10 μm (**c**), 5 s (**l**).

that were active near the boundaries of the circle (Fig. 1f). Following the activity of these neurons in all the other non-circle environments revealed that they exhibited increased spike rates specifically at the corners of the environments (Fig. 1f). To ensure that these neurons were anatomically located in the subiculum, we used a viral strategy to restrict GCaMP expression to the subiculum (Extended Data Fig. 1d) and observed the same corner-associated neural activity (Extended Data Fig. 1e).

To classify neurons that exhibited corner-specific activity patterns, we devised a corner score that measures how close a given spatial field is to the nearest corner (Extended Data Fig. 1f). The score ranged from −1 for fields situated at the centroid of the arena, to +1 for fields perfectly located at a corner (Extended Data Fig. 1f and Methods). We defined a corner cell as a cell with: (1) a corner score greater than the 95th percentile of a distribution of shuffled scores generated by shuffling the spike times along the animal's trajectory (Extended Data Fig. 1g–i, Extended Data Fig. 2a–d and Methods); (2) a distance between any two fields (major fields, if number of fields greater than number of corners) greater than half the distance between the corner and centroid of the environment (Extended Data Fig. 1g,j and Methods); and (3) a within-session spatial stability value greater than 0.3 (Extended Data Fig. 1k and Methods). Using this definition, we classified 7.2 ± 0.9% (mean ± s.e.m., $n = 9$ mice) of neurons as corner cells in the triangle, 7.0 ± 0.7% in the square and 4.2 ± 0.8% in the hexagon (Fig. 1g and Extended Data Fig. 3). Notably, this method classified almost no neurons as corner cells in the circle when four or three equally spaced points on the wall were assigned as the 'corners' for the environment (Fig. 1g: 0.04 ± 0.03%, four points; Extended Data Fig. 1l: 0.0 ± 0.0%, three points). Applying the same procedure to all other environments, we confirmed that no more than 0.5% of neurons were classified as corner cells when we manually moved the corner location to the walls (Extended Data Fig. 1l). Furthermore, we imaged 5,212 CA1 neurons from 12 mice in a square environment. Only 0.6 ± 0.1% of CA1 neurons were classified as corner cells (Extended Data Fig. 1m), a significantly lower proportion than the number of subiculum cells classified as corner cells in the square (Fig. 1g; Mann–Whitney test: $P < 0.0001$).

To verify that neurons classified as corner cells encode locations near corners, we plotted the spike rate for each bin on the rate map as a function of the distance to the nearest corner. As expected, corner cells showed a higher spike rate near the corners than the centroid, which was not observed in non-corner cells from the same animal (Fig. 1h). Second, a decoding analysis revealed that subicular neurons provided significant information regarding the animal's spatial location (Extended Data Fig. 2e,f). Removing corner cells from this decoder resulted in higher decoding errors near the corners than at the centre of the environment, compared to the full decoder (Extended Data Fig. 2g,h). Accounting for the animal's behaviour, as measured by a corrected peak spike rate at each corner (Extended Data Fig. 4a–f and Methods), we did not observe a bias in the corner cell population activity towards encoding specific corners (Extended Data Fig. 4f). Finally, across all non-circle geometries, 1.7 ± 0.4% of neurons were consistently classified as corner cells (referred to as 'across-session corner cells', Fig. 1i). These across-session corner cells exhibited stable corner-associated activity in all environments (Fig. 1j) (mean cross-session stability from 0.57–0.67, Pearson's correlation). Of note, the neural population classified as corner cells in one environment continued to show activity at corners in later sessions/conditions in which they were not classified as corner cells (Extended Data Fig. 4g,h), indicating corner activity generally persisted across different geometries when considering the neurons as a population rather than only single cells classified based on their corner score.

To visualize the representation of corners in the low-dimensional neural manifold of the subiculum, we performed three-dimensional (3D) embedding[42] of the population activity of all recorded neurons in the triangle, square and hexagon (Fig. 1k, Extended Data Fig. 5a–c and Methods). Across different mice, we found that the representation of each corner for a given environment was distinct from other corners and the rest of the space and that the sequential order of corners was effectively preserved in the low-dimensional neural manifold (Fig. 1k and Extended Data Fig. 5a–c). On the other hand, corner representations also converged at a specific point on the manifold (as indicated by the black circles in Extended Data Fig. 5a–c). This convergence suggests that subiculum neurons also generalize the concept of corners, in addition to representing their distinct locations. A prediction of this

'separated yet connected' corner representation is that corner cells more generally encode the presence of a corner and only modestly encode the precise allocentric location of corners (for example, the northwest versus the southwest corner). To test this idea, we first trained a decoder on corner cell activity and used this decoder to predict the animal's quadrant location in the square environment (Fig. 1l). While decoding performance significantly exceeded chance levels, the accuracy of the decoding was only moderate (approximately 35%, Fig. 1l), consistent with the idea that corner cells generalize their coding to all corners. Next, we implemented a decoder to predict the geometry (that is, identity) of the environment and compared the prediction accuracy of the decoder when using data from locations near versus away from the geometric features of the environment (that is, corners, boundaries). This approach revealed that subiculum neurons carried more information about the overall environmental geometry when the animal was closer to a geometric feature (Extended Data Fig. 5d–f). Together, these results point to the subiculum as a region that encodes information related to corners and the geometry of the environment.

## Corner coding is specific to environmental corners

To investigate the degree to which corner cells specifically encode environmental corners, we considered three properties that comprise a corner: (1) the angle of the corner, (2) the height of the walls and (3) the connection between two walls. First, we imaged as animals explored two asymmetric environments: a right triangle (30-60-90° corners) or a trapezoid (55-90-125° corners) (Fig. 2a). In these asymmetric environments, corner cells composed 3.6 ± 0.3% and 2.1 ± 0.3% of all neurons recorded in the right triangle and the trapezoid, respectively (Fig. 2b,c; $n = 8$ mice). By comparison, there were essentially no neurons classified as corner cells when points on the wall were assigned as the 'corners' of these environments (Extended Data Fig. 1n). In the right triangle, corner cell peak spike rates were significantly higher for the 30° (2.32 ± 0.14, mean ± s.e.m.) corner compared to the 60° (1.67 ± 0.16) and 90° (1.76 ± 0.16) corners, but did not differ between the 60° and 90° corners (Fig. 2b,d). To rule out the possibility that this was due to the limited angular range of these acute angles, we compared the peak spike rates at the corners of the trapezoid and found that the peak spike rates of corner cells increased from 125° (1.49 ± 0.12, mean ± s.e.m.) to 90° (1.90 ± 0.10) to 55° (2.23 ± 0.12) (Fig. 2e). We also compared the peak spike rates at the corners using the aforementioned across-session corner cells in the triangle (60°, 1.76 ± 0.12, mean ± s.e.m., $n = 9$ mice), square (90°, 1.47 ± 0.11) and hexagon (120°, 1.44 ± 0.13), and found the peak spike rate was higher in the triangle compared to the square and hexagon (Fig. 2f). Together, these results suggest that corner cells encode information regarding corner angles, particularly within asymmetric environments.

Next, we imaged as animals explored the normal square environment (as in Fig. 1) with 30 cm high walls (normal square), followed by a low-wall square environment with 15 cm high walls (Fig. 2g,h). Quantitative analysis revealed the proportion of corner cells significantly decreased from the normal (7.2 ± 0.8%, mean ± s.e.m.) to the low-wall square (3.3 ± 0.6%) (Fig. 2i). The remaining corner cells in the low-wall square had corner spike rates similar to their corner spike rates in the normal square (1.72 ± 0.04 versus 1.74 ± 0.08; $n = 9$ mice) (Fig. 2j). However, for neurons classified as corner cells in the normal square but not in the low-wall square, their spike rates near the corners of the low-wall square were still higher than those in non-corner cells (Fig. 2h,k), indicating that their corner-related activity decreased by lowing the wall but was not completely lost. Finally, in comparison to corner cells, the proportion of subiculum place cells did not change between the normal (68.5 ± 2.6%) and low-wall squares (66.4 ± 4.5%) (Fig. 2l). Together, these results indicate that the tuning of corner cells is sensitive to the height of the walls that constitute the corner.

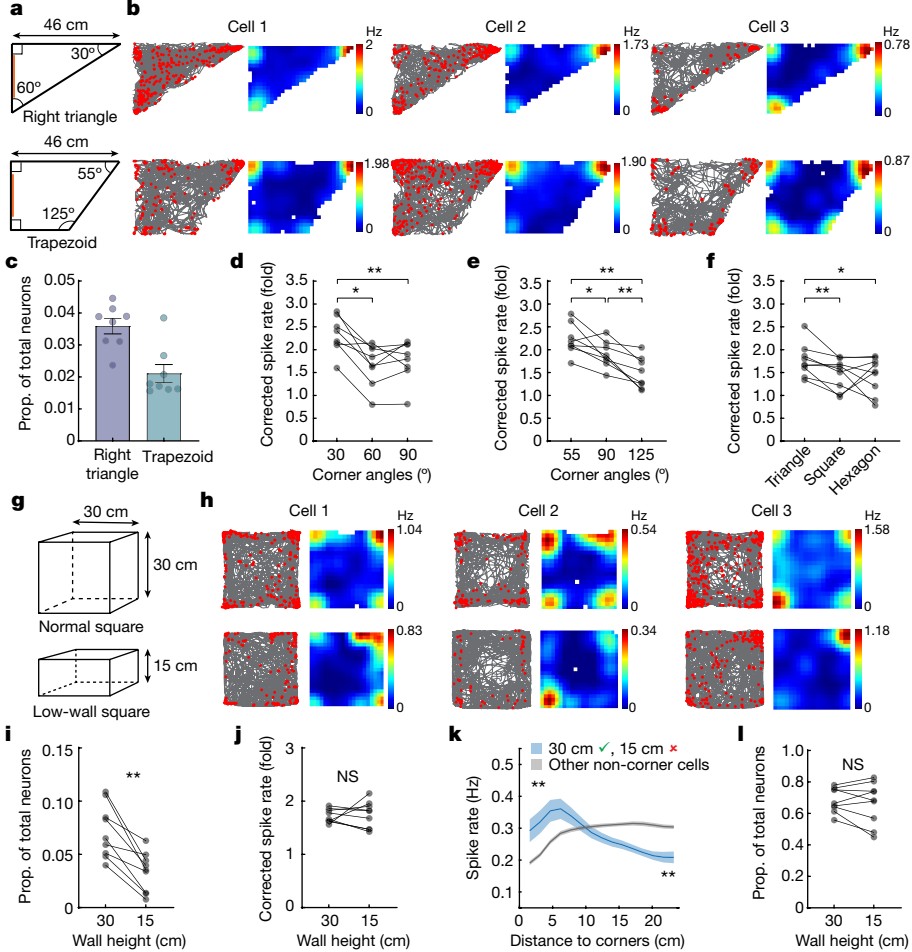

**Fig. 2 | Corner cell coding is sensitive to corner angle and wall height.**
**a**, Asymmetric arena shapes. Orange indicates visual cues. **b**, Three
representative corner cells from three mice, plotted as in Fig. 1f. **c**, Proportion
of corner cells in each environment. Each mouse averaged from two sessions
(mean ± s.e.m.; $n$ = 8 mice). **d**, Corrected peak spike rates of corner cells at each
right triangle corner (repeated measures ANOVA: $F(1.65, 11.53)$ = 18.54,
$P$ = 0.0004; 30° versus 60°: $P$ = 0.016; 30° versus 90°: $P$ = 0.0078; 60° versus
90°: $P$ = 0.31; $n$ = 8 mice). **e**, Same as **d**, for trapezoid (repeated measures
ANOVA: $F(1.85, 12.97)$ = 23.94, $P$ < 0.0001; 55° versus 90°: $P$ = 0.023; 55° versus
125°: $P$ = 0.0078; 90° versus 125°: $P$ = 0.0078; $n$ = 8 mice). **f**, Corrected peak
spike rates of across-session corner cells in different environments (arena
shapes, $x$-axis; repeated measures ANOVA: $F(2,16)$ = 3.88, $P$ = 0.042; triangle

versus square: $P$ = 0.0078; triangle versus hexagon: $P$ = 0.039; square versus
hexagon: $P$ = 0.91; $n$ = 9 mice). **g**, Schematic of normal versus low-wall squares.
**h**, Three representative corner cells from three mice, plotted as in **b**.
**i**, Proportion of corner cells in 30 versus 15 cm square ($P$ = 0.0039; $n$ = 9 mice).
**j**, Corrected peak spike rates of corner cells in 30 and 15 cm square ($P$ = 0.73).
**k**, Spike rates relative to the distance from the nearest corner in the 15 cm square.
Blue curve denotes neurons that were corner cells in the 30 cm (green check)
but not the 15 cm square (red cross). Blue and grey curves, approximately 5 cm
from either the curve's head or tail, were compared (all $P$ = 0.0039). Solid line,
mean; shaded area, s.e.m. **l**, Same as **i**, for place cells ($P$ = 0.50). Pairwise
comparisons throughout the figure use two-tailed Wilcoxon signed-rank test.
NS, not significant.

Finally, we imaged as animals explored a large square environment
in which we inserted a discrete corner and gradually separated its two
connected walls (1.5, 3 or 6 cm separation) (Fig. 3a). We identified cor-
ner cells in the baseline session and tracked their activity across all
manipulations (Fig. 3b). Despite the insertion of the discrete corner,
corner cells identified in baseline did not change their average peak
spike rates at the corners of the square environment (Fig. 3b,c). Upon
the insertion of the discrete corner, corner cells developed a new field
near the inserted corner (Fig. 3b). As the distance between the walls of
the discrete corner increased, the peak spike rate of corner cells at that
corner decreased (Fig. 3d). Even at the largest gap of 6 cm, however,
corner cell peak spike rate at the discrete corner was still significantly
higher than at baseline (1.21 ± 0.12 versus 0.40 ± 0.05, mean ± s.e.m.)
(Fig. 3d), indicating that the animal may still perceive the inserted walls
as a corner. Furthermore, the peak spike rates of corner cells at 1.5 cm
(1.40 ± 0.16), 3 cm (1.34 ± 0.14) and 6 cm (1.21 ± 0.12) gap were signifi-
cantly attenuated compared to the 0 cm (1.86 ± 0.19) gap condition
(Fig. 3d), suggesting that corner cells are sensitive to the connection

of the walls that constitute the corner. In comparison, there was no
effect at the inserted corner when we performed the same analyses
using non-corner cells (Fig. 3e).

## Decoupling corner coding from non-geometric features

We next investigated whether corner cells in the subiculum were
sensitive to non-geometric features of a corner. To test this, we placed
the animals in a shuttle box with two connected square compart-
ments that differed in colour and texture (Extended Data Fig. 6a).
Corner cells showed increased spike rates uniformly across all the
corners, regardless of the context (Extended Data Fig. 6b,c). In addi-
tion, their average peak spike rates at corners were comparable across
the two contexts regardless of the context in which the corner cell
was defined (Extended Data Fig. 6d–f). These results suggest that
corner cells in the subiculum primarily encode corner-associated
geometric features, rather than non-geometric properties, such as
colours and textures.

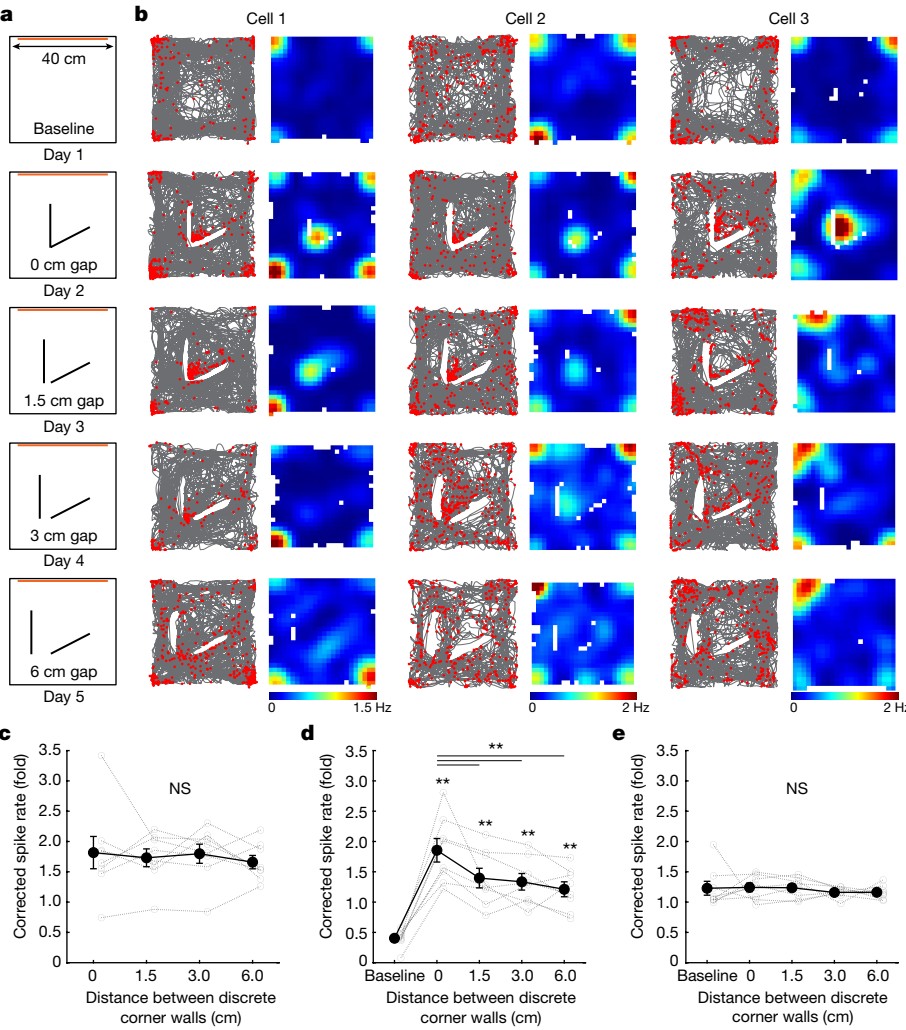

**Fig. 3 | Corner cell coding is sensitive to the proximity of the walls that constitute a corner. a**, Schematic of the open field arena and sessions in which a discrete corner was inserted into the centre of the environment. Orange bars indicate the locations of local visual cues. **b**, Raster plots and the corresponding rate maps of three representative corner cells from three different mice, plotted as in Fig. 1f. Each column is a cell in which its activity was tracked across sessions. Note that rate maps for each cell were plotted to have the same colour-coding scale for maximum (red) and minimum (blue) values. **c**, Corrected peak spike rates of baseline-identified corner cells at the environmental corners (not the inserted corner) across non-baseline sessions (repeated measures ANOVA:

$F(1.99, 13.99) = 0.30$, $P = 0.74$; $n = 8$ mice). Black dots represent mean ± s.e.m.; grey lines represent each animal. **d**, Corrected peak spike rates of baseline-identified corner cells at the inserted corner across all the sessions, plotted as in **c** (repeated measures ANOVA: $F(2.42, 16.96) = 25.62$, $P < 0.0001$; two-tailed Wilcoxon signed-rank test: baseline versus 0 cm, $P = 0.0078$; baseline versus 1.5 cm, $P = 0.0078$; baseline versus 3 cm, $P = 0.0078$; baseline versus 6 cm, $P = 0.0078$; 0 cm versus 1.5 cm, $P = 0.0078$; 0 cm versus 3 cm, $P = 0.0078$; 0 cm versus 6 cm, $P = 0.0078$; $n = 8$ mice). **e**, Same as **d**, but for non-corner cells (repeated measures ANOVA: $F(1.57, 10.97) = 0.33$, $P = 0.68$; $n = 8$ mice).

We then placed the animals in a square arena in complete darkness. In this condition, the representations of corners by corner cells persisted, and the proportion of corner cells remained unchanged (Extended Data Fig. 6g–i). Similarly, trimming the animals' whiskers did not significantly affect the proportion of corner cells (Extended Data Fig. 6g–i). However, compared to the baseline, there was a decrease in the peak spike rates of corner cells in darkness, but not after whisker trimming (Extended Data Fig. 6j). By contrast, recording in darkness or after whisker trimming significantly decreased the number of place cells in the subiculum (Extended Data Fig. 6k). Together, our results suggest that visual information plays a more significant role than tactile information in the corner coding of the subiculum.

## Subiculum neurons encode convex corners

If corner sensitivity in the subiculum has an important role in encoding environmental geometry, it would be reasonable to anticipate distinct

coding for concave versus convex corners, as these qualitative distinctions are critical for defining geometry. We next examined whether corner coding in the subiculum extended to other corner geometries. We designed more complex environments that included both concave and convex corners. We imaged as animals explored a square and rectangle environment (concave corners, 30 min), followed by three environments with convex corners (convex-1, convex-2, convex-3) (Fig. 4a). First, we identified corner cells in the square and followed their activity across other environments. As in our prior experiments, we observed corner cells that increased their spike rate at the concave corners, but less so to the convex corners (Fig. 4b). Further investigation of neurons imaged in the convex-1 environment however, revealed a small subset of neurons that increased their spike rate specifically at the convex corners (Fig. 4c). By tracking the activity of these convex corner cells to the convex-2 and -3 environments, we further found that they responded to convex corners regardless of the location of the corners or the overall geometry of the environment (Fig. 4c). Similar

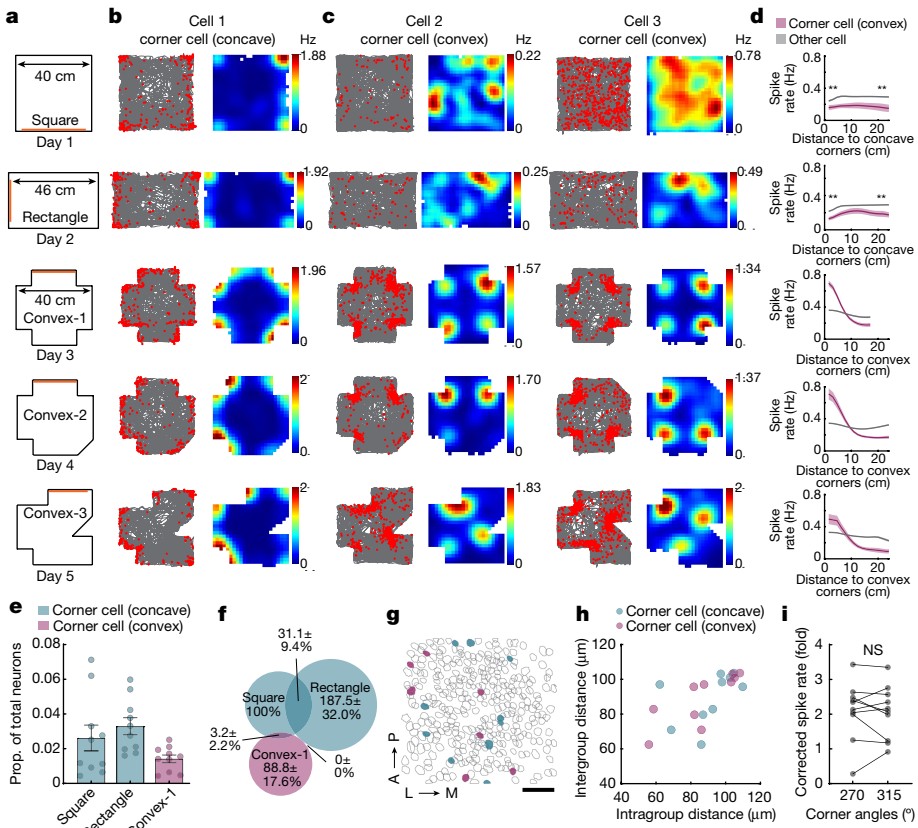

**Fig. 4 | The subiculum contains neurons that exhibit convex corner-associated activity. a**, Schematic of the arenas containing concave and convex corners. Orange bars indicate visual cues. **b**, Raster plots and rate maps of a representative concave corner cell, plotted as in Fig. 1f. **c**, Same as **b**, but for two representative convex corner cells from two mice. **d**, Positional spike rates of convex corner cells relative to the distance to the nearest convex corner, organized as in **a**. Top two plots show positional spike rates of convex-1 identified convex corner cells relative to the distance to the nearest concave corner in the square and rectangle. Spike rates between convex corner cells and other neurons were compared within approximately 5 cm of either the curve's head or tail (two-tailed Wilcoxon signed-rank test: all $P = 0.0020$; $n = 10$ mice). **e**, Proportions of corner cells in the square and rectangle and and of convex corner cells in the convex-1 arena (mean ± s.e.m.; $n = 10$ mice). **f**, Venn diagram of the overlap between concave and convex corner cells. All numbers were normalized to the corner cells in square. **g**, Anatomical locations of concave (teal) and convex (purple) corner cells from a representative mouse, identified from square and convex-1 arena, respectively. Grey, non-corner cells; A, anterior; P, posterior; L, lateral; M, medial. **h**, Pairwise intra- versus intergroup anatomical distances for concave and convex corner cells (repeated measures ANOVA: $F(1.41, 12.69) = 0.26$, $P = 0.70$; $n = 10$ mice). The intergroup distance would be greater if the neuronal groups were anatomically clustered. **i**, Corrected peak spike rates of corner cells (identified in convex-3) at the convex corners (270° versus 315°) in the convex-3 arena (two-tailed Wilcoxon signed-rank test: $P = 0.85$; $n = 10$ mice). Scale bar, 50 μm (**g**).

to corner cells that encode concave corners, corner cells encoding convex corners showed a higher spike rate near the convex corners than at the centroid (Fig. 4d, bottom three panels) ($n = 10$ mice). Tracking the activity of convex corner cells retrogradely to the square and rectangle environments, we observed that they had an overall lower spike rate compared to other subicular neurons (Fig. 4d). This low level of activity in the absence of convex corners suggests these corner cells respond specifically to convex corners. In environments with convex corners, the proportion of convex corner cells was 1.4 ± 0.2%, a slightly smaller proportion than that of concave corner cells identified in the square (2.6 ± 0.7%) and rectangle (3.3 ± 0.5%) in the same set of experiments (Fig. 4e). Corner cells encoding concave or convex corners were non-overlapping neural populations (Fig. 4f,g), as they overlapped less than expected by chance (Extended Data Fig. 7a). Corner cells encoding concave or convex corners were distributed in the subiculum in a salt and pepper pattern without clear clustering, as suggested by the similar intergroup and intragroup anatomical distances (Fig. 4g,h).

The activity of corner cells encoding convex corners was not affected by non-geometric changes to the corners, as they showed consistent spike rates for the same corner regardless of its colour or texture (Extended Data Fig. 7b–d). However, unlike corner cells that encode concave corners, corner cells encoding convex corners showed

comparable spike rates for corners at various angles in an asymmetric environment (315°, 2.06 ± 0.24; 270°, 2.10 ± 0.27 and 2.09 ± 0.13; 225°, 2.12 ± 0.20; mean ± s.e.m.) (Fig. 4c,i and Extended Data Fig. 7e–g). We then introduced a triangular and cylindrical object to the centre of the environment. Corner cells encoding convex corners showed higher spike rates at the vertices of the triangular object compared to the faces (Extended Data Fig. 7h-k). Furthermore, most of the corner cells encoding convex corners increased their spike rates around the cylinder (Extended Data Fig. 7l). Together, these results demonstrate that the subiculum encodes both concave and convex corners.

## Corner coding in the subiculum is primarily allocentric

To determine whether the previously described corner cells encode corners from an allocentric or egocentric reference frame, we first trained a linear–nonlinear Poisson (LN) model with behavioural variables including the animal's allocentric position (P), head direction (H), running speed (S) and egocentric bearing to the nearest corner (E) (Model 1, Extended Data Fig. 8a–d). We used corner cells from the square environment (40 cm) and the convex-1 environment for this analysis. For both corner cells encoding concave and convex corners, the majority (note, 15 concave and 12 convex corner cells could not

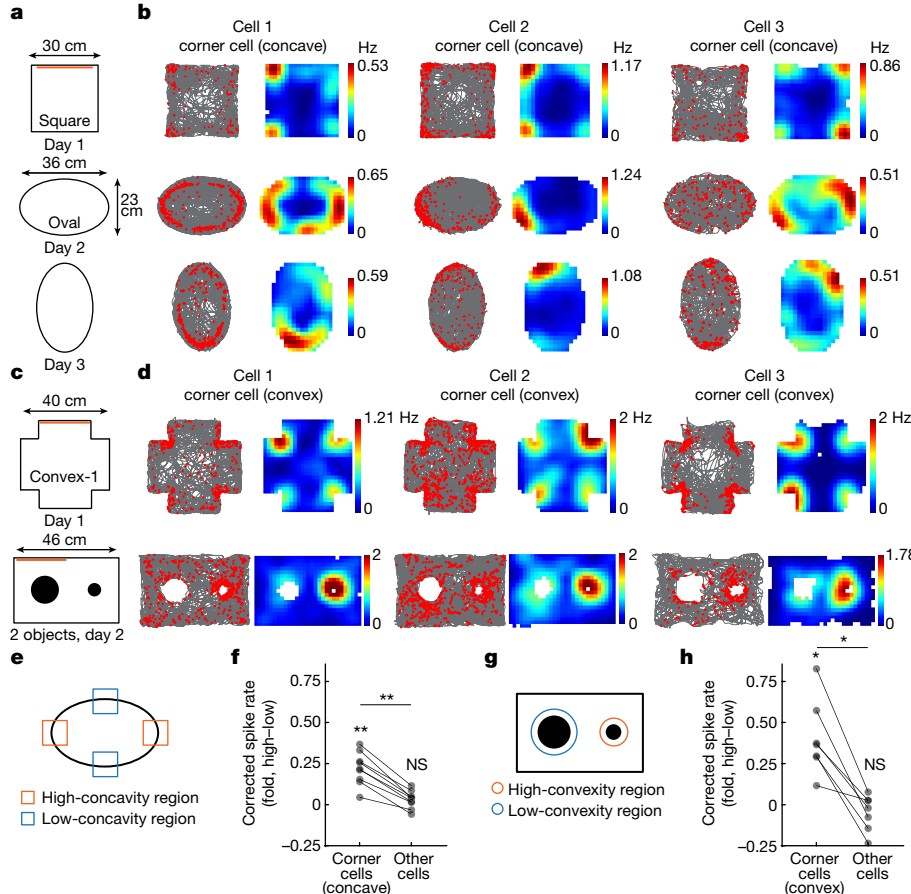

**Fig. 5 | Corner cells generalize their activity to encode environmental concavity and convexity. a**, Schematic of the arenas. Orange bars indicate visual cues. The oval on day 3 was rotated 90° relative to day 2. **b**, Raster plots and rate maps of three corner cells from three mice. Each column is a cell that was tracked across sessions, plotted as in Fig. 1f. **c**, Schematic of the experiments. Orange bars indicate visual cues. On day 2, two cylindrical objects (3 cm and 9 cm in diameter) were placed in the rectangle environment. **d**, Raster plots and rate maps of three convex corner cells from three mice. Each column is a cell which was tracked across sessions. **e**, Illustration of the high- versus low-concavity regions in the oval arena. **f**, Spike rate differences between high- and low-concavity regions in the oval arena for corner and non-corner cells. Corner

cells were identified in the square environment (two-tailed Wilcoxon signed-rank test against zero: corner cells: $P = 0.0039$; non-corner cells: $P = 0.16$; two-tailed Wilcoxon signed-rank test: corner cells versus non-corner cells: $P = 0.0039$; $n = 9$ mice, data averaged from day 2 and day 3 for each mouse). **g**, Illustration showing the high- versus low-convexity regions around the objects. **h**, Spike rate differences between high- and low-convexity regions around the objects for both corner and non-corner cells. Convex corner cells were identified in the convex-1 environment (two-tailed Wilcoxon signed-rank test against zero: corner cells, $P = 0.016$; non-corner cells, $P = 0.58$; two-tailed Wilcoxon signed-rank test: corner cells versus non-corner cells, $P = 0.016$; $n = 7$ mice).

be classified in the LN model) fell into the allocentric position only category (P), which means that adding variables did not improve the model performance (Extended Data Fig. 8e–h). A smaller number of corner cells encoded head direction, running speed and/or egocentric corner bearing in conjunction with position (Extended Data Fig. 8g,h), indicating that corner cell coding in the subiculum is largely independent of modulation by the animal's head direction, running speed and egocentric corner bearing.

Inspired by recent studies of egocentric boundary or centre-bearing cells[12,26,27,30,43,44], we expanded our investigation to consider egocentric corner coding across the entire population of subiculum neurons. We introduced additional LN models that incorporated egocentric corner bearing and distance to identify egocentric corner cells (Model 2, Extended Data Fig. 9a,b and Methods) and filtered out neurons that encoded egocentric boundaries or the centre of the environment (Models 3 and 4, Extended Data Fig. 9c and Methods). Results from both rotationally symmetric and asymmetric (for example, 30–60–90 triangle) environments consistently revealed that a small proportion of subiculum neurons (less than or equal to 0.75%) encoded corners in an egocentric reference frame (Extended Data Fig. 9d,e). This corresponded to 65 egocentric corner cells out of 12,550 total subiculum neurons,

summed from 38 sessions (square, rectangle, rightTri and convex-1 combined, $n = 10$ mice). Two-thirds of these egocentric corner cells conjunctively encoded the animal's head direction (Extended Data Fig. 9d,e). These neurons minimally overlapped (2 out of 65) with the allocentric corner cells classified in the corresponding session. Together, our results suggest that corner coding in the subiculum is primarily allocentric, a reference frame consistent with boundary vector cell (BVC) and place cell coding in the subiculum[25,33,34].

## Corner coding differs from boundary coding

We next examined the relationship between corner cells and previously reported BVCs in the subiculum[25]. We observed BVCs in the square ($10.7 \pm 0.9\%$, $n = 10$ session from 10 mice) and rectangle ($7.2 \pm 0.8\%$) environments (Extended Data Fig. 10a,b). Tracking the activity of BVCs identified in the square environment revealed stable boundary coding across both concave and convex environments (Extended Data Fig. 10a). We observed a lower than chance overlap ($3.5 \pm 1.1\%$ versus 12.5%) between BVCs and corner cells encoding concave corners (Extended Data Fig. 10c), reflective of neurons that were active at both corners and boundaries (Extended Data Fig. 10d). However, we did not

observe any overlap between BVCs and corner cells encoding convex corners (Extended Data Fig. 10c). Anatomically, BVCs and corner cells did not form distinct clusters but instead showed a salt-and-pepper distribution in the subiculum (Extended Data Fig. 10e,f). Together, this suggests that corner cells are a separate neuronal population from BVCs in the subiculum.

## Corner coding generalizes to concavity and convexity

The observation of increased activity at the boundaries of the circular environment (Fig. 1e,f) and around the cylinder in corner cells (Extended Data Fig. 7h,i,l), led us to ask whether corner coding reflect a broader coding scheme for concavity and convexity in the subiculum. To test this idea, animals explored an oval environment to examine concavity coding (Fig. 5a,b) and different sizes of cylinders (3 cm versus 9 cm in diameter) to examine convexity coding (Fig. 5c,d). Corner cells, initially identified in the square environment, were examined for their activity in the high- versus low-concavity regions of the oval (Fig. 5a,b,e,f). Indeed, corner cells encoding concave corners showed higher spike rates at the high-concavity regions compared to the low-concavity regions (oval high–low, 0.22 ± 0.03, mean ± s.e.m.) (Fig. 5e,f). Similarly, corner cells encoding convex corners, identified in the convex-1 environment, showed higher spike rates around the high-convexity cylinder compared to the low-convexity cylinder (cylinder 3 cm–9 cm, 0.41 ± 0.09) (Fig. 5c,d,g,h). These effects were not observed in non-corner cells (oval high–low, 0.03 ± 0.02; cylinder 3 cm–9 cm, −0.05 ± 0.04) and the increase in the activity of corner cells was higher than in that of non-corner cells (Fig. 5f,h). Together, our results indicate that the subiculum encodes the concave and convex curvature of the environment through distinct neuronal populations.

## Discussion

Animals use boundaries and corners to orient themselves during navigation[1–8]. These features define the geometry of an environment and can serve as landmarks or indicate locations associated with ethologically relevant needs, such as a nest site or an entryway. Here, we report that alongside neurons that encode environmental boundaries[25,29], the subiculum also contains distinct neural populations that encode concave and convex corners. This encoding is consistent across environments, with the activity of these neurons reflecting specific geometric properties of the corners, and generalized to a broader framework for coding environmental concavity and convexity. Such coding may have particular relevance to animals navigating natural environments, in which features such as burrows or nesting sites are often high in concavity or convexity.

A remaining question is how corner-specific firing patterns are generated. Given the dense CA1 to subiculum connectivity[31,32,45] and recent observations that CA1 population codes can indicate the distance to objects and walls[20], one possibility is that corner cell firing patterns arise from the convergent inputs of CA1 place cells. Namely, they could arise from a thresholded sum of the activity of place cells near environmental corners. This idea aligns with the previously observed clustering of place fields near environmental corners in CA1 place cells[37,46], and could explain the sensitivity of corner cell firing rates to corner angles, as hippocampal place fields may show more overlap in smaller corner regions. Understanding how corner-specific patterns are generated could provide important insight into the algorithms the brain uses to construct a single cell code for geometric features and future work using targeted manipulations in the hippocampus may help resolve this question[47].

Cells that explicitly encode geometric properties of an environment, such as the corner cells described here, differ from cells that respond to manipulations of an environment's geometry. For example, entorhinal grid cells transiently change the physical distance between their firing fields when a familiar box is stretched or compressed[23,48] and distort in polarized environmental geometries[16]. These changes in grid cell firing patterns represent alterations to either a familiar geometry or the geometric symmetry of the environment, but the grid pattern itself is not encoding geometric properties or specific elements that define the geometry. Likewise, changes in place cell firing rates, field locations or field size are indicative of an alteration to environmental geometry[18,19] but provide little information about the specific elements that compose the geometry. On the other hand, corner coding in the subiculum represents a geometric feature universally across environmental shapes and tracks the explicit properties of corners, including angle, height and the degree to which the walls were connected. Thus, the subiculum may be well positioned to provide information to other brain regions regarding the geometry of the environment in an allocentric reference frame. To guide behaviour however, this allocentric information needs to interface with egocentric information regarding an animal's movements[30]. One possibility is that corner cells in the subiculum provide a key input to the recently observed corner-associated activity in the lateral entorhinal cortex (LEC)[49]. Unlike corner coding in the subiculum, LEC corner-associated activity is largely egocentric and speed modulated, raising the possibility that LEC integrates allocentric corner information with egocentric and self-motion information to prepare an animal to make appropriate actions when approaching a corner or curved areas (for example, deceleration or turning).

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

## Methods

### Subjects

All procedures were conducted according to the National Institutes of Health guidelines for animal care and use and approved by the Institutional Animal Care and Use Committee at Stanford University School of Medicine and the University of California, Irvine. For subiculum imaging, eight *Camk2a-Cre; Ai163* (ref. 36) mice (four male and four female), one *Camk2-Cre* mouse (female, JAX: 005359) and one C57BL/6 mouse (male) were used. For the *Camk2-Cre* mouse, AAV1-CAG-FLEX-GCaMP7f was injected in the right subiculum at anteroposterior (AP): −3.40 mm; lateromedial (ML): +1.88 mm; and dorsoventral (DV): −1.70 mm. For the C57BL/6 mouse, AAV1-Camk2a-GCaMP6f was injected in the right subiculum at the same coordinates. For CA1 imaging, 12 *Ai94; Camk2a-tTA; Camk2a-Cre* (JAX id: 024115 and 005359) mice (seven male and five female) were used. Mice were group housed with same-sex littermates until the time of surgery. At the time of surgery, mice were 8–12 weeks old. After surgery mice were singly housed at 21–22 °C and 29–41% humidity. Mice were kept on a 12-hour light/dark cycle and had ad libitum access to food and water in their home cages at all times. All experiments were carried out during the light phase. Data from both males and females were combined for analysis, as we did not observe sex differences in, for example, corner cell proportions, spike rates to different corners angles, and concavity and convexity.

### GRIN lens implantation and baseplate placement

Mice were anesthetized with continuous 1–1.5% isoflurane and head fixed in a rodent stereotax. A three-axis digitally controlled micromanipulator guided by a digital atlas was used to determine bregma and lambda coordinates. To implant the gradient refractive index (GRIN) lens above the subiculum, a 1.8-mm-diameter circular craniotomy was made over the posterior cortex (centred at −3.28 mm anterior/posterior and +2 mm medial/lateral, relative to bregma). For CA1 imaging, the GRIN lens was implanted above the CA1 region of the hippocampus centred at −2.30 mm anterior/posterior (AP) and +1.75 mm medial/lateral (ML), relative to bregma. The dura was then gently removed and the cortex directly below the craniotomy aspirated using a 27- or 30-gauge blunt syringe needle attached to a vacuum pump under constant irrigation with sterile saline. The aspiration removed the corpus callosum and part of the dorsal hippocampal commissure above the imaging window but left the alveus intact. Excessive bleeding was controlled using a haemostatic sponge that had been torn into small pieces and soaked in sterile saline. The GRIN lens (0.25 pitch, 0.55 NA, 1.8 mm diameter and 4.31 mm in length, Edmund Optics) was then slowly lowered with a stereotaxic arm to the subiculum to a depth of −1.75 mm relative to the measurement of the skull surface at bregma. The GRIN lens was then fixed with cyanoacrylate and dental cement. Kwik-Sil (World Precision Instruments) was used to cover the lens at the end of surgery. Two weeks after the implantation of the GRIN lens, a small aluminium baseplate was cemented to the animal's head on top of the existing dental cement. Specifically, Kwik-Sil was removed to expose the GRIN lens. A miniscope was then fitted into the baseplate and locked in position so that the GCaMP-expressing neurons and visible landmarks, such as blood vessels, were in focus in the field of view. After the installation of the baseplate, the imaging window was fixed for long-term, in respect to the miniscope used during installation. Thus, each mouse had a dedicated miniscope for all experiments. When not imaging, a plastic cap was placed in the baseplate to protect the GRIN lens from dust and dirt.

### Behavioural experiments with imaging

After mice had fully recovered from the surgery, they were handled and allowed to habituate to wearing the head-mounted miniscope by freely exploring an open arena for 20 min every day for one week. The actual experiments took place in a different room from the habituation.

The behaviour rig, an 80/20 built compartment, in this dedicated room had two white walls and one black wall with salient decorations as distal visual cues, which were kept constant over the course of the entire study. For experiments described below, all the walls of the arenas were acrylic and were tightly wrapped with black paper by default to reduce potential reflections from the LEDs on the scope. A local visual cue was always available on one of the walls in the arena, except for the oval environment. In each experiment, the floors of the arenas were covered with corn bedding. All animals' movements were voluntary.

**Circle, equilateral triangle, square, hexagon and low-wall square.** This set of experiments was carried out in a circle, an equilateral triangle, a square, a hexagon and a low-wall square environment. The diameter of the circle was 35 cm. The side lengths were 30 cm for the equilateral triangle and square, and 18.5 cm for the hexagon. The height of all the environments was 30 cm except for the low-wall square, which was 15 cm. In total, we conducted 15, 18, 17, 18 and 12 sessions (20 min per session) from nine mice in the circular, triangular, square, hexagonal and low-wall square arenas, respectively. We recorded a maximum of two sessions per condition per mouse. For each mouse, we recorded 1–2 sessions in each day. If two sessions were made from the same animal on a given day, recordings were carried out from different conditions with at least a two-hour gap between sessions. For each mouse, data from this set of experiments were aligned and concatenated, and the activity of neurons was tracked across the sessions. As described above, all the walls of the arenas were black. A local visual cue (strips of white masking tape) was present on one wall of each arena, covering the top half of the wall. For CA1 imaging, mice were placed into a familiar 25 × 25 cm square environment for a single, 20 min session recording.

**Trapezoid and 30-60-90 right triangle.** This set of experiments was carried out in a right triangle (30°, 60°, 90°) and a trapezoid environment. Corner angles from the trapezoid were 55°, 90°, 90° and 125°. The dimensions of the mazes were 46 (L) × 28 (W) × 30 (H) cm. In total, we conducted 16 sessions each (25 min per session) from eight mice for the right triangle and trapezoid. Data from this set of experiments were aligned and concatenated, and the activity of neurons was tracked across the sessions for each mouse. Other recording protocols were the same as described above.

**Insertion of a discrete corner in a square environment.** This set of experiments was carried out in a large square environment with dimensions of 40 (L) × 40 (W) × 40 (H) cm. The experiments comprised a baseline session followed by four sessions with the insertion of a discrete corner into the square maze. In these sessions, the walls that formed the discrete corner were gradually separated by 0, 1.5, 3 and 6 cm. Starting from 3 cm, the animals were able to pass through the gap without difficulty. The dimensions of the inserted walls were 15 (W) × 30 (H) cm. For each condition, we recorded eight sessions (30 min per session) from eight mice by conducting a single session from each mouse per day. Data from this set of experiments were aligned and concatenated, and the activity of neurons was tracked throughout the sessions.

**Square, rectangle, convex-1, convex-2, convex-3 and convex-m1.** This set of experiments was carried out in a large square, rectangle and multiple convex environments that contained both concave and convex corners. The dimensions of the square were 40 (L) × 40 (W) × 40 (H) cm and the rectangle were 46 (L) × 28 (W) × 30 (H) cm. The convex arenas were all constructed based on the square environment using wood blocks or PVC sheets that were tightly wrapped with the same black paper. There convex corners had angles at 270° and 315° in the convex environments. Note that, for four out of ten mice, their convex-2 and -3 arenas were constructed in a mirrored layout compared to the arenas of the other six mice to control for any potential biases that could arise from the specific geometric configurations in the environment

(Fig. 4c). For convex-m1 (Extended Data Fig. 7b), the northeast convex corner was decorated with white, rough surface masking tape from the bottom all the way up to the top of the corner. For each condition, we recorded ten sessions (30 min per session) from ten mice, a single session from each mouse per day. For each mouse, data from this set of experiments were aligned and concatenated, and the activity of neurons was tracked across all the sessions.

**Convex environment with an obtuse convex corner.** This set of experiments was carried out in a convex environment that contained two 270° convex corners and one 225° convex corner (Extended Data Fig. 7e). The arena was constructed in the same manner as the other convex environments described above. For two days, we recorded a total of 18 sessions (30 min per session) from nine mice, two sessions per mouse. Please note, although the maze was rotated by 90° in the second session, we combined the two sessions together for the analysis.

**Triangular and cylindrical objects.** This set of experiments was first carried out in the convex-1 environment, followed by a 40 cm square environment containing two discrete objects (Extended Data Fig. 7h). The first object was an isosceles right triangle with the hypotenuse side measuring 20 cm in length and 7 cm in height (occasionally, animals climbed on top of the object). The second object was a cylinder with a diameter of 3 cm and a height of 14 cm. For this experiment, we recorded a total of eight sessions (30 min per session) from eight mice for each environment.

**Shuttle box.** The shuttle box consisted of two connected, 25 (L) × 25 (W) × 25 (H) cm compartments with distinct colours and visual cues (Extended Data Fig. 6a). The opening in the middle was 6.5 cm wide, so that the mouse could easily run between the two compartments during miniscope recordings. The black compartment was wrapped in black paper, but not the grey compartment. For two days, we recorded a total of 18 sessions (20 min per session) from nine mice, two sessions per mouse.

**Recordings in the dark or with trimmed whiskers.** This set of experiments was carried out in a square environment with dimensions of 30 (L) × 30 (W) × 30 × (H) cm. The animals had experience in the environment before this experiment. The experiments consisted of three sessions: a baseline session, a session recorded in complete darkness, and a session recorded after the mice's whiskers were trimmed. For the dark recording, the ambient light was turned off immediately after the animal was placed inside the square box. The red LED (approximately 650 nm) on the miniscope was covered by black masking tape. This masking did not completely block the red light, so the behavioural camera could still detect the animal's position. Before the masking, the intensity of the red LED was measured as approximately 12 lux from the distance to the animal's head. However, after the masking, the intensity of the masked red LED was comparable to the measurement taken with the light metre sensor blocked (complete darkness, approximately 2 lux). The blue LED on the miniscope was completely blocked from the outside. For the whisker-trimmed session, facial whiskers were trimmed (not epilated) with scissors until no visible whiskers remained on the face 12 h before the recording. For each condition, we recorded nine sessions (20 min per session) from nine mice by conducting a single session from each mouse per day. For each mouse, data from this set of experiments were aligned and concatenated, and the activity of neurons was tracked across these sessions. Note that according to previous reports[50–52], the number of hippocampal place cells decrease in both darkness and whisker trimming conditions.

**Square and oval.** This set of experiments was carried out in the 30 cm square environment (day 1) and an oval environment (days 2 and 3) (Fig. 5a). The oval environment had an elliptical shape, with its major axis measuring 36 cm and minor axis measuring 23 cm. Notably, the oval experiment on day 3 was rotated 90° relative to day 2 (Fig. 5a). For each condition, we recorded nine sessions (25 min per session) from nine mice, a single session from each mouse per day. For each mouse, data from this set of experiments were aligned and concatenated, and the activity of neurons was tracked across all the sessions. Data from both the oval and rotated oval conditions were combined for analysis.

**Two cylindrical objects.** This set of experiments was first carried out in the convex-1 environment, followed by a 46 (L) × 28 (W) × 30 (H) cm rectangle environment containing two cylindrical objects (Fig. 5c). The first cylinder had a diameter of 3 cm and a height of 14 cm, while the second cylinder had a diameter of 9 cm and a height of 14 cm. For this experiment, we recorded a total of seven sessions (30 min per session) for each environment from seven mice.

### Miniscope imaging data acquisition and preprocessing
Technical details for the custom-constructed miniscopes and general processing analyses are described in[32,37,53] and at http://miniscope.org/index.php/Main_Page. In brief, this head-mounted scope had a mass of about 3 g and a single, flexible coaxial cable that carried power, control signals and imaging data to the miniscope open-source data acquisition (DAQ) hardware and software. In our experiments, we used Miniscope v.3, which had a 700 μm × 450 μm field of view with a resolution of 752 pixels × 480 pixels (approximately 1 μm per pixel). For subiculum imaging, we measured the effective image size (the area with detectable neurons) for each mouse and combined this information with histology. The anatomical region where neurons were recorded was approximately within a 450-μm diameter circular area centred around AP: −3.40 mm and ML: +2 mm. Owing to the limitations of 1-photon imaging, we believe the recordings were primarily from the deep layer of the subiculum. Images were acquired at approximately 30 frames per second (fps) and recorded to uncompressed avi files. The DAQ software also recorded the simultaneous behaviour of the mouse through a high-definition webcam (Logitech) at approximately 30 fps, with time stamps applied to both video streams for offline alignment.

For each set of experiments, miniscope videos of individual sessions were first concatenated and down-sampled by a factor of two, then motion corrected using the NoRMCorre MATLAB package[54]. To align the videos across different sessions for each animal, we applied an automatic two-dimensional (2D) image registration method (github.com/fordanic/image-registration) with rigid $x$–$y$ translations according to the maximum intensity projection images for each session. The registered videos for each animal were then concatenated together in chronological order to generate a combined dataset for extracting calcium activity.

To extract the calcium activity from the combined dataset, we used extended constrained non-negative matrix factorization for endoscopic data (CNMF-E)[38,55], which enables simultaneous denoising, deconvolving and demixing of calcium imaging data. A key feature includes modelling the large, rapidly fluctuating background, allowing good separation of single-neuron signals from background and the separation of partially overlapping neurons by taking a neuron's spatial and temporal information into account (see ref. 38 for details). A deconvolution algorithm called OASIS[39] was then applied to obtain the denoised neural activity and deconvolved spiking activity (Extended Data Fig. 1b). These extracted calcium signals for the combined dataset were then split back into each session according to their individual frame numbers. As the combined dataset was large (greater than 10 GB), we used the Sherlock HPC cluster hosted by Stanford University to process the data across 8–12 cores and 600–700 GB of RAM. While processing this combined dataset required significant computing resources, it enhanced our ability to track cells across sessions from different days. This process made it unnecessary to perform individual footprint alignment or cell registration across sessions. The position, head direction and speed of

the animals were determined by applying a custom MATLAB script to the animal's behavioural tracking video. Time points at which the speed of the animal was lower than 2 cm s$^{-1}$ were identified and excluded from further analysis. We then used linear interpolation to temporally align the position data to the calcium imaging data.

## Corner cell analyses

**Calculation of spatial rate maps.** After we obtained the deconvolved spiking activity of neurons, we binarized it by applying a threshold using a ×3 standard deviation of all the deconvolved spiking activity for each neuron. The position data was sorted into 1.6 × 1.6 cm non-overlapping spatial bins. The spatial rate map for each neuron was constructed by dividing the total number of calcium spikes by the animal's total occupancy in a given spatial bin. The rate maps were smoothed using a 2D convolution with a Gaussian filter that had a standard deviation of two.

**Corner score for each field.** To detect spatial fields in a given rate map, we first applied a threshold to filter the rate map. After filtering, each connected pixel region was considered a place field, and the $x$ and $y$ coordinates of the regional maxima for each field were the locations of the fields. We used a filtering threshold of 0.3 times the maximum spike rate for identifying corner cells in smaller environments (for example, the circle, triangle, square and hexagon), and a filtering threshold of 0.4 for identifying corner cells in larger environments (for example, 40 cm square, rectangle and convex environments, Fig. 4). These thresholds were determined from a search of threshold values that ranged from 0.1–0.6. The threshold range that resulted in the best corner cell classification, as determined by the overall firing-rate difference between the corner and the centroid of an environment (for example, Fig. 1h), was 0.3–0.4 across different environments. The coordinates of the centroid and corners of the environments were automatically detected with manual corrections. For each field, we defined the corner score as:

$$\text{cornerscore}_{\text{field}} = \frac{d1 - d2}{d1 + d2}$$

where $d1$ is the distance between the environmental centroid and the field, and $d2$ is the distance between the field and the nearest environmental corner. The score ranges from −1 for fields situated at the centroid of the arena to +1 for fields perfectly located at a corner (Extended Data Fig. 1f).

**Corner score for each cell.** There were two situations that needed to be considered when calculating the corner score for each cell (Extended Data Fig. 1g). First, if a cell had $n$ fields in an environment that had $k$ corners ($n \le k$), the corner score for that cell was defined as:

$$\text{cornerscore}_{\text{cell}} = \frac{\sum_n \text{cornerscore}_{\text{field}}}{k}, (n \le k);$$

Second, if a cell had more fields than the number of environmental corners ($n > k$), the corner score for that cell was defined as the sum of the top $k$th corner scores minus the sum of the absolute values of the corner scores for the extra fields minus one, and divided by $k$. Namely,

$$\text{cornerscore}_{\text{cell}} = \frac{\sum_{\text{top}(n,k)} \text{cornerscore}_{\text{field}} - \sum_{\text{extra}} | \text{cornerscore}_{\text{field}} - 1 |}{k}, (n > k)$$

where top($n,k$) indicates the fields (also termed 'major fields') that have the top $k$th cornerscore$_{\text{field}}$ out of the $n$ fields, and 'extra' refers to the corner scores for the remaining fields (Extended Data Fig. 1g). In this case, the absolute values of the corner scores for the extra fields were used to penalize the final corner score for the cell, so that the score decreased if the cell had too many fields. The penalty for a given extra

field ranged from 0 to 2, with 0 for the field at the corner and 2 for the field at the centre. As a result, as the extra field moves away from a corner, the penalty for the overall corner score gradually increases. Note, among all the corner cells identified in the triangle, square and hexagon environments, only 7.8 ± 0.5% (mean ± s.e.m.; $n$ = 9 mice) of them were classified under this situation.

**Final definition of corner cells.** To classify a corner cell, the timing of calcium spikes for each neuron was circularly shuffled 1,000 times. For each shuffle, spike times were shifted randomly by 5–95% of the total data length, rate maps were regenerated and the corner score for each cell was recalculated. Note, for the recalculation of corner scores for the shuffled rate maps, we did not use the aforementioned penalization process. This is because shuffled rate maps often exhibited a greater number of fields than the number of corners, and thus applying the penalization lowers the 95th percentile score of the shuffled distribution (that is, more neurons would be classified as corner cells). Thus, not using this penalization process in calculating shuffled corner scores kept the 95th percentile of the shuffled distribution as high as possible for each cell to ensure a stringent selection criteria for corner cells (Extended Data Fig. 2a–d). Alternatively, we also attempted to generate the null distribution by shuffling the locations of place fields directly on the original rate map. Although the two methods gave similar results in terms of characterizing corner cells, the latter approach tended to misclassify neurons with few place fields as a corner cell (for example, a neuron has only one field and the field is in the corner). Therefore, we used the former shuffling method to generate the null distribution. Finally, we defined a corner cell as a cell: (1) whose corner score passed the 95th percentile of the shuffled score (Extended Data Fig. 1h,i), (2) whose distance between any two fields (major fields, if the number of fields is greater than the number of corners) was greater than half the distance between the corner and centroid of the environment (Extended Data Fig. 1j) and (3) whose within-session (two halves) stability was higher than 0.3 (Extended Data Fig. 1k), as determined by the 95th percentile of the random within-session stability distribution using shuffled spikes.

**Identification of convex corner cells.** To identify convex corner cells, we used similar methods as described above for the concave corner cells, with a minor modification. Namely, after the detection of the field locations on a rate map, we applied a polygon mask to the map using the locations of convex corners as vertices. This polygon mask was generated using the build-in function poly2mask in MATLAB. We then considered only the extracted polygon region for calculating corner scores and corresponding shuffles. The reason for using the polygon mask is to avoid nonlinearity in corner score calculation in the convex environment, in particular, when the distance between the location of a field (for example, a field at a concave corner in the convex-1 environment) and the environment centre is greater than the distance between the centre and the convex corner.

**Measuring the peak spike rate at corners.** To measure the peak spike rate at each corner of an environment, we first identified the area near the corner using a 2D convolution between two matrices, $M$ and $V$. $M$ is the same size as the rate map, containing all zero elements except for the corner bin, which is set to one. $V$ is a square matrix containing elements of ones and can be variable in size. For our analysis, we used a 12 × 12 matrix $V$, which isolated a corresponding corner region equal to approximately 10 cm around the corner. We then took the maximum spike rate in the region as the peak spike rate at the corner. For some specific analyses, due to the unique position or geometry of the region of interest (for example, the inserted discrete corner and objects), we decreased the size of the matrix $V$ to obtain a more restricted region of interest for measurement. Specifically, we measured approximately 5 cm around the discrete corner (Fig. 3), approximately 5 cm around

the vertices and faces of the triangular object (Extended Data Fig. 7) and approximately 5 cm outside of the cylinders (Fig. 5). To ensure the robustness of our findings, we tried various sizes of the 2D convolution in our analyses, and found that the results were largely consistent with those presented in the manuscript.

**Corrections of spike rates on the rate map.** When comparing spike rates across different corners, it is important to consider the potential impact of the animal's occupancy and movement patterns on the measurements (Extended Data Fig. 4a–f). To account for any measurements that might have been associated with the animal's behaviour, we generated a simulated rate map using a simulated neuron that fired along the animal's trajectory using the animal's measured speed at the overall mean spike rate observed across all neurons of a given mouse (Extended Data Fig. 4c). We then used the raw rate map divided by the simulated rate map to obtain the corrected rate map (Extended Data Fig. 4e). This method ensured that behaviour-related factors were present in both the raw and simulated rate maps, and therefore were removed from the corrected rate map (Extended Data Fig. 4a–f).

**Measuring paired-wise anatomical distances.** To measure the pairwise anatomical distances between neurons, we calculated the Euclidian distance between the centroid locations of each neuron pair under the imaging window for each mouse. We then quantified the average intragroup and intergroup distances for each neuron based on its group identity (for example, concave versus convex corner cells). The final result for each group was averaged across all the neurons. We hypothesized that if functionally defined neuronal groups were anatomically clustered, the intergroup distance would be greater than the intragroup distance.

**Boundary vector cell analyses**
Rate maps of all the neurons were generated by dividing the open arena into 1.6 cm × 1.6 cm bins and calculating the spike rate in each bin. The maps were smoothed using a 2D convolution with a Gaussian filter that had a standard deviation of 2. To detect boundary vector cells (BVCs), we used a method based on border scores, which we calculated as described previously[29,56]:

$$\text{borderscore} = \frac{\text{CM} - \text{DM}}{\text{CM} + \text{DM}}$$

where CM is the proportion of high firing-rate bins located along one of the walls and DM is the normalized mean product of the firing rate and distance of a high firing-rate bin to the nearest wall. We identified BVCs as cells with a border score above 0.6 and whose largest field covered more than 70% of the nearest wall and whose within-session stability was higher than 0.3. Additionally, BVCs needed to have significant spatial information (that is, as in place cells, described below). Of note, our conclusion regarding BVCs and corner cells remained the same when we varied the wall coverage from 50% to 90% for classifying BVCs.

**Place cell analyses**
**Spatial information and identification of place cells.** To quantify the information content of a given neuron's activity, we calculated spatial information scores in bits per spike (that is, calcium spike) for each neuron according to the following formula[57],

$$\text{bits per spike} = \sum_{i=1}^{n} P_i \frac{\lambda_i}{\lambda} \log_2 \frac{\lambda_i}{\lambda},$$

where $P_i$ is the probability of the mouse occupying the $i$th bin for the neuron, $\lambda_i$ is the neuron's unsmoothed event rate in the $i$th bin, while $\lambda$ is the mean rate of the neuron across the entire session. Bins with total occupancy time of less than 0.1 s were excluded from the calculation.

To identify place cells, the timing of calcium spikes for each neuron was circularly shuffled 1,000 times and spatial information (bits per spike) recalculated for each shuffle. This generated a distribution of shuffled information scores for each individual neuron. The value at the 95th percentile of each shuffled distribution was used as the threshold for classifying a given neuron as a place cell, and we excluded cells with an overall mean spike rate less than the 5th percentile of the mean spike rate distribution (that is, approximately 0.1 Hz) of all the neurons in that animal.

**Position decoding using a naïve Bayes classifier**
We used a naive Bayes classifier to estimate the probability of animal's location given the activity of all the recorded neurons. The method is described in detail in our previous publication[37]. In brief, the binarized, deconvolved spike activity from all neurons was binned into non-overlapping time bins of 0.8 s. The $M \times N$ spike data matrix, where $M$ is the number of time bins and $N$ is the number of neurons, was then used to train the decoder with an $M \times 1$ vectorized location labels (namely, concatenating each column of position bins vertically). The posterior probability of observing the animal's position $Y$ given neural activity $X$ could then be inferred from the Bayes rule as:

$$P(Y=y|X_1, X_2..., X_N) = \frac{P(X_1, X_2, ..., X_N|Y=y)P(Y=y)}{P(X_1, X_2, ..., X_N)},$$

where $X = (X_1, X_2, ... X_N)$ is the activity of all neurons, $y$ is one of the spatial bins that the animal visited at a given time, and $P(Y=y)$ is the prior probability of the animal being in spatial bin $y$. We used an empirical prior as it showed slightly better performance than a flat prior. $P(X_1, X_2, ..., X_N)$ is the overall firing probability for all neurons, which can be considered as a constant and does not need to be estimated directly. Thus, the relationship can be simplified to:

$$\hat{y} = \underset{y}{\text{argmax}} \; P(Y=y) \prod_{i=1}^{N} P(X_i|Y=y),$$

where $\hat{y}$ is the animal's predicted location, based on which spatial bin has the maximum probability across all the spatial bins for a given time. To estimate $P(X_i|Y=y)$, we applied the built-in function fitcnb in MATLAB to fit a multinomial distribution using the bag-of-tokens model with Laplace smoothing.

To reduce occasional erratic jumps in position estimates, we implemented a two-step Bayesian method by introducing a continuity constraint[58], which incorporated information regarding the decoded position in the previous time step and the animal's running speed to calculate the probability of the current location y. The continuity constraint for all the spatial bins $Y$ at time $t$ followed a 2D gaussian distribution centred at position $y_{t-1}$, which can be written as:

$$\mathcal{N}(y_{t-1}, \sigma_t^2) = c \times \exp\left(\frac{-\|y_{t-1} - Y\|^2}{2\sigma_t^2}\right),$$

$$\sigma_t = a v_t,$$

where $c$ is a scaling factor and $v_t$ is the instantaneous speed of the animal between time $t-1$ and $t$. $v_t$ is scaled by $a$, which is empirically selected as 2.5. The final reconstructed position with two-step Bayesian method can be further written as:

$$\hat{y}_{2\text{step}} = \underset{y}{\text{argmax}} \; \mathcal{N}(y_{t-1}, \sigma_t^2)P(Y=y) \prod_{i=1}^{N} P(X_i|Y=y).$$

Decoded vectorized positions were then mapped back onto 2D space. The final decoding error was averaged from ten-fold cross-validation.

For each fold, the decoding error was calculated as the mean Euclidean distance between the decoded position and the animal's true position across all time bins.

To test the contribution of corner cells to spatial coding, we first trained the decoder using all neurons and then replaced the neural activity of corner cells with vectors of zeroes from the test data before making predictions. It is important to note that this activity removal procedure was only applied to the data used for predicting locations and not for training, as ablating neurons directly from the training data will result in the model learning to compensate for the missing information[59]. We performed this analysis using ten-fold cross-validation for each mouse. To compare the performance of the corner cell removed decoder to the full decoder, we first calculated the 2D decoding error map of a session for each condition, and then obtained a map for error ratio by dividing the error map from the corner cell removed decoder by the error map from the full decoder (Extended Data Fig. 2g). We then compared the error ratio at the corners of the environment to the centre of the environment. For quadrant decoding in the square environment (Fig. 1l), we trained and tested the decoder using only the identified corner cells without the two-step constraint using ten-fold-cross-validation. For the shuffled condition, the decoder was trained and tested for 100 times using circularly shuffled calcium spikes over time. The probability in the correct quadrant was compared between the corner cell trained and shuffled decoders. For decoding the geometry of different environments (Extended Data Fig. 5d–f), we concatenated the data (time bin = 400 ms) with neurons tracked from circle, triangle, square and hexagon environments for each animal. The data was then resampled from an 8 cm diameter circular area either in the centre or near the corner/boundary of the environment. The data length was matched between the two areas and the decoding labels for each environment were identical (numerical, 1 for circle, 2 for square, 3 for triangle, 4 for hexagon). Then the decoder was trained and tested for each mouse using 10-fold-cross-validation.

### Visualization of low-dimensional neural manifold

We implemented a two-step dimensionality reduction method based on a prior publication[42]. First, we took the binarized, deconvolved spike activity from all neurons for each session (time bin size = 67 ms) and convolved it with a Gaussian filter with $\sigma$ = 333 ms. As a result, each column of the matrix represents the smoothed firing rate of each cell over time. Then, we $z$-scored the smoothed firing rate of each cell. Next, we proceeded with dimensionality reductions on this smoothed and $z$-scored data matrix (number of time bins × number of neurons). First, to improve robustness to noise, we performed a principal component analysis (PCA) on the data matrix. Next, we selected the top ten principal components from the PCA results to carry out Uniform Manifold Approximation and Projection (UMAP), reducing the ten principal components into a 3D visualization. The parameters for this UMAP were set as follows: min_dist = 0.1, n_neighbors = 100 and n_components = 3. Note that the general structure of the low-dimensional neural manifold remained largely the same when we varied the number of principal components from 5 to 30 and adjusted the parameters for UMAP.

### Linear–nonlinear Poisson (LN) model

**Calculation of allocentric and egocentric corner bearing.** For each time point in the recording session, the allocentric bearing of the animal to the nearest corner (Extended Data Fig. 8b) was calculated using the $x$, $y$ coordinates of the corners and the animal as follows:

$$\text{cornerbearing}_{\text{allocentric}} = \arctan 2(y_{\text{corner}} - y_{\text{animal}}, x_{\text{corner}} - x_{\text{animal}})$$

Similarly, allocentric bearings to the nearest walls or centre of the environment was calculated as:

$$\text{wallbearing}_{\text{allocentric}} = \arctan 2(y_{\text{wall}} - y_{\text{animal}}, x_{\text{wall}} - x_{\text{animal}})$$

$$\text{centerbearing}_{\text{allocentric}} = \arctan 2(y_{\text{center}} - y_{\text{animal}}, x_{\text{center}} - x_{\text{animal}})$$

We then derived the egocentric corner bearing of the animal (Extended Data Fig. 8a–c) by subtracting the animal's allocentric head direction from the allocentric corner bearing:

$$\text{cornerbearing}_{\text{egocentric}} = \text{cornerbearing}_{\text{allocentric}} - \text{head direction}$$

Note that a corner bearing of 0 degrees indicates that the corner was directly in front of the animal, as illustrated in Extended Data Fig. 8c. Similarly, egocentric bearing to the nearest walls or centre were calculated as follows:

$$\text{wallbearing}_{\text{egocentric}} = \text{wallbearing}_{\text{allocentric}} - \text{head direction}$$

$$\text{centerbearing}_{\text{egocentric}} = \text{centerbearing}_{\text{allocentric}} - \text{head direction}$$

**Implementation of the linear–nonlinear Poisson (LN) model.** The LN model is a generalized linear model (GLM) framework which allows unbiased identification of functional cell types encoding multiplexed navigational variables. This framework was described in a previous publication[60] and here, we applied the same method to our calcium imaging data in the subiculum. Briefly, for Model 1 in Extended Data Fig. 8, 15 models were built in the LN framework, including position (P), head direction (H), speed (S), egocentric corner bearing (E), position & head direction (PH), position & speed (PS), position & egocentric corner bearing (PE), head direction & speed (HS), head direction & egocentric bearing (HE), speed & egocentric bearing (SE), position & head direction & speed (PHS), position & head direction & egocentric bearing (PHE), position & speed & egocentric bearing (PSE), head direction & speed & egocentric bearing (HSE) and position & head direction & speed & egocentric bearing (PHSE). For each model, the dependence of spiking on the corresponding variable(s) was quantified by estimating the spike rate ($r_t$) of a neuron during time bin $t$ as an exponential function of the sum of variable values (for example, the animal's position at time bin $t$, indicated through an 'animal-state' vector) projected onto a corresponding set of parameters (Extended Data Fig. 8d). This can be mathematically expressed as:

$$\mathbf{r} = \frac{\exp(\sum_i X_i^T \mathbf{w}_i)}{\mathrm{d}t}$$

where $\mathbf{r}$ is a vector of firing rates for one neuron over $T$ time points, $i$ indexes the variable ($i \in [P, H, S, E]$), $X_i$ is the design matrix in which each column is an animal-state vector $\mathbf{x}_i$ for variable $i$ at one time bin, $\mathbf{w}_i$ is a column vector of learned parameters that converts animal-state vectors into a firing-rate contribution and d$t$ is the time bin width.

We used the binarized deconvolved spikes as the neuron spiking data with a time bin width equal to 500 ms. The design matrix contained the animal's behavioural state, in which we binned position into 2 cm² bins, head direction and egocentric corner bearing into 20-degree bins, and speed into 2 cm s⁻¹ bins. Each vector in the design matrix denotes a binned variable value. All elements of this vector are 0, except for a single element that corresponds to the bin of the current animal-state. To learn the variable parameters $\mathbf{w}_i$, we used the built-in fminunc function in MATLAB to maximize the Poisson log-likelihood of the observed spike train ($n$) given the model spike number ($r \times$ d$t$) and under the prior knowledge that the parameters should be smooth. Model performance for each cell is computed as the increase in Pearson's correlation (between the predicted and the true firing rates) of the model compared to the 95th percentile of shuffled correlations (true firing rate was circularly shuffled for 500 times). Performance was quantified through ten-fold cross-validation, where each fold is a random selection of 10% of the data. To determine the best fit

model for a given neuron, we used a heuristic forward-search method that determines whether adding variables significantly improved model performance ($P < 0.05$ for a one-sided sign-rank test, $n = 10$ cross-validation folds).

**Using LN models to identify egocentric corner cells.** To identify egocentric corner coding in an unbiased manner, we replaced the allocentric position (P) in Model 1 with egocentric corner distance (D, bin size = 2 cm) to facilitate the identification of egocentric corner cells (Model 2, Extended Data Fig. 9a). However, encoding for egocentric corner bearing, particularly in rotationally symmetric environments, could potentially be confounded by other correlated variables, such as egocentric wall bearing (circular correlation with corner bearing = 0.43)[27,44] or egocentric centre bearing (circular correlation with corner bearing = −0.73)[12]. To rule out the possibility that the observed encoding for egocentric corner bearing in Model 2 was actually due to encoding for egocentric wall or centre bearing, we next trained two separate LN models in which egocentric corner bearing and corner distance was replaced by egocentric wall bearing and wall distance (Model 3, Extended Data Fig. 9c), or with egocentric centre bearing and centre distance (Model 4, Extended Data Fig. 9c). As Models 2, 3 and 4 were trained and tested using the same data, we compared the model fitting of neurons with egocentric corner modulation in Model 2 to the fitting of the same neurons in Model 3 and Model 4. Neurons that exhibited a significantly better fit (higher increased correlation, $n = 10$-fold) in Model 2 compared to Model 3 or 4 were considered as potential neurons encoding egocentric corner bearing. Finally, to rule out the possibility that egocentric corner coding could artifactually result from the conjunction of position and head direction[12], we also compared the neurons' fittings in Model 2 to the position and head direction groups (P, H, PH, PHS) in Model 1 (Extended Data Fig. 8). Neurons that met these criteria were considered as significantly encoding corners in an egocentric reference frame.

To further disentangle the correlations among egocentric bearing variables in rectilinear environments, we repeated the same analysis (as described above) in the right triangle environment. In the right triangle, the circular correlation between corner and wall bearings decreased to 0.09, and the correlation between corner and centre bearings shifted to −0.38. Correlations between egocentric distances also shifted by 0.2 to 0.4 towards zero. Thus, in the right triangle environment, tuning between corner versus wall/centre becomes sufficiently distinct.

## Histology

After the imaging experiments were concluded, mice were deeply anesthetized with isoflurane and transcardially perfused with 10 ml of phosphate-buffered saline (PBS), followed by 30 ml of 4% paraformaldehyde-containing phosphate buffer. The brains were removed and left in 4% paraformaldehyde overnight. The next day, samples were transferred to 30% sucrose in PBS and stored in 4 °C. At least 24 h later, the brains were sectioned coronally into 30-μm-thick samples using a microtome (Leica SM2010R, Germany). All sections were counterstained with 10 μM DAPI, mounted and cover-slipped with antifade mounting media (Vectashield). Images were acquired by an automated fluorescent slide scanner (Olympus VS120-S6 slide scanner, Japan) under ×10 magnification.

## Data inclusion criteria and statistical analysis

After a certain period postsurgery, the imaging quality began to decline in some animals, and this thus led to slight variations in the number of mice used in each set of experiments, ranging from 7 to 10. We evaluated the imaging quality for each mouse before executing each set of experiments. No mice were excluded from the analyses as long as the experiments were executed. For experiments with two identical sessions for a given condition (for example, Figs. 1 and 2), sessions with less

than 3 identified corner cells were excluded to minimize measurement noise in spike rates. This criterion only resulted in the exclusion of one session from one mouse in Fig. 2e.

Analyses and statistical tests were performed using MATLAB (2020a) and GraphPad Prism 9. Data are presented as mean ± s.e.m. For normality checks, different test methods (D'Agostino and Pearson, Anderson–Darling, Shapiro–Wilk and Kolmogorov–Smirnov) indicated only a portion of the data in our statistical analyses followed a Gaussian distribution. Thus, a two-tailed Wilcoxon signed-rank test was used for two-group comparisons throughout the study. We also validated that conducting statistical analyses with a two-tailed paired $t$-test yielded consistent results and did not alter any conclusions. For statistical comparisons across more than two groups, repeated measures analysis of variance (ANOVA) was used before pairwise comparisons. All statistical tests were conducted on a per-mouse basis. In cases where an experiment involved two sessions, the data were averaged across these sessions, as indicated in the corresponding text or figure legend. For example, in Fig. 1g, the proportion of corner cells was determined by averaging the proportions of corner cells in session 1 (a single number) and session 2 (a single number). Similarly, in Fig. 1l, the decoding accuracy for each mouse was averaged using the mean decoding accuracy of session 1 (a single number) and session 2 (a single number). In all experiments, the level of statistical significance was defined as $P \le 0.05$.

### Reporting summary

Further information on research design is available in the Nature Portfolio Reporting Summary linked to this article.

## Data availability

Calcium imaging data generated in this study are available on Mendeley Data: https://doi.org/10.17632/5sj8d5vtg2.1. Source data are provided with this paper.

## Code availability

The codes for replicating the analyses in this study have been deposited on GitHub at: https://github.com/yanjuns/Sun_et_al_2024_Nature.

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

**Acknowledgements** We thank Z. Yu for helpful input in designing the method for characterizing corner cells, H. Zeng for providing the Ai163 mouse line, S. Solis for histology assistance, and E. Velazquez and G. Wu for animal husbandry assistance. This work was supported by National

Institute of Health Grants 1R01MH126904-01A1 (L.M.G.), R01MH130452 (L.M.G.), BRAIN Initiative U19NS118284 (L.M.G.), the Vallee Foundation (L.M.G.), the James S. McDonnell Foundation (L.M.G.), the Simons Foundation 542987SPI (L.M.G.), National Institute of Health Grants R01NS104897 (X.X. and D.A.N.), RF1 AG065675 (X.X.), and K01DA058743 (Y.S.).

**Author contributions** Y.S., D.A.N. and L.M.G. contributed to the conceptualization of the study. X.X. and Y.S. contributed to the methodology. Y.S. performed the investigation and visualization. The funding acquisition was performed by X.X. and L.M.G. Y.S. and L.M.G. contributed to the project administration, supervision and writing of the original draft. Y.S., X.X., D.A.N. and L.M.G. contributed to the reviewing and editing of the paper.

**Competing interests** The authors declare no competing interests.

**Additional information**
**Correspondence and requests for materials** should be addressed to Yanjun Sun or Lisa M. Giocomo.

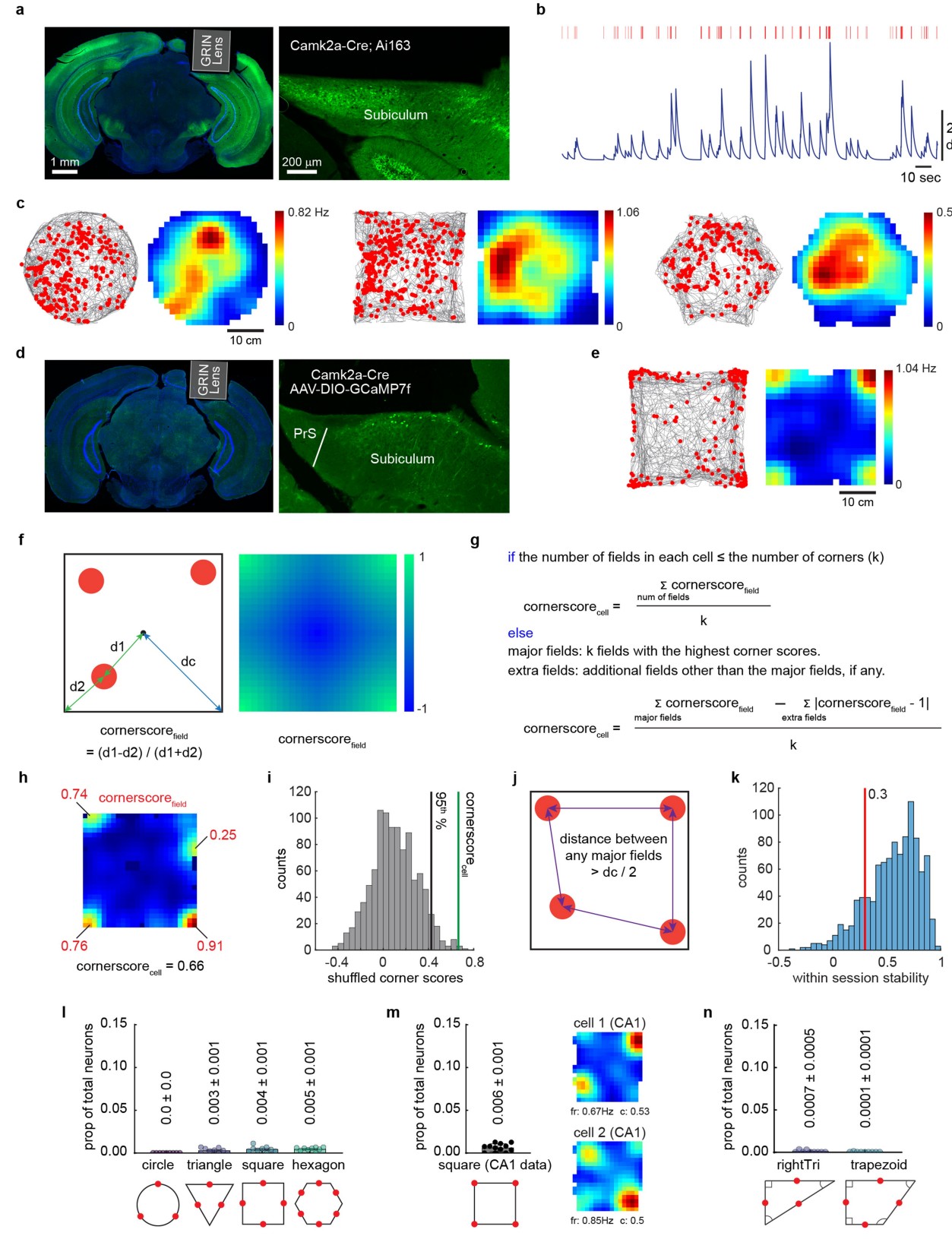

**Extended Data Fig. 1** | See next page for caption.

**Extended Data Fig. 1 | Histology and the development of a score to classify corner cells.** (**a**) Histology of GRIN lens implantation in the dorsal subiculum of an example Camk2a-Cre; Ai163 mouse. Green: GCaMP6, Blue: DAPI. Right, enlarged view of GCaMP6-expressing subiculum neurons. The experiment was replicated in 8 mice with similar results. (**b**) Representative de-noised calcium signal traces (dark blue) and de-convolved inferred spikes (red bars) extracted from CNMF-E. (**c**) Raster plots and rate maps of a representative place cell in the subiculum. Raster plot (left) indicates extracted spikes (red dots) on top of the animal's running trajectory (grey lines) and the spatial rate map (right) is colour-coded for maximum (red) and minimum (blue) values. Activity was tracked across different environments. (**d**) Same as (a), but from a Camk2a-Cre mouse with AAV-DIO-GCaMP7f injected in the subiculum. GCaMP expression was restricted to the subiculum. PrS: presubiculum. (**e**) An example corner cell from the animal in (d), plotted as in (c). (**f**) Left, the definition of corner score for a given spatial field (cornerscore$_{field}$). d1: distance from the centre of the arena to the field; d2: distance from the field to the nearest corner; dc: the mean distance from the corners to the centre of the arena. Right, the distribution of cornerscore$_{field}$ in a square environment. This represents the expected corner score if a neuron were active in a given pixel of this plot. Note that the cornerscore can range from −1 (blue) to 1 (green). (**g**) The definition of the corner score for a given cell (cornerscore$_{cell}$, see Methods). (**h**) An example corner cell with corner score values for each field labeled in red, the final corner score for this cell is shown below. (**i**) Shuffling of cornerscore$_{cell}$ to determine a threshold for classifying a neuron as a corner cell. This example is from the same cell as in (h). See also Extended Data Fig. 2a–d and Methods. (**j**) To be classified as a corner cell, the distance between any two fields (major fields, if the number of fields > number of corners) needed to be greater than half of the dc value, as indicated by the blue line in (f). (**k**) As an additional criterion, to be classified as a corner cell, within-session stability needed to be greater than 0.3 (Pearson's correlation between the two halves of the data). The distribution shows the within-session stability of all corner cells from the triangle, square, and hexagon sessions before applying this criterion (n = 1018 cells from 9 mice). (**l**) Proportion of neurons that passed the definition for a corner cell when corners of the environments were manually assigned to the walls. Red dots in the bottom schematic denote the locations that were assigned as 'corners'. (**m**) Left: Proportion of corner cells in CA1 (n = 12 mice). Right: Rate maps of two example CA1 corner cells. Peak spike rates (fr) and corner scores (c) for the cells are indicated at the bottom. (**n**) Same as (l), but for the right triangle and trapezoid environments.

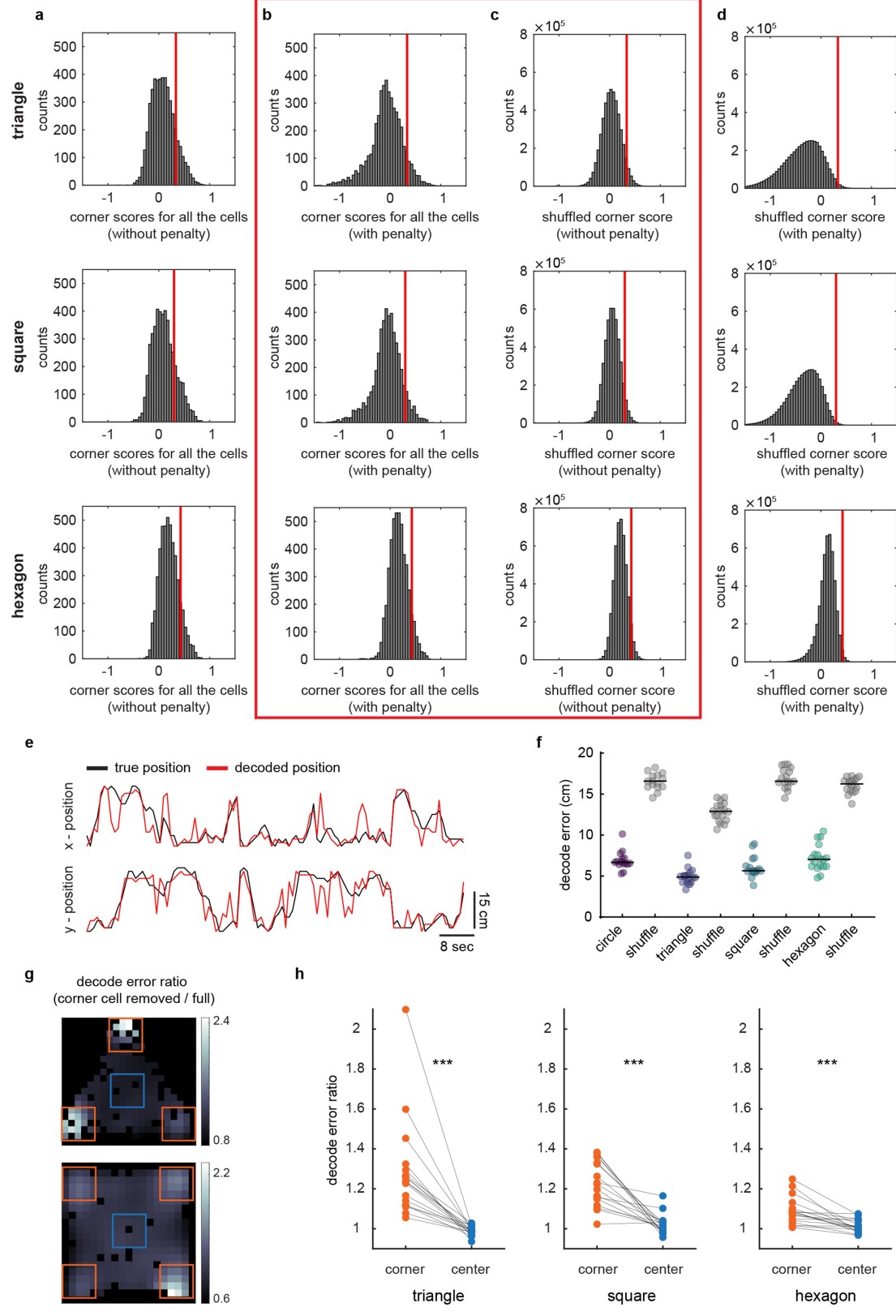

**Extended Data Fig. 2** | See next page for caption.

**Extended Data Fig. 2 | Corner score shuffling and spatial decoding.**
(**a**) Distributions of corner scores calculated without applying the penalty (Methods) for all subiculum neurons recorded in the triangle (n = 4774 cells from 18 sessions of 9 mice), square (n = 4685 cells from 17 sessions of 9 mice), and hexagon (n = 4774 cells from 18 sessions of 9 mice) environments. The red lines represent the 95th percentile of the distributions in (b). Note: This calculation is for display purposes only and was not used for the analyses in this paper, also see Methods for details. (**b**) Distributions of corner scores calculated with penalty applied (Extended Data Fig. 1g), which was used in the analyses. The red lines represent the 95th percentile of the corresponding distributions. (**c**) Distributions of shuffled corner scores calculated without applying the penalty, which was used in the analyses. Each cell was shuffled 1000 times. The red lines represent the 95th percentile of the distributions in (b). The red box over (b) and (c) indicates the method used in corner score calculation and shuffling procedures in the current work, which resulted in the most stringent corner score criteria for defining corner cells. (**d**) Same as c, but distributions of shuffled corner scores calculated with penalty applied. The red lines represent the 95th percentile of the distributions in (b). Note: this method was not used in this paper. (**e**) An example of the true vs. decoded spatial x-y position using the full decoder (all recorded subiculum neurons). (**f**) Decoding performance of the full decoder in different environments. The decoder was trained and tested within each session using 10-fold cross-validation. Each dot is a session, black lines represent the median. Decoding errors were compared with the corresponding shuffle within each condition (two-tailed Wilcoxon signed-rank test: all p < 0.0001; n = 15, 18, 17, and 18 sessions for circle, triangle, square, and hexagon, respectively, from 9 mice). (**g**) Decoding error ratio from a triangle (top) or square (bottom) session, color-coded for larger (white) and smaller (black) error ratios. The error ratio was obtained by taking the error map from the corner cell removed decoder and dividing it by the error map of the full decoder. Corner areas (orange boxes) showed higher error ratios than the center area (blue box). (**h**) Quantitative comparisons of the decoding error ratios between the corner and the center areas (shown in g) for all sessions. Each line is a session (two-tailed Wilcoxon signed-rank test: triangle: p < 0.0001; square: p < 0.0001; hexagon: p = 0.0004; n = 18, 17, and 18 sessions for triangle, square, and hexagon, respectively, from 9 mice).

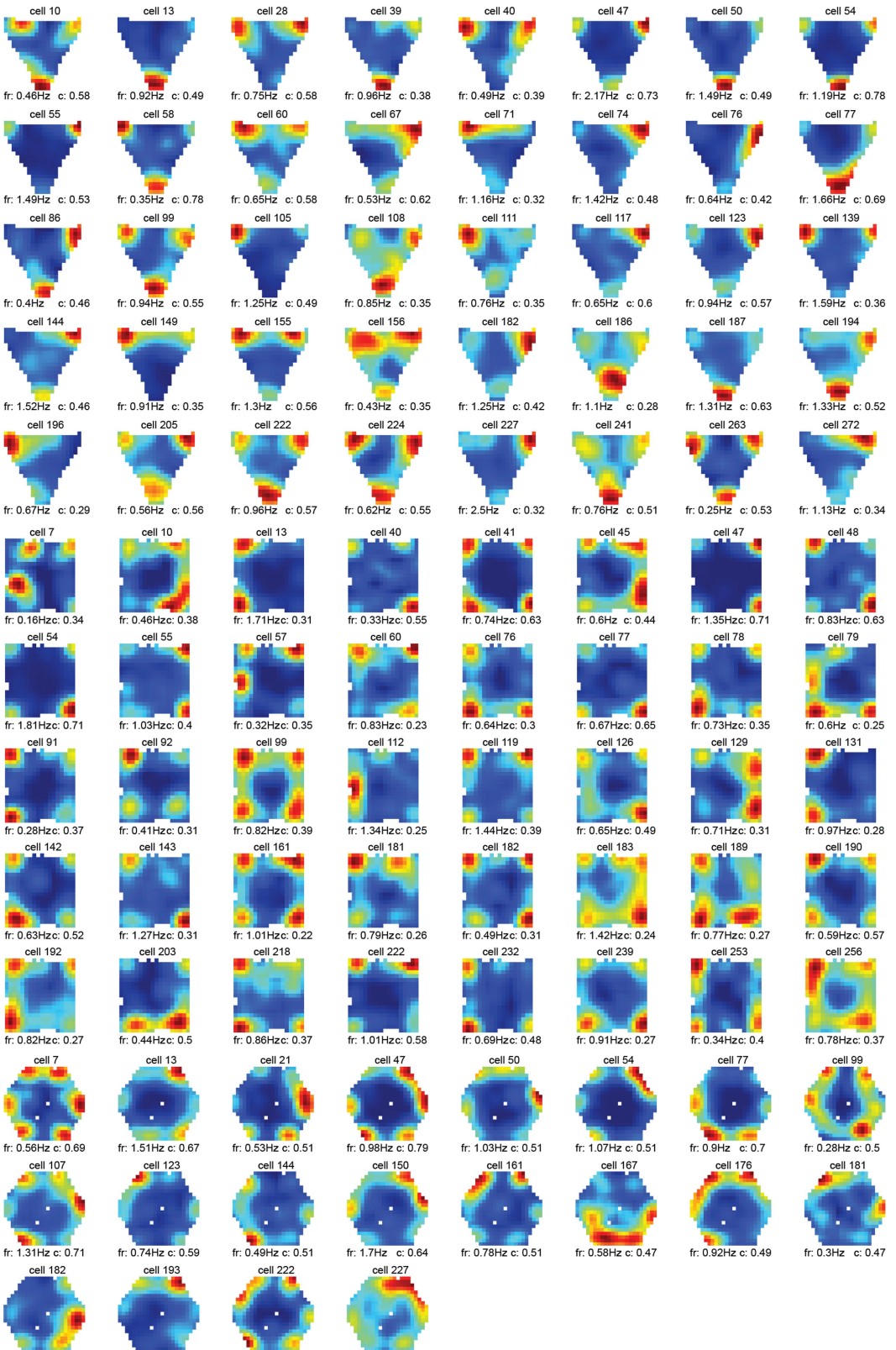

**Extended Data Fig. 3 | Rate map examples of identified corner cells.** Rate maps of identified corner cells in the triangle, square, and hexagon environments from a representative mouse. Corresponding peak spike rate (fr) and corner score (c) are labeled under each rate map.

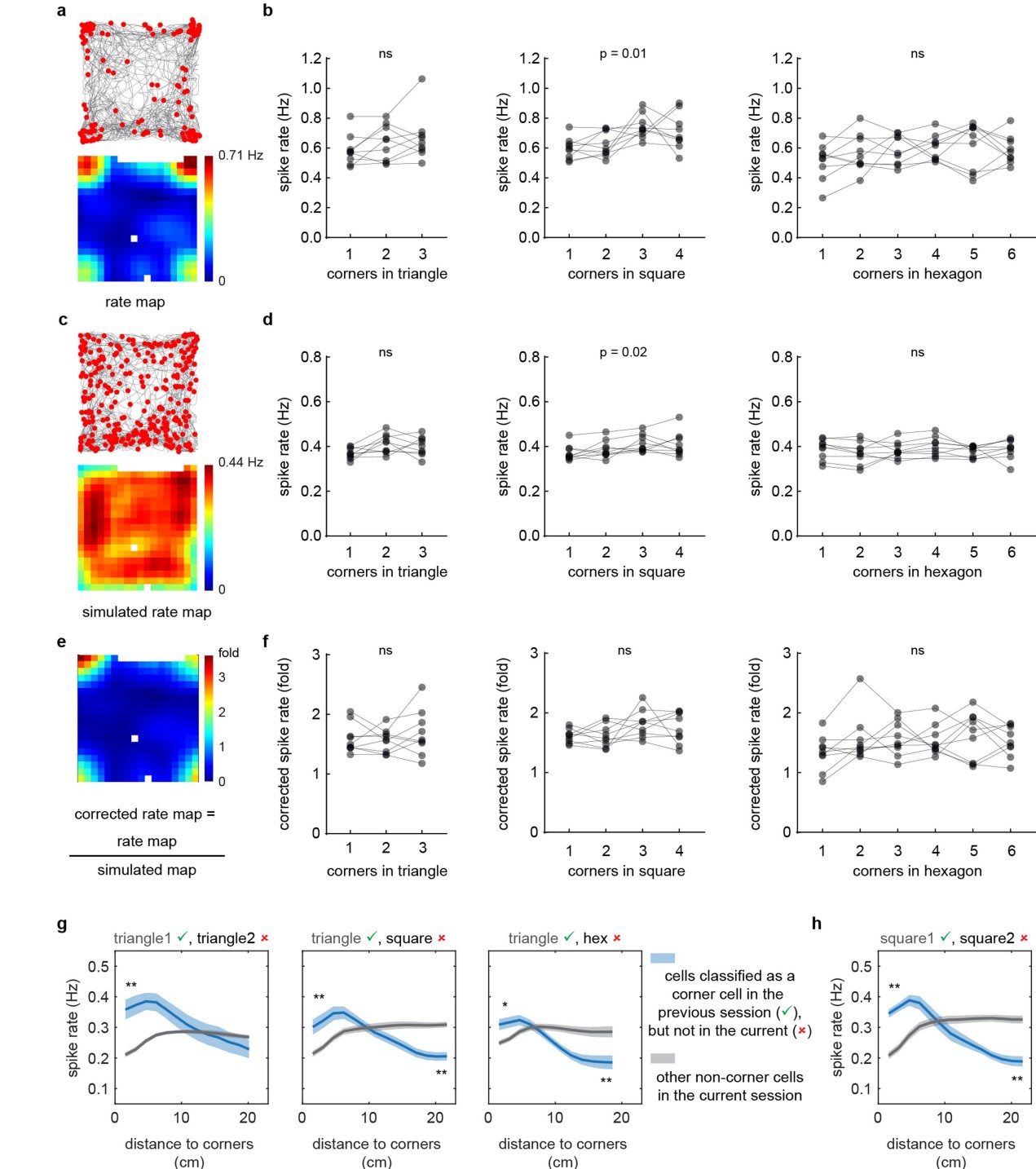

**Extended Data Fig. 4 |** See next page for caption.

**Extended Data Fig. 4 | Corrected rate map and persistent activity of corner cells.** (**a**) Raster plot and the corresponding rate map of an example corner cell. The raster plot (top) indicates extracted spikes (red dots) on top of the animal's running trajectory (grey lines) and the spatial rate map (bottom) is color-coded for maximum (red) and minimum (blue) values. (**b**) Spike rates of corner cells at each corner of the triangle, square, and hexagon, respectively, calculated using the rate maps of corner cells. Each line represents a mouse. There is a significant difference in the corner spike rates across different corners in the square (repeated measures ANOVA: $F_{(2.24, 12.92)} = 5.76$, $p = 0.010$; $n = 9$ mice). ns: not significant. (**c**) Raster plot and the corresponding rate map of a simulated cell. The simulated rate map was generated using a simulated neuron that fires along the animal's trajectory using the animal's own speed at the overall mean spike rate observed across all neurons of a given mouse (Methods). (**d**) Same as (b), but calculated using simulated rate maps for each mouse. The difference in corner spike rates in the square persists even in the simulated rate maps (repeated measures ANOVA: $F_{(2.56, 20.48)} = 4.24$, $p = 0.022$; $n = 9$ mice), indicating this effect is due to animals' behavior. (**e**) An example of corrected rate map, by dividing the original rate map (i.e., a) by the simulated rate map (i.e., c). Therefore, the spike rates on the corrected rate map were automatically converted to fold changes relative to the simulated rate map. This method was used to correct for any measurements that might have been associated with the animal's movement or occupancy, as purely behavior-related changes should be evident in both the original and simulated rate maps. (**f**) Same as (b), but calculated using the corrected rate maps of corner cells. Each line represents a mouse (repeated measures ANOVA: all $p > 0.05$; $n = 9$ mice). (**g-h**) Spike rates plotted relative to the distance to the nearest corner. Blue curves indicate neurons that were corner cells in the first session (green check) but not in the second session (red cross). The plots were generated based on the activity of neurons in the second session. Grey curves indicate other non-corner cells. Solid line: mean; Shaded area: SEM. Statistical tests were carried out as in Fig. 2k (two-tailed Wilcoxon signed-rank test: * $p < 0.05$, ** $p < 0.01$; $n = 9$ mice for each plot).

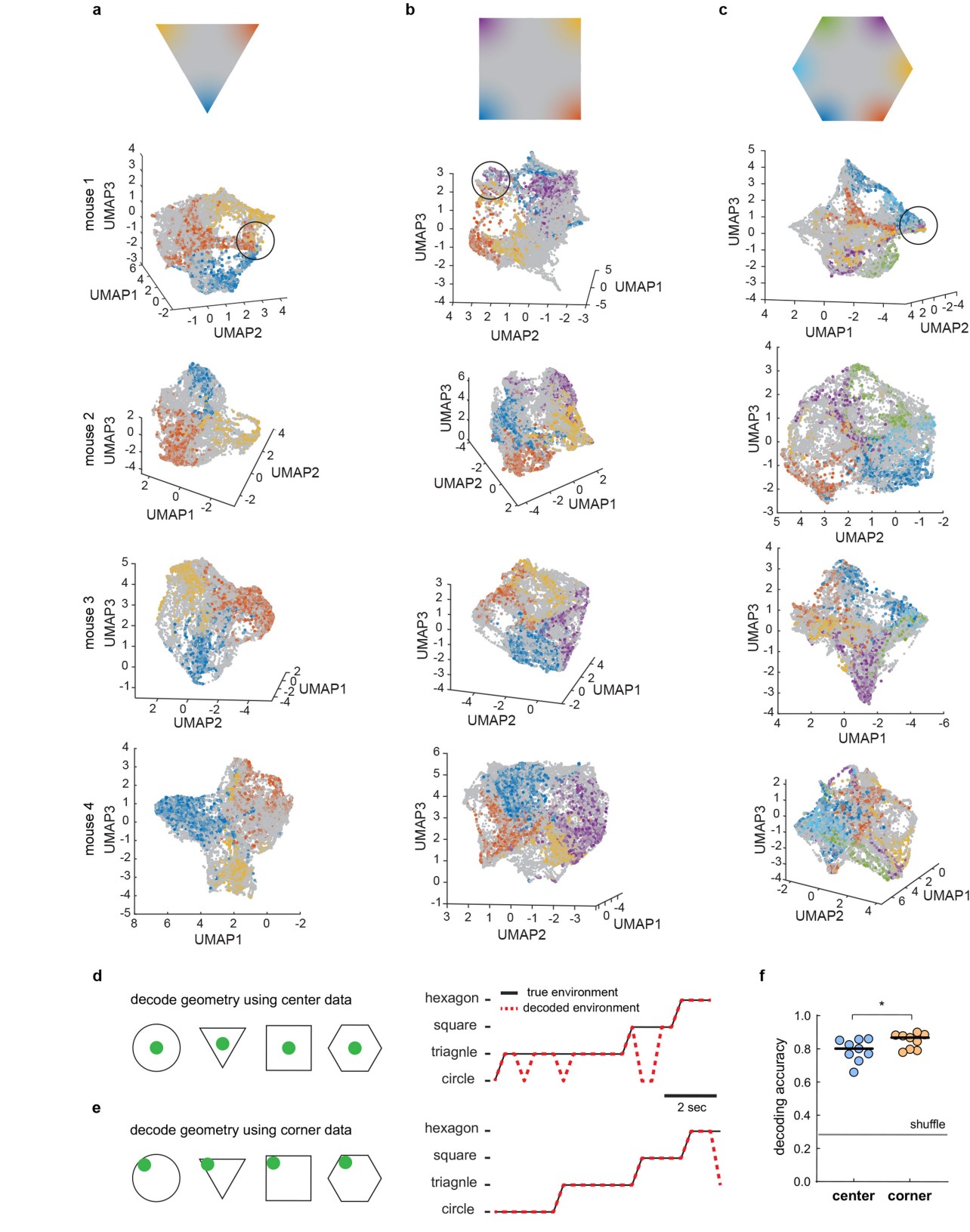

**Extended Data Fig. 5 | Neural manifold embedding and geometry decoding.**
(**a**) Three-dimensional (3D) embedding of the population activity of recorded subiculum neurons in the triangle from four different mice. We applied a sequential dimensionality reduction method using PCA and UMAP to obtain this neural manifold embedding (Methods). Each dot represents the population state at one time point. Time points within 5 cm of the corners are color-coded, with the color graded to grey as a function of the distance away from the corner (top row). The example from mouse 1 is the same as Fig. 1k but from a rotated view. The black circle denotes the place that corner representations converged on each manifold. (**b**) same as (a), but in the square environment. (**c**) same as (a), but in the hexagon environment. For a-c, analyses were replicated in 9 mice with similar results. (**d**) Left: Schematic of using the data from the environmental centre (8 cm diameter) to decode the environmental identity (geometry). Right: A decoding example to predict the geometry of the environment. (**e**) Left: Schematic of using the data near (within 8 cm) a geometric feature of the environment (e.g., a corner) to decode the environmental identity (geometry). Right: A decoding example to predict the geometry of the environment. (**f**) Comparison of decoding accuracy in (d) and (e) (mean ± SEM: center vs. corner: $0.79 \pm 0.02$ vs. $0.85 \pm 0.02$; two-tailed Wilcoxon signed-rank test: $p = 0.019$; $n = 9$ mice).

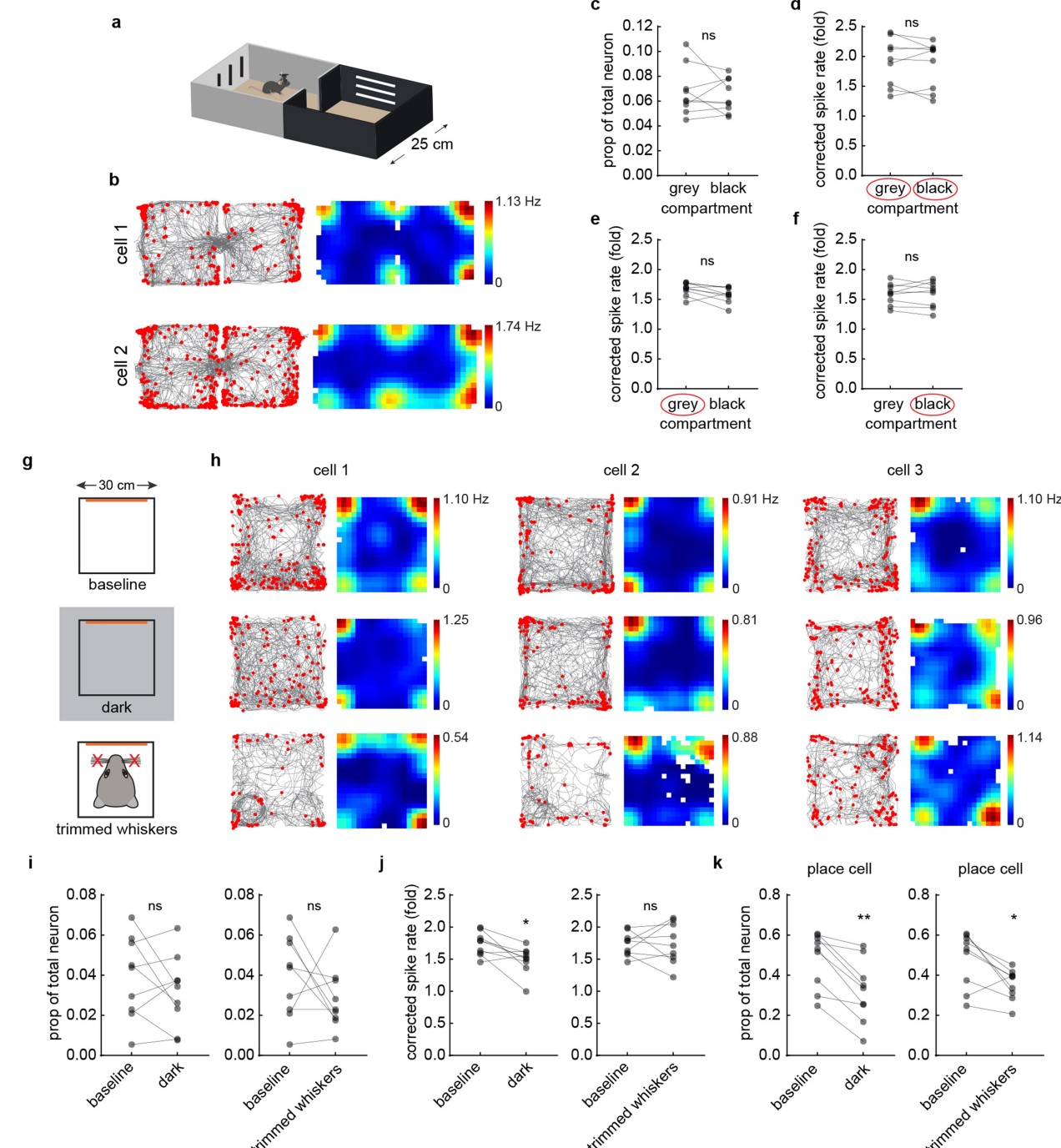

**Extended Data Fig. 6 | The response of corner cells to non-geometric and sensory manipulations.** (**a**) Schematic of a shuttle box composed of two compartments that differed in their visual and tactile cues. (**b**) Two example corner cells from two different mice recorded in the shuttle box shown in (a). Raster plot (left) indicates extracted spikes (red dots) on top of the animal's running trajectory (grey lines) and the spatial rate map (right) is color-coded for maximum (red) and minimum (blue) values. (**c**) Proportion of neurons classified as corner cells in the grey vs. black compartments of the shuttle box (two-tailed Wilcoxon signed-rank test: p = 0.91; n = 9 mice). ns: not significant. (**d**) Average corrected peak spike rates of corner cells at the corners in the grey vs. black compartments (two-tailed Wilcoxon signed-rank test: p = 0.43; n = 9 mice). Corner cells included in this quantification were defined as corner cells in both grey and black compartments. (**e**) Same as (d), but using corner cells that defined in the grey compartment (two-tailed Wilcoxon signed-rank test: p = 0.07). (**f**) Same as (d), but using corner cells that defined in the black compartment (two-tailed Wilcoxon signed-rank test: p = 0.91). (**g**) Schematic

of imaging in the dark or after trimming the whiskers. Orange bars indicate the location of local visual cues. (**h**) Raster plots and the corresponding rate maps of three corner cells from three different mice, as in (b). Each column is a neuron with activity tracked across all the conditions indicated on the left. (**i**) Left: Proportion of corner cells compared between baseline and dark (two-tailed Wilcoxon signed-rank test: p = 0.50; n = 9 mice). Right: Proportions of corner cells compared between baseline and whisker trimming (two-tailed Wilcoxon signed-rank test: p = 0.46, n = 9 mice). (**j**) Left: Comparison of the corrected peak spike rates of corner cells at square corners between baseline and dark (two-tailed Wilcoxon signed-rank test: p = 0.019; n = 9 mice). Right: Comparison of the corrected peak spike rates of corner cells at square corners between baseline and whisker trimming (two-tailed Wilcoxon signed-rank test: p > 0.99; n = 9 mice). (**k**) Same as (i), but for neurons classified as place cells in the subiculum (two-tailed Wilcoxon signed-rank test: left: p = 0.0039; right: p = 0.019; n = 9 sessions from 9 mice).

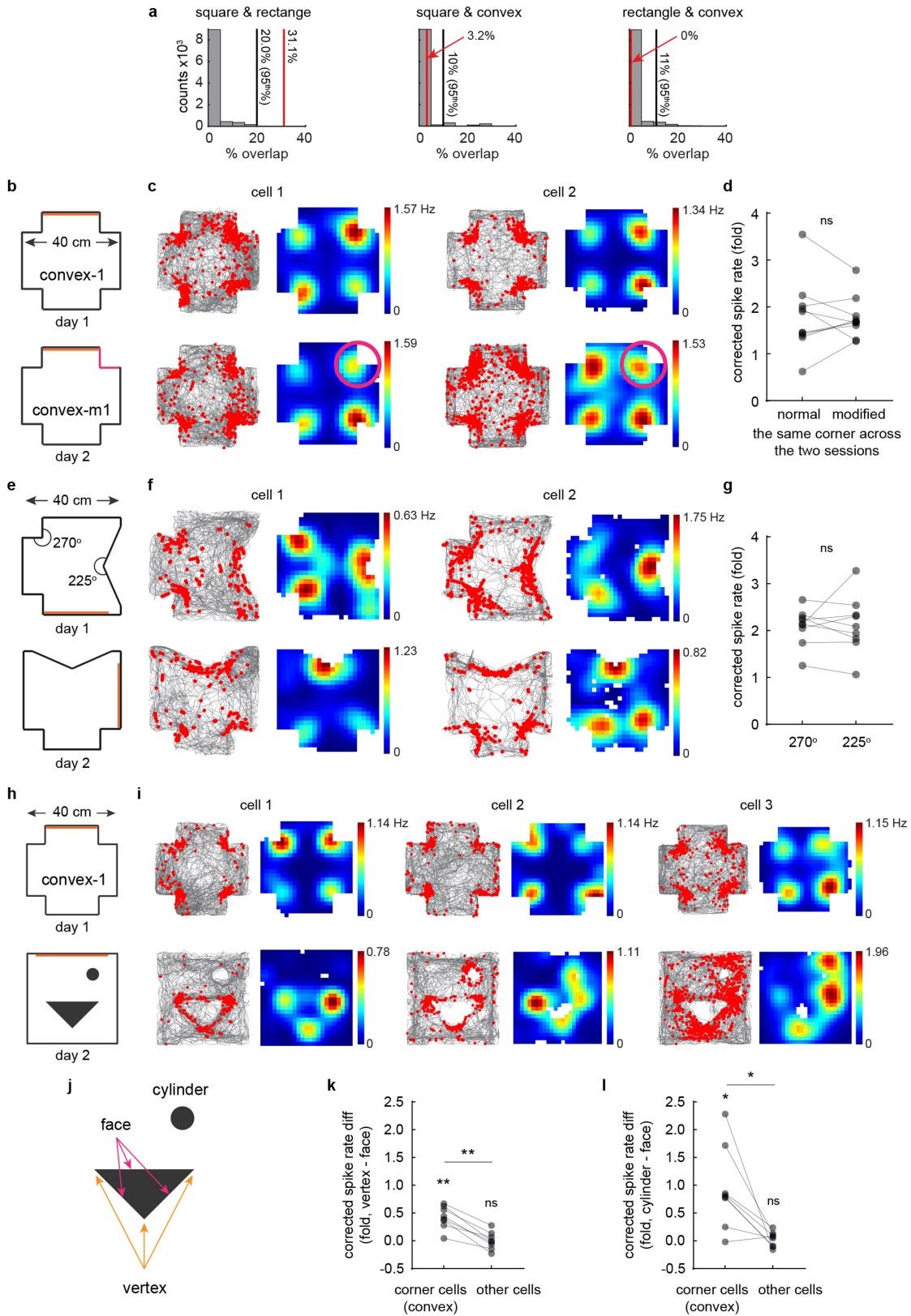

**Extended Data Fig. 7** | See next page for caption.

**Extended Data Fig. 7 | Convex corner cells are not sensitive to non-geometric changes or corner angles but respond to certain properties of discrete objects.** (**a**) Related to Fig. 4f: comparing the overlap of corner cells with the chance. Left: overlap of corner cells (both concave) between square and rectangle environments (red bar). Middle: overlap of corner cells (concave vs. convex) between square and convex-1 environments (red bar). Right: overlap of corner cells (concave vs. convex) between rectangle and convex-1 environments (red bar). The gray histogram illustrates the corresponding distribution of overlap expected by chance, with the black bar denotes the 95th percentile of each distribution. This distribution is generated by randomly selecting the same number of neurons, as indicated above for each environment, 1000 times in each mouse (n = 9 mice). Corner cells in the square and rectangle showed an overlap that is higher than chance (left), while the overlap between corner cells encoding concave or convex corners was minimal and below the chance level. (**b**) Schematic of the normal (convex-1) and the modified (convex-m1) convex environments. In convex-m1, one of the convex corners (in pink) was composed of walls of a different color and texture from the other three. Orange bars indicate the location of local visual cues. (**c**) Two representative corner cells encoding convex corners from two different mice. Each column is a neuron in which its activity was tracked across the two conditions indicated in (b). Raster plot (left) indicates extracted spikes (red dots) on top of the animal's running trajectory (grey lines) and the spatial rate map (right) is color-coded for maximum (red) and minimum (blue) values. Pink circles delineate the location of the modified corner in the convex-m1 arena. (**d**) Corrected peak spike rates of corner cells (convex) at the location of the modified corner in the convex-1 vs. convex-m1 arenas (two-tailed Wilcoxon signed-rank test: p = 0.85; n = 10 mice). Corner cells were defined in each session. (**e**) Schematic of a convex environment containing 270° and 225° corners, as in (b). The second session was also rotated 90 degrees counterclockwise, but was combined with the first session for analysis. (**f**) Raster plots and the corresponding rate maps of two corner cells encoding convex corners from two different mice, as in (c). Each column is a neuron in which its activity was tracked across the two sessions. (**g**) Corrected peak spike rates of corner cells (convex) at 270° and 225° corners (two-tailed Wilcoxon signed-rank test: p = 0.73; n = 9 mice). (**h**) Schematic of the experiments, as in (b). Corner cells (convex) were identified in the convex-1 environment on day 1, then their activity was tested with inserted objects (a triangle and a cylinder) on day 2. (**i**) Raster plots and the corresponding rate maps of three corner cells encoding convex corners from three different mice. Each column is a neuron in which its activity was tracked across the two sessions. (**j**) Illustration showing vertex and face locations for the triangular object. (**k**) Differences between spike rates at the vertices and faces of the triangular object in corner (convex) and non-corner cells (two-tailed Wilcoxon signed-rank test against zero: corner cells: p = 0.0078; non-corner cells: p = 0.95; two-tailed Wilcoxon signed-rank test: corner cells vs. non-corner cells: p = 0.0078; n = 8 mice). (**l**) Differences between spike rates at the cylinder and the faces of the triangular object in corner (convex) and non-corner cells (two-tailed Wilcoxon signed-rank test against zero: corner cells: p = 0.016; non-corner cells: p = 0.74; two-tailed Wilcoxon signed-rank test: corner cells vs. non-corner cells: p = 0.016; n = 8 mice).

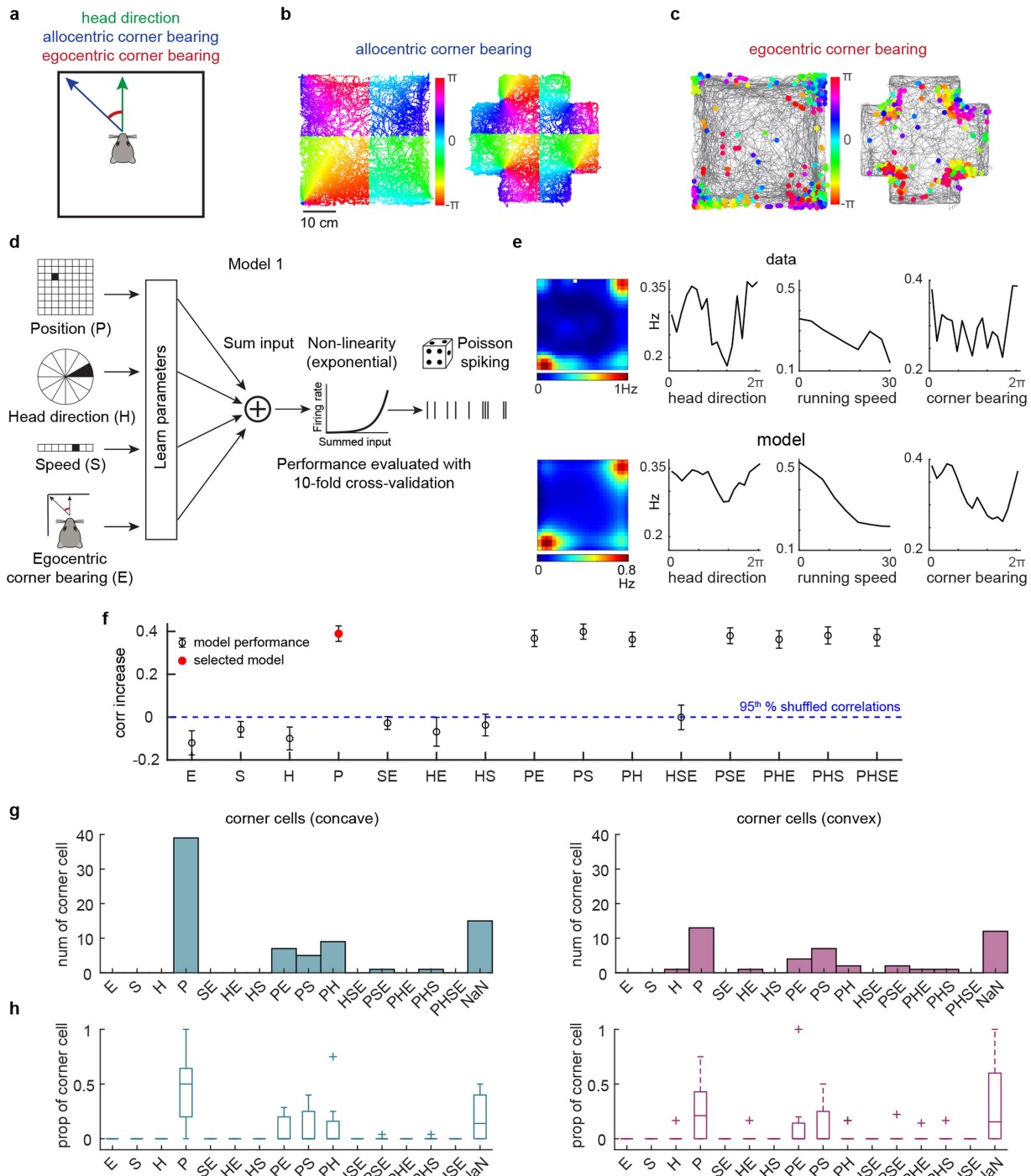

**Extended Data Fig. 8 | Corner cells primarily correspond to an allocentric reference frame.** (**a**) Schematic for calculating egocentric corner bearing (red) using head direction (green) and allocentric corner bearing (blue) (Methods). (**b**) Behavioral data of allocentric corner bearing in square and convex-1 environments from one representative mouse. Each position is color-coded for the allocentric bearing of the nearest corner relative to the animal. Note the discrete color shifts represent changes in the closest corner to the animal (e.g. the northwest versus southwest corner). (**c**) A corner cell example with spikes color-coded according to the egocentric corner bearing in square and convex-1 environments. 0 degrees indicates the animal is directly facing the nearest corner. (**d**) Schematic of the linear-non-linear Poisson (LN) model framework with behavioral variables including allocentric position (P), allocentric head direction (H), linear speed (S) and egocentric corner bearing (E) (Model 1,

see Methods). (**e**) True tuning curves (top) and model-derived response profiles (bottom) from an example corner cell. (**f**) An example of evaluating the model performance and selecting the best model using a forward search method. This example is from the corner cell in (e) and the best fit model (red dot) is the position (P) only model. (**g**) Number of corner cells (concave or convex) that were classified in each cell type category. This plot combined all the corner cells from all mice identified from the large square or convex-1 in Fig. 4 (a total of 77 corner cells (concave) and 44 corner cells (convex) from 10 mice). Some corner cells could not be classified potentially due to low spike rates (NaN). (**h**) Same as (g), but plotted using the proportion of total corner cells in each animal. For the box plots, the center indicates median, and the box indicates 25th and 75th percentiles. The whiskers extend to the most extreme data points without outliers (+).

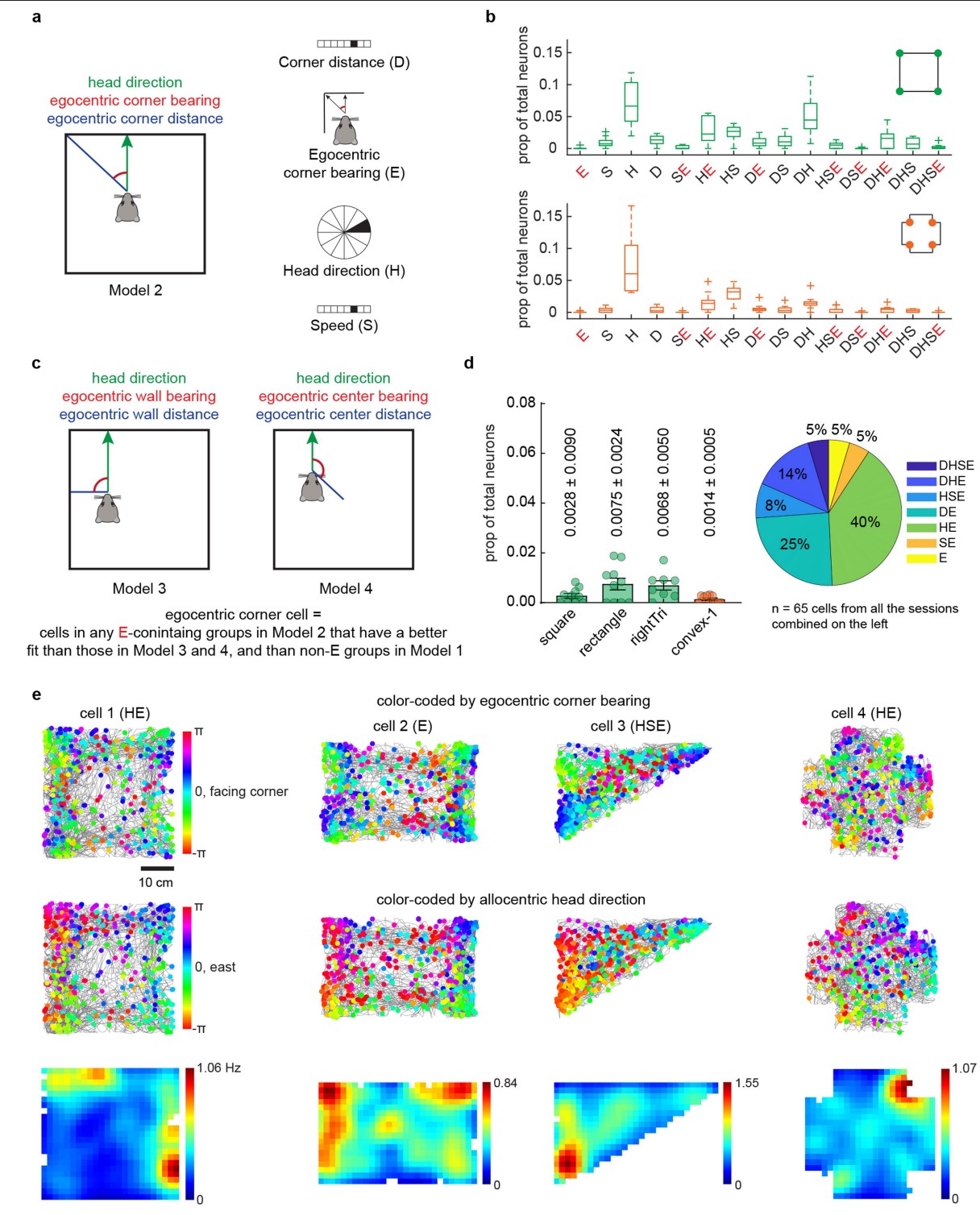

**Extended Data Fig. 9** | See next page for caption.

**Extended Data Fig. 9 | A small number of subiculum neurons encode egocentric bearing from corners.** (**a**) Schematic of the linear-non-linear Poisson (LN) model framework with behavioral variables including egocentric corner bearing (E), egocentric corner distance (D), allocentric head direction (H) and linear speed (S) (Model 2). (**b**) Proportion of subiculum neurons that were classified by Model 2 in large square (green) or convex-1 (orange) environments (n = 10 mice). Neurons combined from all model groups featuring egocentric corner-bearing (E, highlighted in red) account for 6.24 ± 1.20 % (n = 10 mice) of the total recorded subiculum neurons in the square environment. Similarly, 3.1 ± 0.6 % of neurons featuring egocentric corner bearing for convex corners in the convex-1 environment. For the box plots, the center indicates median, the box indicates 25th and 75th percentiles. The whiskers extend to the most extreme data points without outliers (+). (**c**) Schematics of LN model 3 and 4.

Model 3 contains egocentric wall bearing (E), egocentric wall distance (D), allocentric head direction (H) and linear speed (S). Model 4 contains egocentric center bearing (E), egocentric center distance (D), allocentric head direction (H) and linear speed (S) (see Methods). (**d**) Left: Proportion of egocentric corner cells in the subiculum from the square (mean ± SEM; n = 10 mice), rectangle (n = 10), right triangle (n = 8), and convex-1 (bearing to convex corners; n = 10) environments. Right: pie chart showing the conjunctive coding of egocentric corner cells with other behavioral variables. (**e**) Representative egocentric corner cells from the square, rectangle, right triangle, and convex-1 (bearing to convex corners) environments. Each column represents a neuron. The first row shows spike raster plots color-coded with egocentric corner bearing on top of the animal's running trajectory (grey lines). Similarly, the second row is color-coded with allocentric head direction. The third row shows positional rate maps.

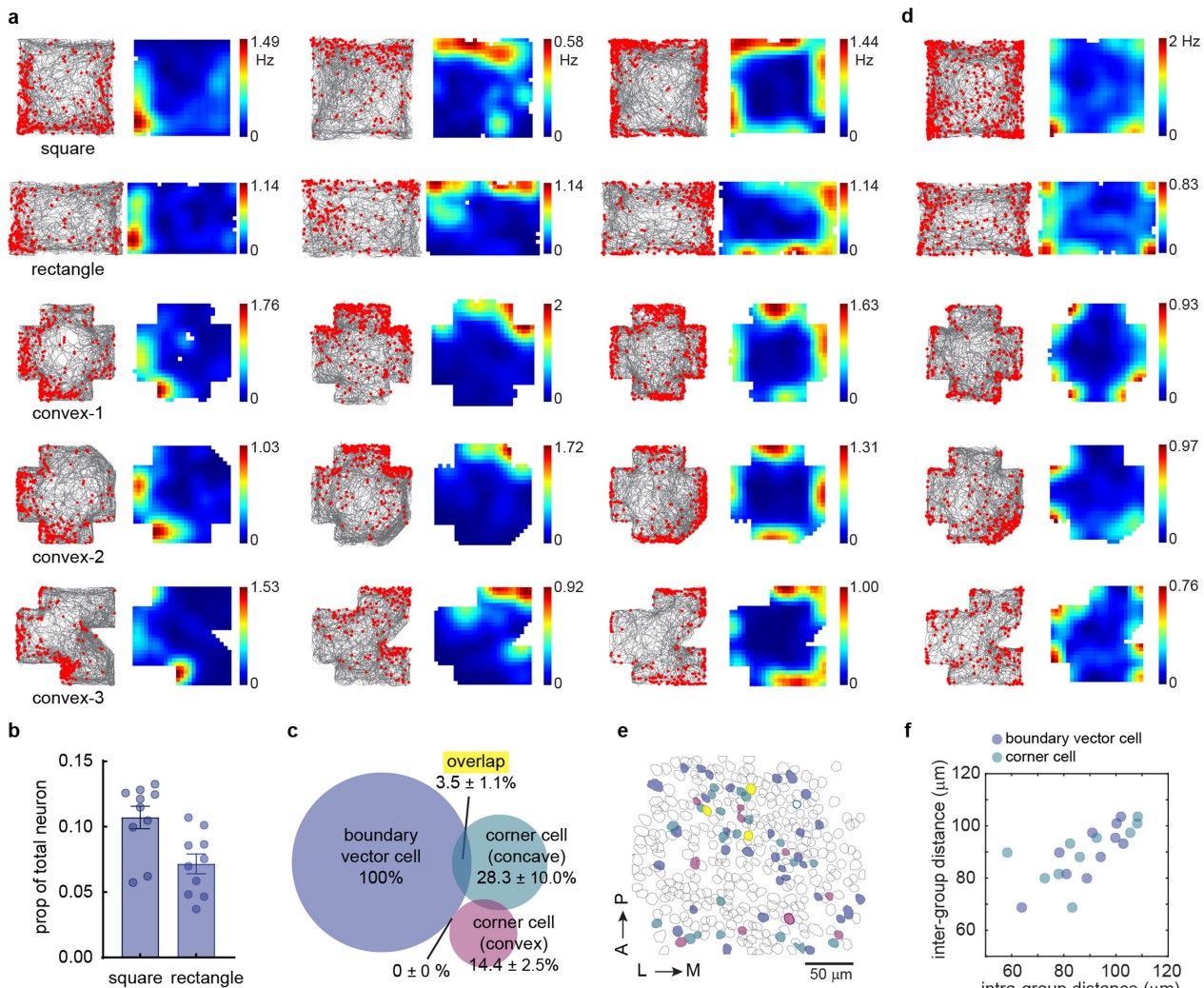

**Extended Data Fig. 10 | Corner coding differs from boundary vector coding in the subiculum.** (**a**) Raster plots and rate maps of three boundary vector cells (BVCs) from three different mice, plotted as in Fig. 1f. BVCs were identified in the square environment. Each column is a cell in which its activity was tracked across sessions. (**b**) Proportion of neurons classified as BVCs in the square and rectangle sessions. Each dot represents a session (n = 10 mice). Histogram and error bars indicate mean ± SEM. (**c**) Venn diagram showing the overlap between BVCs and corner cells (concave or convex). BVCs and corner cells encoding concave corners were identified in the square environment, while corner cells encoding convex corners were identified in the convex-1 arena. All numbers

were normalized to the number of BVCs. The overlap between corner cells and BVCs (3.5 ± 1.1%) was not higher than the threshold above the random overlap level (12.5%) (**d**) An example neuron classified as both a BVC and corner cell based on its activity in the square environment. (**e**) Anatomical locations of BVCs and corner cells (concave or convex) from a representative mouse. Color codes are the same as in (c). Unfilled grey circles represent other subicular neurons. A: anterior; P: posterior; L: lateral; M: medial. (**f**) Pairwise intra- vs. inter-group anatomical distances for BVCs and corner cells (concave + convex) (repeated measures ANOVA: F(1.37, 12.36) = 0.30, p = 0.66; n = 10 mice).

# Reporting Summary

## Statistics

For all statistical analyses, confirm that the following items are present in the figure legend, table legend, main text, or Methods section.

| n/a | Confirmed | |
|---|---|---|
| ☐ | ☒ | The exact sample size (*n*) for each experimental group/condition, given as a discrete number and unit of measurement |
| ☐ | ☒ | A statement on whether measurements were taken from distinct samples or whether the same sample was measured repeatedly |
| ☐ | ☒ | The statistical test(s) used AND whether they are one- or two-sided <br> *Only common tests should be described solely by name; describe more complex techniques in the Methods section.* |
| ☐ | ☒ | A description of all covariates tested |
| ☐ | ☒ | A description of any assumptions or corrections, such as tests of normality and adjustment for multiple comparisons |
| ☐ | ☒ | A full description of the statistical parameters including central tendency (e.g. means) or other basic estimates (e.g. regression coefficient) AND variation (e.g. standard deviation) or associated estimates of uncertainty (e.g. confidence intervals) |
| ☐ | ☒ | For null hypothesis testing, the test statistic (e.g. *F*, *t*, *r*) with confidence intervals, effect sizes, degrees of freedom and *P* value noted <br> *Give P values as exact values whenever suitable.* |
| ☐ | ☒ | For Bayesian analysis, information on the choice of priors and Markov chain Monte Carlo settings |
| ☒ | ☐ | For hierarchical and complex designs, identification of the appropriate level for tests and full reporting of outcomes |
| ☐ | ☒ | Estimates of effect sizes (e.g. Cohen's *d*, Pearson's *r*), indicating how they were calculated |

*Our web collection on statistics for biologists contains articles on many of the points above.*

## Software and code

Policy information about availability of computer code

| Data collection | Calcium Imaging data were acquired using custom-developed and validated software (Compiled DAQ Software OLD version 1.0.0, https://github.com/daharoni/Miniscope_DAQ_Software) |
|---|---|
| Data analysis | MATLAB packages that were used for the manuscript include NoRMCorre v1.0.0 (https://github.com/flatironinstitute/NoRMCorre) for image motion correction, image-registration v1.0.0 (github.com/fordanic/image-registration) for image alignment across days, CNMF-E v1.1.2 (https://github.com/zhoupc/CNMF_E) for Ca++ signal extraction, and OASIS v1.0.0 (https://github.com/zhoupc/OASIS_matlab) for Calcium signal deconvolution. MATLAB 2020a and and GraphPad Prism 9 were used to perform statistical analyses. Calcium imaging data were further preprocessed and analyzed using MATLAB based packages and scripts (https://github.com/yanjuns/Sun_et_al_2024_Nature). |

For manuscripts utilizing custom algorithms or software that are central to the research but not yet described in published literature, software must be made available to editors and reviewers. We strongly encourage code deposition in a community repository (e.g. GitHub). See the Nature Portfolio guidelines for submitting code & software for further information.

## Data

Policy information about availability of data

All manuscripts must include a data availability statement. This statement should provide the following information, where applicable:
- Accession codes, unique identifiers, or web links for publicly available datasets
- A description of any restrictions on data availability
- For clinical datasets or third party data, please ensure that the statement adheres to our policy

> Calcium imaging data generated in this study are available on Mendeley Data: https://doi.org/10.17632/5sj8d5vtg2.1
> Calcium imaging data were further preprocessed and analyzed using MATLAB based packages and scripts: https://github.com/yanjuns/Sun_et_al_2024_Nature

## Research involving human participants, their data, or biological material

Policy information about studies with human participants or human data. See also policy information about sex, gender (identity/presentation), and sexual orientation and race, ethnicity and racism.

| Reporting on sex and gender | NA |
|---|---|
| Reporting on race, ethnicity, or other socially relevant groupings | NA |
| Population characteristics | NA |
| Recruitment | NA |
| Ethics oversight | NA |

Note that full information on the approval of the study protocol must also be provided in the manuscript.

# Field-specific reporting

Please select the one below that is the best fit for your research. If you are not sure, read the appropriate sections before making your selection.

☒ Life sciences          ☐ Behavioural & social sciences          ☐ Ecological, evolutionary & environmental sciences

For a reference copy of the document with all sections, see nature.com/documents/nr-reporting-summary-flat.pdf

# Life sciences study design

All studies must disclose on these points even when the disclosure is negative.

| Sample size | Sample sizes were based upon on convention in the field (Sun et al., 2022; Mallory et al., 2021; Low et al., 2021; Campbell et al., 2021; Alexander et al., 2020; Hardcastle et al., 2015). The number of mice used in each set of experiments ranged from 7 to 10. |
|---|---|
| Data exclusions | After a certain period post-surgery, the imaging quality began to decline in some animals, and this thus led to slight variations in the number of mice used in each set of experiments, ranging from 7 to 10. We evaluated the imaging quality for each mouse before executing each set of experiments. No mice were excluded from the analyses as long as the experiments were executed. For experiments with two identical sessions for a given condition (e.g., Fig. 1 and 2), sessions with less than 3 identified corner cells were excluded to minimize measurement noise in spike rates. This criterion only resulted in the exclusion of one session from one mouse in Fig. 2e. |
| Replication | Up to six mice were used as a cohort for each batch of experiments. All experiments were repeated with at least two different cohorts of mice and the results are reproducible. |
| Randomization | Randomization is not applicable to this study, as all animals in a given experiment were subjected to the same behavioral and imaging protocols. |
| Blinding | Blinding is not applicable to this study, as there were no separated groups in the experiments. |

# Reporting for specific materials, systems and methods

We require information from authors about some types of materials, experimental systems and methods used in many studies. Here, indicate whether each material, system or method listed is relevant to your study. If you are not sure if a list item applies to your research, read the appropriate section before selecting a response.

## Materials & experimental systems

| n/a | Involved in the study |
|-----|----------------------|
| ☒ ☐ | Antibodies |
| ☒ ☐ | Eukaryotic cell lines |
| ☒ ☐ | Palaeontology and archaeology |
| ☐ ☒ | Animals and other organisms |
| ☒ ☐ | Clinical data |
| ☒ ☐ | Dual use research of concern |
| ☒ ☐ | Plants |

## Methods

| n/a | Involved in the study |
|-----|----------------------|
| ☒ ☐ | ChIP-seq |
| ☒ ☐ | Flow cytometry |
| ☒ ☐ | MRI-based neuroimaging |

## Animals and other research organisms

Policy information about studies involving animals; ARRIVE guidelines recommended for reporting animal research, and Sex and Gender in Research

| | |
|---|---|
| Laboratory animals | For subiculum imaging, 8 Camk2a-Cre; Ai163 mice (Ref 43, 4 male and 4 female), 1 Camk2-Cre mouse (female, JAX: 005359), and 1 C57BL/6 mouse (male) were used. For CA1 imaging, 12 Ai94; Camk2a-tTA; Camk2a-Cre (JAX id: 024115 and 005359) mice (7 male and 5 female) were used. Mice were group housed with same-sex littermates until the time of surgery. At the time of surgery, mice were 8 -12 weeks old. After surgery mice were singly housed at 21 -22°C and 29 - 41% humidity. Mice were kept on a 12-hour light/dark cycle and had ad libitum access to food and water in their home cages at all times. |
| Wild animals | No wild animals were used in this study |
| Reporting on sex | Data from both males and females were combined for analysis, as we did not observe sex differences in, for example, corner cell proportions, spike rates to different corners angles, or concavity and convexity. |
| Field-collected samples | No field collected samples were used in this study. |
| Ethics oversight | All procedures were approved by the Institutional Animal Care and Use Committee at Stanford University School of Medicine and the University of California, Irvine. |

Note that full information on the approval of the study protocol must also be provided in the manuscript.

## Plants

| | |
|---|---|
| Seed stocks | NA |
| Novel plant genotypes | NA |
| Authentication | NA |

