## [Peer Review File · Nature]

Manuscript Title: Subicular neurons encode concave and convex geometries

Reviewer Comments & Author Rebuttals

Reviewer Reports on the Initial Version:

Referees' comments:

Referee #1 (Remarks to the Author):

It was a pleasure to read the manuscript recently submitted for publication at Nature, "The subiculum encodes environmental geometry" by Sun, Y., Nitz, D.A., Xu, X., and Giacomo, L.M. Through a series of experiments, they combine classical geometric manipulations of a foragable open field environment with cutting-edge neuroimaging techniques in freely-behaving mice, the authors show evidence for a novel spatially-tuned neuronal response in the dorsal subiculum: 'corner cells'. While prior work has demonstrated the presence of boundary-vector coding in subicular neurons, the authors state that, "whether and how the brain encodes the concavity and convexity of environmental features (e.g., corners and protrusions) remains largely unexplored." The authors then demonstrate the existence of subicular neurons that represent corners and their specific compositional features, namely: corner angle, wall height, and degree of intersection. They also observed cells that also encode convexity at corners, walls and objects. Further, using a generalized linear-non-linear model (LM), they demonstrate that corner cells in the subiculum are allocentrically-based. These findings suggest that the subiculum contains neural codes to comprehensively represent the geometric layout of environments in an allocentric reference frame. These findings mark an interesting discovery within the context of cognitive mapping, navigation, and memory, and contribute to our understanding of the systems-level organization of space. After closely reading the manuscript, I have several general comments and minor concerns described below.

General thoughts and comments:

I think the authors have done a nice and convincing analysis to support results on 'corner cells' reported throughout the manuscript (I have a few minor thoughts/concerns/additional analyses that I think could be addressed by the authors). However, after reading this manuscript carefully several times, my main and recurring thought is that I am not sure these results represent a major discovery. I have dissected the paper for insights this manuscript provides, as stated by the authors, and have found the following:

In the Introduction the authors frame the issue: "whether and how the brain encodes the concavity and convexity of environmental features (e.g., corners and protrusions) remains largely unexplored." In the Discussion, "...the subiculum contains two distinct populations of neurons that encode concave and convex environmental corners."

"Unlike concave corner cells, whose activity tracked both the angles of the corners and curvatures of the oval, the activity of convex corner cells only tracked the curvatures of the cylinders and was not sensitive to the angles of convex corners."

"This raises the possibility that the animals process environmental convexity differently from

concavity”

In my opinion, these insights fall short of representing a breakthrough discovery worthy of publication at Nature. Various areas within and outside the hippocampus, such as the subiculum and entorhinal cortices are known to also encode geometric spatial properties, thereby challenging the somewhat misleading implication of the paper's title. Both the hippocampal place cells and entorhinal grid cells are known to encode spatial features and demonstrate a variety of changes - rate remapping, global remapping, and receptive field size alteration - when subjected to manipulations in the geometry of their environment. Furthermore, the subiculum's boundary vector cells and egocentric boundary vector cells - located in the striatum, retrosplenial cortex, postrhinal cortex, and lateral entorhinal cortex - also play a role in encoding geometric information.

The paper introduces the identification of a novel cellular response that encodes corners, a specific and important feature of geometry. This finding, although innovative, is not particularly unexpected given the established understanding of the various cell types and their functions in geometric encoding. Corners, being a distinctive feature of environmental geometry, are naturally likely candidates for encoding by the hippocampus, entorhinal cortex, and subiculum. This probability is dependent on the design of the experiment, further demonstrating the influence experimental conditions have on cellular responses.

Despite this specific discovery, the authors did not reference or establish any existing conceptual framework or model of spatial coding or cognition that predicts or stands to gain from the knowledge that a small percentage of neurons are specifically dedicated to encoding corners. The paper's title, 'The subiculum encodes environmental geometry,' is factually correct but does not necessitate the existence of 'corner cells' to substantiate this claim. In reality, a myriad of regions such as the striatum, retrosplenial cortex, postrhinal cortex, pre- and para- subiculum, medial and lateral entorhinal cortex, all hippocampal subregions, and the subiculum itself all play integral roles in encoding environmental geometry.

Therefore, the overall impact and significance of the discovery of 'corner cells' remain unclear. This ambiguity is present when reviewing both the introductory and discussion sections of the manuscript.

- In the discussion, the authors suggest that corner representation could be produced from the threshold sum of inputs from nearby BVCs (lines 288-9). This seems to be a somewhat odd suggestion, since the boundary-vector to place model makes the same claim and applies to place fields located in any location (Barry et al., 2006). It is unclear how place cells and corner cells would differ if both could construct spatial representation in an allocentric reference frame from BVC inputs. Thus, there seems to be an implicit assumption in this suggestion that place cells do not construct place fields from BVCs, but corner cells do. The authors could perhaps expand their discussion of an alternative hypothesis, that CA1 place cells provide input to subicular corner cells. Recent observations have revealed that CA1 population codes reflect distance to objects in large environments, and distance to walls (Nagelhus et al., 2023). It seems perhaps more plausible that subicular coding of geometry could be constructed from a threshold sum of place cell inputs.

- The penalty for each cell's corner score when more fields than corners have been observed seems arbitrary, particularly when considering the detection method for fields themselves. If a tertiary field were detected in a single corner after thresholding (for a cell that has fields in each corner), and that tertiary field was very close to the nearby field exactly in the corner, this would heavily penalize that cell's corner score. It seems plausible this could happen with the thresholding method used to detect fields from the rate maps. The authors should re-run the analyses for detection of corner cells with an altered metric, either in field detection (e.g. spatial coherence or distance between "fields" after thresholding, or varying threshold level: 0.3 seems a bit arbitrary as well). The authors should also demonstrate how reliable corner-associated activity is within cells. I would recommend using a split-half measure to further discern which cells have reliable corner firing throughout a recorded session. This would likely remove cells with non-selective firing (that might produce many fields). It also appears they did not use this penalization in the final calculation of corner scores (line 522), and so why use the penalization to begin with?

- How were the criteria for BVCs decided? They identified BVCs as those cells with significant spatial information and whose field covered more than 70% of the nearest wall. Given the current lack of observed BVCs in mice, it is important that the authors justify their choice of criteria (which might not be best applied with the same criteria from rat studies).

- I have a major concern with the use of the GLM method to determine whether corner cells are allocentric or egocentric, which is a large claim of the paper. If rate maps are used to define the corner score of cells, and only those cells that are identified with significant corner scores are used for the GLM analysis, I am worried there is an inherent bias to detect allocentric tuning in corner cells. The corner cell metric based on rate maps has strong allocentric priors baked in – the spatial binning of neurons is in an allocentric reference frame. Perhaps some subicular cells have egocentric tuning, but would not pass the corner cell criterion as-is, and therefore could bias the analysis to suggest all corner-selective activity in the subiculum is allocentric. This seems to be double-dipping. I suggest the authors use the GLM method itself as an alternative means to detect corner cells, and compare results across metrics. Alternatively, the authors could devise a corner-cell detection method based on an egocentric reference frame, and demonstrate whether there are any cells that pass such a criterion.

- A general claim of the paper is that the subiculum represents the geometry of familiar environments. I feel the authors could do more with their large-scale population recordings to this effect. Namely, manifold embedding techniques can extract neural representations from such large-scale population recordings. The authors could present evidence that the subiculum in fact constructs geometric representation, using such procedures given the power that large population recordings afford with miniscope calcium imaging. Further, if the subiculum contains geometric information, then it should be possible to decode the geometry of environments across sessions. The authors should show if this is the case in animals that have been recorded across many geometries, across rooms.

- While the authors used relatively rigorous methods to detect corner cells, it is still unclear to me how many cells would be identified as corner cells by chance based on the combined shuffling and corner distance measures. A shuffling-only criterion with significance considered at $p < 0.05$ might

yield 5% of cells as corner cells, but it's unclear with the additional corner distance measure how this would affect chance levels of corner cell detection. I suggest the authors add a control analysis wherein they probe randomized (simulated) corner locations in each environment, and detect how many cells would be classified as corner cells if simulated "corners" were randomly distributed throughout the environment. It seems they attempted a similar procedure with simulated place cells in random locations, but I suggest they attempt the randomization of "corner locations" using their actual data.

- The sub-heading on line 160 is misleading. From my reading of the paper and respective figures (Extended data Fig 3), they did not observe any effect of darkness + whisker trimming on corner cell coding in the subiculum, but only on subicular place cells. Thus, they cannot claim that corner cells depend on visual and/or tactile senses given they have not demonstrated that combined darkness and whisker trimming impair corner coding, and I do not believe they have strong grounds to speculate on this (line 178).

- The authors suggest that concave and convex corner coding exist in two separate subpopulations in the subiculum. However, I do not see what this adds to their conclusion. If subicular neurons encode corners, irrespective of concavity/convexity, and their tunings parameters are highly specific, then you would expect these cells to be non-overlapping, and distributed with a salt-and-pepper topography. In the spirit of not over-complicating a growing list of functional cell types in the parahippocampal system, I suggest they scrap this added distinction, since it is not necessary to view these as functionally heterogeneous cell types. I do not see there would be a theoretical value in treating these concave and convex corner cells as distinct populations.

Minor comments:

- In illustrations of the environment (e.g., Fig 1.) the authors should show where the visual cues were in the apparatus throughout all geometric manipulations relative to the recorded shape (described in methods, line 356-7).

- The authors should add one sentence stating why the additional polygon masking procedure was implemented for convex corner cell detection (line 537).

- In the description of the anatomical clustering method (line 564), they hypothesize that if functionally-defined neuronal groups were anatomically clustered, then inter-group distance should be greater than intra-group distance. I suggest the authors make this clearer in the main text and accompanying figure, as this will help readers to interpret the lack of clustering shown.

- The authors should specify that their deconvolution procedure was used to infer spikes, but not to detect spikes per se. A small change in language would suffice. For example, on line 60, they could instead state: "Calcium signals were extracted with CNMF and OASIS deconvolution, and subsequently binarized to estimate spikes for all cells."

- On line 66, the authors state many cells exhibited place-cell like firing patterns that did not change significantly across environments. Given the coarseness of this spatial firing, and lack of remapping across environments, I suggest they rephrase this description to suggest that the firing across

environments was “spatially modulated but not geometry-specific”.

- The authors should state approximate atlas coordinates where they imaged in the subiculum based on the approximate, combined lens implant and expression location (line 74).

Referee #2 (Remarks to the Author):

Environmental geometry is an important, sometime overriding, feature that animals use to orient and navigate in an environment. The present study investigates the coding of convex and concave corners of an environment by recording the calcium activity of large populations of neurons in the subiculum of freely moving mice. The authors make a thorough case that a subset of subiculum neurons encode these corners, in addition to the known place and boundary-related firing already described in the subiculum. This is an important study and represents the most complete characterization of corner-coding properties in the larger hippocampal formation. The results have strong implications for how this system encodes spatial environments, adding to the list of cell types in this system such as place cells, grid cells, head direction cells, border cells, vector cells, and others that collectively allow the system to recognize different environments, localize themselves within an environment, and guide the animal through an environment as it navigates the terrain.

The paper is written clearly and the data are presented clearly and comprehensively. There are a number of concerns with the present version of the manuscript, however, mostly related to improper statistical procedures and questions about the robustness of some of the effects. These concerns must be addressed before the paper can be published.

Major

1) The first problem is that the authors define a corner cell based on it exceeding the 95th percentile of a shuffled distribution, which is a standard procedure. However, in many cases, the percentage of cells that are classified as corner cells are at or below the predicted percentage based on chance. That is, using this shuffling test, one expects that ~ 5% of cells will pass the test by chance, and it is incumbent on the authors to demonstrate that the percentage of cells actually passing the criterion exceed the chance level by a statistically significant amount. In a number of cases, the percentage of neurons that pass their test is barely above, or below, 5% (lines 85, 113-114, 131, 204; Extended Data instances not included here). Although many examples in the paper are clearly corner cells (especially in the square and triangle environments), it is not clear whether these are representative or not. If they are representative, rather than best examples (this should be stated either way), then it appears that the shuffling procedure is not doing what the authors intend. (The hexagon corner cells are less convincing, as are the asymmetric environments.) It is possible that the percentages of corner cells are much less than those that pass the shuffling test because of the second criterion of field distance (line 82-83); if so this should be explained. The authors need to resolve this statistical problem to provide any confidence in the validity of the percentages of corner cells and devise a test to show that the percentage of corner cells in the various experiments exceeds that expected by chance. I recognize that this is difficult when the true prevalence of a cell type in the population is near or below the standard alpha level of 0.05, but it is a problem that needs to be addressed in some fashion.

2) Similarly, the percentage of corner cells in the circle is much less than 5%, which would indicate that these fictive corner cells are less likely than chance to be identified in the circle; one would have expected that the number should be $\sim 5\%$. Were the arbitrary points at the edges of the circle or could they be anywhere? It would seem that they should be at the edges to match the real corner conditions.

3) In some analyses, the authors analyze at most 2 sessions/mouse and use the session as the unit of analysis, which is a welcome improvement over many studies that combine data over many sessions that record unknown numbers of repetitions of the same cells, and use the same repeated cells as the unit of analysis. However, this is still a problem, given that in the two sessions the authors are probably recording from a very large percentage of the same cells twice, making the two recording sessions not independent samples and violating the statistical tests. The authors should either take an average of sessions for each mouse and use that as the measure, or choose one session per mouse based on unbiased criteria. It is unlikely to make any difference in the results or interpretation, but is the proper statistical methodology. For example, ED Figure 1K should replace Fig 1g, and similar throughout manuscript where necessary.

4) The statistical methodology used to compare the firing of corner cells near and far from the corners to the corresponding firing of non-corner cells is fundamentally flawed. A core assumption of the statistical tests is that the data are sampled randomly from the two populations being compared. In this case, the authors are identifying cells from a single sample that have a certain characteristic (corner-related firing) and then comparing that biased subsample to the remainder of the sample. It is similar to selecting a sample of tall people from a population and then performing a statistical test on whether these tall people are taller than the rest of the population. Any p values that come from such a test are statistically meaningless because of this violation of the assumption of random sampling. Thus, Figure 1h and all further results based on the same procedure (Fig 2k, 4d, ED Fig 2g,h) are invalid statistical tests and need to be removed from the manuscript (hopefully replaced by another analysis that is statistically valid).

5) Figure 1j suffers from the same problem, as the authors are preselecting the cells that have stable corner firing across sessions and then testing whether these cells have high correlation across sessions (if they are preselected accurately, then of course they will). There is no purpose in doing a statistical test on this biased sample.

6) Lines 99-100: 2.5% of neurons were consistently classified as corner cells across all non-circle environments. What is the chance level? Does 2.5% statistically significantly exceed this chance level?

7) Lines 130-131, 138-139: It is not appropriate to do parametric statistics comparing percentages across groups, as percentage data are not normally distributed. An arcsin transform is required, or the use of nonparametric tests.

8) In a number of places, statements are made that two groups (e.g., corner cells vs noncorner cells) have different responses to a particular manipulation. However, it is necessary to statistically

compare these groups directly to each other to support this claim. It is not sufficient to merely show a significant effect for one group and a nonsignificant effect for the other. See Nieuwenhuis et al. (2011) doi:10.1038/nn.2886. Examples: Fig 5f, 5g, Extended Data Fig 2h, ED Fig 4j vs. 4k, ED Fig 4l vs. 4m. In each case, 2-way ANOVAs need to be run to test for main effects of the 2 factors and interactions.

9) Lines 167-169. The analysis of ED Fig 3c,d does not address the question posed here. The statement on the previous sentence is that color and texture did not affect the activity of corner cells. ED Fig 3c,d show that the proportions of Corner cells and the peak rates of corner cells in each session on average do not change based on nonspatial changes. They do not show that individual cells are not affected (e.g, by firing rate changes, some corner cells losing their corner tuning while other cells gain corner tuning, etc.).

10) Lines 172-173: Similar concern as Item 7

11) Lines 206-207: Given how rare the corner cells are, it is not surprising that the combination of these properties is even smaller. However, it may still be the case that corner and convex tuning appears in the same population of cells across conditions more than expected by chance. The authors need to perform some calculation and test to determine if these numbers truly reflect different populations or whether they just appear that way due to the low prevalence of these properties. Line 222, 276-77 may be overstatements if this result is not confirmed statistically. At present, I am not convinced the authors have demonstrated this.

12) On a similar note, were any cells identified as conjunctive corner and place cells? Is it possible that a cell can have both properties, but criterion 2 of defining a corner cell excludes such cells?

13) It is not clear what is accomplished by the corrections of spike rates procedure described in lines 551-559, using the mean firing rate across all cells. What if a particular cell has a behavioral confound that is not reflected in the population mean activity? I don't understand how this analysis works and whether it does what the authors intend. How is this any different from doing an occupancy normalized rate map? Can the authors show a real or hypothetical example of a cell whose rate map is altered appreciably by this procedure?

Minor

14) Line 45: The subiculum receives input from both the medial and lateral entorhinal cortex, and thus the word "medial" should be deleted.

15) Words like "Surprisingly" and "Interestingly" at the start of sentences should be used sparingly, and preferably deleted entirely. Let the data speak for themselves and let the reader decide whether they are interesting or surprising.

16) Line 302-307: Alternatively, the egocentric corner cells of LEC may combine to produce the allocentric corner cells of subiculum (similar to the model of Gofman et al. (2019) from the Derdikman lab of egocentric border cells in postsubiculum combining to form allocentric border cells in MEC and parasubiculum). In general, it is not clear that the subiculum is unique in having cells that

can encode geometry (boundaries, corners, curvatures), as similar properties have been seen in CA1, LEC, and MEC as well. This does not negate the importance of this paper in terms of its thorough analysis of corner tuning in the system, but the evidence is not strong that these geometric coding properties are specially associated with the subiculum in the hippocampal formation processing loops.

17) Line 304 does not entirely accurately represent the corner-associated activity of LEC shown by Wang et al. (2023). 8/12 of the corner cells in that paper appeared to be egocentric in terms of CW vs CCW direction selectivity and about 7/12 were speed-tuned. Thus, not all corner cells in LEC were egocentric and speed-modulated.

18) Although the authors recorded males and females and combined data (which is fine if there are no differences between groups), the authors should include a statement that they investigated whether there were sex differences and combined the data if there were no significant differences.

19) Line 431: Were whiskers completely trimmed? Please provide some details for replication purposes.

20) Line 494: Please describe how a place field was defined and how its limits were determined.

21) If possible, please provide some indication of where the recordings were from along the transverse axis and along the longitudinal axis of the subiculum. Also, can the authors estimate how deep into the layer the recordings were taken from? Were these all deep-layer cells?

22) Line 1062: How is a 2D surface linearized?

23) ED Fig 5b is unclear. Are these real data, or a schematic? It is not clear how to read this figure.

Author Rebuttals to Initial Comments:

Referees' comments:

Referee #1 (Remarks to the Author):

It was a pleasure to read the manuscript recently submitted for publication at Nature, "The subiculum encodes environmental geometry" by Sun, Y., Nitz, D.A., Xu, X., and Giocomo, L.M. Through a series of experiments, they combine classical geometric manipulations of a foragable open field environment with cutting-edge neuroimaging techniques in freely-behaving mice, the authors show evidence for a novel spatially-tuned neuronal response in the dorsal subiculum: 'corner cells'. While prior work has demonstrated the presence of boundary-vector coding in subicular neurons, the authors state that, "whether and how the brain encodes the concavity and convexity of environmental features (e.g., corners and protrusions) remains largely unexplored." The authors then demonstrate the existence of subicular neurons that represent corners and their specific compositional features, namely: corner angle, wall height, and degree of intersection. They also observed cells that also encode convexity at corners, walls and objects. Further, using a generalized linear-non-linear model (LM), they demonstrate that corner cells in the subiculum are allocentrically-based. These findings suggest that the subiculum contains neural codes to comprehensively represent the geometric layout of environments in an allocentric reference frame. These findings mark an interesting discovery within the context of cognitive mapping, navigation, and memory, and contribute to our understanding of the systems-level organization of space. After closely reading the manuscript, I have several general comments and minor concerns described below.

We thank the reviewer for their constructive feedback. Below, we describe the new analyses, figures and text that we believe addresses the concerns.

General thoughts and comments: I think the authors have done a nice and convincing analysis to support results on 'corner cells' reported throughout the manuscript (I have a few minor thoughts/concerns/additional analyses that I think could be addressed by the authors). However, after reading this manuscript carefully several times, my main and recurring thought is that I am not sure these results represent a major discovery. I have dissected the paper for insights this manuscript provides, as stated by the authors, and have found the following: In the Introduction the authors frame the issue: "whether and how the brain encodes the concavity and convexity of environmental features (e.g., corners and protrusions) remains largely unexplored." In the Discussion, "...the subiculum contains two distinct populations of neurons that encode concave and convex environmental corners." "Unlike concave corner cells, whose activity tracked both the angles of the corners and curvatures of the oval, the activity of convex corner cells only tracked the curvatures of the cylinders and was not sensitive to the angles of convex corners." "This raises the possibility that the animals process environmental convexity differently from concavity".

1. We appreciate the reviewer's careful consideration of the structure of our manuscript. In the revised version, we have re-framed a significant portion of our writing to better reflect our key findings and have also expanded the scope of the paper. Specifically, our work highlights that the subiculum explicitly encodes a fundamental geometric property that defines an environmental shape: concavity and convexity. Importantly, single-cell activity tracks the curvatures of this geometric feature and corner cells, unlike boundary vector cells, are active across all corners of an environment. Consequently, this explicit encoding of concave and

convex curvatures, in combination with the encoding of straight environmental boundaries, allows the brain to adapt to encoding and reconstructing any geometric shapes within an environment. This achievement is impossible to attain using only allocentric and egocentric boundary vector cells. Moreover, this encoding of curvature is fundamentally different from the coding of geometry as a whole in place cells and grid cells (see our response 2 for more detail on this point), and we have not come across similar reports of this curvature encoding in other brain regions. Thus, the subiculum appears to be a unique brain structure for encoding this geometric property and has the potential to effectively reconstruct the detailed geometric shape of an environment. Therefore, we revised our writing for both the introduction and discussion in numerous locations. Below, for ease of reference, we highlight some key passages from the revised manuscript:

“However, it is still unclear which geometric properties are encoded in the brain at the single-cell level, except for prior reports on egocentric (self-centered) and allocentric (world-centered) boundary coding in the hippocampal formation and associated brain regions (refs 32-37). Unlike traditional laboratory conditions, which typically have straight walls, natural environments are full of concave and convex shapes, from networks of tree branches to winding burrow tunnels. Given that the combination of straight lines and curves can give rise to any shapes, we hypothesize that the brain encodes the concave and convex curvatures in an environment (e.g., corners and curved protrusions) in addition to straight boundaries.”

“Unlike the activity of corner cells that encoded concave corners, which tracked both the angles of the corners and curvatures of the oval, the activity of corner cells that encoded convex corners only tracked the curvatures of the cylinders and was not sensitive to the convex angle. One possibility is that this difference reflected the animal’s perception of convex corners. As there was a sharp bend - or kink - at the vertex of the convex corner may be perceived by as a more salient feature than the corner itself. This idea aligns with the encoding of convex curvatures, as the pronounced curvature at kinks may be distinct when viewed from different angles. This phenomenon may also parallel the perceptual differences in concave vs. convex scenes and shapes in the visual and somatosensory systems (refs 55-58).”

In my opinion, these insights fall short of representing a breakthrough discovery worthy of publication at Nature. Various areas within and outside the hippocampus, such as the subiculum and entorhinal cortices are known to also encode geometric spatial properties, thereby challenging the somewhat misleading implication of the paper's title. Both the hippocampal place cells and entorhinal grid cells are known to encode spatial features and demonstrate a variety of changes - rate remapping, global remapping, and receptive field size alteration - when subjected to manipulations in the geometry of their environment. Furthermore, the subiculum's boundary vector cells and egocentric boundary vector cells - located in the striatum, retrosplenial cortex, postrhinal cortex, and lateral entorhinal cortex - also play a role in encoding geometric information.

2. The reviewer raises an important set of points. First, in terms of encoding spatial properties of an environment, we agree there is a foundational set of literature that demonstrates that hippocampal place cells and grid cells show various changes in response to manipulations of an environment’s geometry. However, we view a distinction between cells that respond to manipulations of an environment’s boundaries or geometry, and cells that explicitly encode geometric properties of an environment. For example (as the reviewer notes), entorhinal grid cells transiently change the distance between their firing fields when a familiar square box is stretched or compressed, (Barry et al., 2007; Munn et al., 2020; Stensola et al., 2012) and the hexagonal structure of their firing pattern distorts in polarized environmental geometries (Krupic et al., 2015). These grid pattern changes represent alterations to either a familiar geometry or to the geometric symmetry of the environment but the grid pattern itself is not

encoding geometric properties or specific elements that define the geometry. Likewise, changes in place cell firing rates, field locations or field size are indicative of an alteration to environmental geometry (Leutgeb et al., 2005; Muller and Kubie, 1987; O'Keefe and Burgess, 1996; Wills et al., 2005) but provide very little information about the specific elements that compose the geometry. On the other hand, the corner coding we observe in the subiculum represents a geometric feature universally across environmental shapes. In addition, the firing properties of corner cells tracked the explicit features of corners, including corner angle, height, and the degree to which the walls were connected. In this sense, the coding we observe in the subiculum (e.g. 'corner' cells), is quite distinct from the coding property changes observed in grid or place cells across environmental geometries.

Second, we do agree completely that boundary vector cells and egocentric boundary vector cells do play a role in encoding geometric information. Coding by boundary vector cells does differ in an important way from the convex/concave coding we observe however, as BVC activity is typically restricted to one or two walls of the environment, whereas convex/concave coding encodes this feature more universally across environments.

Third, in new analyses in the revised version of the manuscript, we show that the geometry of an environment can be decoded from the population level activity of all subiculum neurons. Importantly, we observed that the decoding accuracy improved when the animal was close to geometric features of the environment, such as corners. This result suggests that the neural code in the subiculum near the geometric features of the environment carries more information about the overall environmental geometry (Extended Data Fig. 5d-f). This idea is further supported by new analyses (suggested by the reviewer) in which we consider the population dynamics of all subiculum neurons (Fig. 1k and Extended Data Fig. 5c) and show that corners are coded for both specifically (there is information about which corner the mouse is in) and universally (there is overlap in this information across corners, suggested a general code for this geometry property of environments). Detailed texts on these new analyses are in response 11.

The paper introduces the identification of a novel cellular response that encodes corners, a specific and important feature of geometry. This finding, although innovative, is not particularly unexpected given the established understanding of the various cell types and their functions in geometric encoding. Corners, being a distinctive feature of environmental geometry, are naturally likely candidates for encoding by the hippocampus, entorhinal cortex, and subiculum. This probability is dependent on the design of the experiment, further demonstrating the influence experimental conditions have on cellular responses.

3. We agree with the reviewer that experimental conditions are an important factor in identifying functional coding properties. Even so, allocentric corner coding appears to potentially be a somewhat unique feature of the subiculum. To consider this explicitly, we examined a population of 5212 neurons in the hippocampal CA1 from 12 mice, imaged using miniscopes during exploration of a square box. Given the proportions of corner coding we observe in the subiculum, we would expect to see corner coding in ~350 hippocampal neurons in this dataset. Contrary to this, we only observed 34 neurons that passed the corner cell score threshold (0.62 ± 0.14 % of the population). Examples of the small number of CA1 corner cells we observed are now shown in Extended Data Fig. 1m.

Despite this specific discovery, the authors did not reference or establish any existing conceptual framework or model of spatial coding or cognition that predicts or stands to gain from the knowledge that a small percentage of neurons are specifically dedicated to encoding corners.

4. The reviewer makes an important set of points. First, in terms of the observation that 'a small percentage of neurons are specifically dedicated to encoding corners', we would note that the percentages of cells classified as a given functional cell type will depend on the 'score' generated to define that population. Here, we aimed to use a relatively conservative score such that we identified cells that very strongly encode convex/concave geometric features. We felt this would aid in the characterization of this previously undescribed functional 'cell type'. Moreover, the percentage of cells we identify as corner coding neurons is not dissimilar from the percentage of cells identified as border cells (less than 10% of neurons) (Solstad et al., 2008) or grid cells (in mice, 8-10%) (Mallory et al., 2018) in medial entorhinal cortex; functionally defined cell classes that have been foundational in understanding how the brain encodes an internal map of external space.

Second, as detailed in responses 1-3, we have made significant changes to the title, abstract, introduction and discussion of the manuscript, in addition to new analyses (as detailed in response 11). As noted above, the coding by corner cells for all features in the environment that exhibit concavity or convexity is fundamentally different from coding by grid, place, border/boundary vector cells that either respond to changes in environmental geometry (place and grid cells, point 2 in the response) or to one or few features of geometry (border and boundary vector cells, point 2 in the response). We believe this new text and analyses have provided a richer conceptual framework in which to consider the unique significance of the information encoded by the subiculum neurons we describe.

Third, there are no existing computational models (that we are aware of) that explicitly predict the existence of corner cells. Perhaps the closest are models that generate boundary vector cells and describe their potential relationship to the emergence of place cell coding. However, both BVCs and place cells differ from the coding described in the present manuscript – in that they either code for boundaries *and* direction in which case even a short range BVC would be active in only one or few corners (BVCs), or they code for a specific corner (place cells). We would hope the observation of neurons devoted to encoding corners as a geometric feature will inspire future computational work aimed at understanding how corner coding might be generated, how it could contribute to coding features observed in other navigationally relevant regions or how it might play a role in specific ethologically relevant behaviors.

The paper's title, 'The subiculum encodes environmental geometry,' is factually correct but does not necessitate the existence of 'corner cells' to substantiate this claim. In reality, a myriad of regions such as the striatum, retrosplenial cortex, postrhinal cortex, pre- and para- subiculum, medial and lateral entorhinal cortex, all hippocampal subregions, and the subiculum itself all play integral roles in encoding environmental geometry. Therefore, the overall impact and significance of the discovery of 'corner cells' remain unclear. This ambiguity is present when reviewing both the introductory and discussion sections of the manuscript.

5. As noted above, we have made significant changes to the introduction and discussion sections of the manuscript. See our points in response 2 and 3. We have also changed the title to: 'Subicular neurons encode concave and convex geometries'.

In the discussion, the authors suggest that corner representation could be produced from the thresholded sum of inputs from nearby BVCs (lines 288-9). This seems to be a somewhat odd suggestion, since the boundary-vector to place model makes the same claim and applies to place fields located in any location (Barry et al., 2006). It is unclear how place cells and corner cells would differ if both could construct spatial representation in an allocentric reference frame from BVC inputs. Thus, there seems to be an implicit assumption in this suggestion that place cells do not construct place fields from BVCs, but corner cells do. The authors could perhaps expand their discussion of an alternative hypothesis, that CA1 place cells provide input to subicular

corner cells. Recent observations have revealed that CA1 population codes reflect distance to objects in large environments, and distance to walls (Nagelhus et al., 2023). It seems perhaps more plausible that subicular coding of geometry could be constructed from a thresholded sum of place cell inputs.

6. The reviewer raises an interesting suggestion. We apologize as we did not intend to indicate that place cells do not construct their fields from BVCs. We have amended the discussion to clarify that there are several potential inputs that could play a role in the generation of corner coding, which includes hippocampal place cells and BVCs, and include relevant references to this point. Given the anatomical evidence of projections from the hippocampus to the subiculum, we also agree with the reviewers that a highly plausible mechanism for subicular coding of geometry could be the thresholding of place cell inputs. Thus, we revised our discussion as follows:

“Given the anatomical observations of dense CA1 to subiculum connectivity (Sun et al., 2019; Witter, 2006; Witter and Amaral, 2004) and recent observations that CA1 population codes can indicate the distance to objects and walls (Nagelhus et al., 2023), one possibility is that corner cell firing patterns may arise from the convergent inputs of CA1 place cells. Namely, a thresholded sum of the activity of place cells near the environmental corners. This idea aligns with the previously observed clustering of place fields near environmental corners in individual CA1 place cells and temporally synchronized CA1 ensembles (Muir and Bilkey, 2001; Sun and Giocomo, 2022), and could explain the sensitivity of corner cell firing rates to corner angles, as hippocampal place fields may show more overlap in smaller corner regions. ”

The penalty for each cell’s corner score when more fields than corners have been observed seems arbitrary, particularly when considering the detection method for fields themselves. If a tertiary field were detected in a single corner after thresholding (for a cell that has fields in each corner), and that tertiary field was very close to the nearby field exactly in the corner, this would heavily penalize that cell’s corner score. It seems plausible this could happen with the thresholding method used to detect fields from the rate maps. The authors should re-run the analyses for detection of corner cells with an altered metric, either in field detection (e.g. spatial coherence or distance between “fields” after thresholding, or varying threshold level: 0.3 seems a bit arbitrary as well). The authors should also demonstrate how reliable corner-associated activity is within cells. I would recommend using a split-half measure to further discern which cells have reliable corner firing throughout a recorded session. This would likely remove cells with non-selective firing (that might produce many fields).

7. First, the reviewer is correct that this scenario (i.e. a tertiary field being detected close to the nearby field exactly in the corner) would penalize the corner cell score more heavily than the scenario where the tertiary (referred to as 'extra' in the following text) field is slightly further away from the corner. We agree that the original method for penalization was not ideally suited to address this concern, as the penalty values for a given extra field followed a U-shaped curve (1 to 0 to 1) from the corners to the center of the environment. In the revised manuscript, we have updated the method. Instead of using the absolute value of the corner score for each extra field, we now use the absolute value of the corner score for each extra field minus one (see Extended Data Fig. 1g, copied below; Methods) to penalize the corner score for the cell. This way, the penalty for a given extra field ranges from 0 to 2, with 0 for the field at the corner and 2 for the field at the center. As a result, as the extra field moves away from a corner, the penalty for the overall corner score gradually increases. Nevertheless, among all the corner cells identified in the triangle, square, and hexagon environments in the revised manuscript,

only $7.8\% \pm 0.5\%$ (mean \pm SEM; $n = 9$ mice) of them were classified under the condition that involved penalization.

We also agree with the reviewer that we did not provide adequate justification for the threshold we used for field detection in the previous version of the manuscript. In that version, the threshold of 0.3 was based on the threshold used to classify border cells in the medial entorhinal cortex (Solstad et al., 2008). Thus, we re-ran our analyses for detecting corner cells using a threshold that varied from 0.1 to 0.6 in the triangle, square, hexagon, and circle environments. We found that the proportion of corner cells remained relatively consistent across these different thresholds in each environment (Fig. R1a). However, starting at a threshold of 0.5, we noted the number of fictive corner cells in the **circle** environment doubled

Fig. R1. (a) Proportion of corner cells identified using different field detection thresholds from 0.1 to 0.6 in the triangle, square, hexagon and circle arenas. Solid line: mean; Shaded area: SEM. ($n = 9$ mice). **(b)** Positional spike rates of corner cells identified with different thresholds. The spike rates were plotted relative to the distance to the nearest corner for the triangle, square and hexagon arenas. **(c)** The difference between the maximum and minimum of each curve in (b). A higher score indicates a larger activity difference between the corner and center of an environment, which inferring a better quality of the identified corner cells. Red arrow points to the best threshold for each arena compared to the threshold range of 0.1 – 0.4 (Fig. R1a).

Next, to determine the best threshold for classifying corner cells, we plotted the spike rate of the identified corner cells at each threshold as a function of the distance to the nearest corner (Fig. R1b). We then calculated the difference between the maximum and minimum values for each plotted curve (Fig. R1c). A larger difference in this metric indicates a higher activity

difference in the corner cell spike rate between the corner and the center of an environment. We observed that the maximum difference was typically at a threshold value of 0.3 or 0.4, indicating that a reasonable threshold for corner cell field detection is $\sim 0.3 - 0.4$. Thus, in the revised manuscript, we used 0.3 as the threshold for identifying corner cells in smaller environments such as the circle, triangle, square, and hexagon, and 0.4 as the threshold for identifying corner cells in larger environments such as the 40-cm square, rectangle, and convex environments (e.g., Fig. 4). We have also added text to the methods section describing our rationale for selecting these values. Note this figure (Fig. R1) is only presented in the response to reviewer comments but if the reviewer(s) or editor feels including this figure would improve the manuscript, we would be happy to include it in a revised version of the manuscript.

Extended Data Fig. 1f-k. (f) Left, the definition of the corner score for a given spatial field ($\text{cornerscore}_{\text{field}}$). $d1$: distance from the center of the arena to the field; $d2$: distance from the field to the nearest corner; dc : the mean distance from the corners to the center of the arena. Right, the distribution of $\text{cornerscore}_{\text{field}}$ in a square environment for a given field. Namely, this represents the corner score you would expect if a neuron was active in a given pixel of this plot. Note that the cornerscore can range from -1 (blue) to 1 (green). (g) The definition of the corner score for a given cell ($\text{cornerscore}_{\text{cell}}$, see Methods). (h) An example corner cell with corner score values for each field labeled in red, the final corner score for this cell is shown below. (i) Shuffling of $\text{cornerscore}_{\text{cell}}$ to determine a threshold for classifying a neuron as a corner cell. This example is from the same cell as in (h). (j) To be classified as a corner cell the distance between any two fields (major fields, if the number of fields $>$ number of corners) needed to be greater than half of the dc value, as indicated by the blue line in (f). (k) As an additional criterion, to be classified as a corner cell, within-session stability needed to be greater than 0.3 (Pearson's correlation between the two halves of the data). The distribution shows the within-session stability of all corner cells from the triangle, square, and hexagon sessions

We also implemented two additional criteria for classifying corner cells. First, as suggested by the reviewer, we now consider within-session stability. Extended Data Fig. 1k shows the within-session stability distribution of all the corner cells that met the criteria from Extended Data Fig. 1i - j in the triangle, square, and hexagon sessions before applying this stability criterion ($n = 1018$ cells from 9 mice). In the revised manuscript, corner cells are now required to have within-session (two halves) stability higher than 0.3, as determined by the 95th % of the random within-session stability distribution using shuffled spikes. Second, we implemented a criterion regarding the distance between fields when defining corner cells (see Extended Data Fig. 1j, copied above). Specifically, the distance between any two fields (major fields, if the number of fields $>$ number of corners) needed to be greater than half of the distance from

the corners to the center of the arena (denoted as 'dc' in Extended Data Fig. 1f). This criterion was applied to avoid classifying neurons with multiple fields clustered around a single or a subset of corners as corner cells. The field distance criterion filtered out one-third (33.5% \pm 3.9%, n = 9 mice) of the corner cells that passed the shuffling threshold.

It also appears they did not use this penalization in the final calculation of corner scores (line 522), and so why use the penalization to begin with?

Extended Data Fig. 2a-d. (a) Distributions of corner scores calculated without applying the penalty (Methods and Extended Data Fig. 1g) for all subiculum neurons recorded in the triangle (n = 4774 cells, 18 sessions, 9 mice), square (n = 4685 cells, 17 sessions, 9 mice), and hexagon (n = 4774 cells, 18 sessions, 9 mice) environments. The red lines represent the 95th percentile of the distributions in (b). Note: This calculation is for display purposes only and was not used for the analyses in this paper, also see Methods for details. (b) Distributions of corner scores calculated with penalty applied, which was used in the analyses. The red lines represent the 95th percentile of the corresponding distributions. (c) Distributions of shuffled corner scores calculated without applying the penalty, which was used in the analyses. Each cell was shuffled 1000 times. The red lines represent the 95th percentile of the distributions in (b). The red box over (b) and (c) indicates the method used in corner score calculation and shuffling procedures in the current work. (d) Same as c, but distributions of shuffled corner scores calculated with penalty applied. The red lines represent the 95th percentile of the distributions in (b). Note: this method was not used in this paper.

8. We apologize for the lack of clarity in our previous submission. Our intent was to convey that we did not use the penalization process when calculating shuffled corner scores. This is because shuffled rate maps often exhibit a greater number of fields than the number of corners, and thus applying the penalization lowers the 95th % score of the shuffled distribution (i.e. more neurons would actually be classified as corner cells) (Extended Data Fig. 2 c-d, copied above, note the red bars for each row are at the same value, which is the 95th % of

panel b). Thus, we did not use this penalization process in calculating shuffled corner scores for each cell, as we aimed to maintain the 95th percentile of the shuffled distribution as high as possible to ensure a stringent selection criterion for corner cells. To provide a clearer and more straightforward explanation, in the revised manuscript, we have included plots of the distributions of corner scores and shuffled corner scores for all subiculum neurons, both with and without the penalty enabled (Extended Data Fig. 2a-d, copied above). As the Figure shows, calculating corner scores for each cell with penalty shifted the distribution towards left compared to without the penalty (Extended Data Fig. 2a-b). In contrast, calculating shuffled corner scores for each cell without the penalty shifted the distribution towards right compared to with the penalty (Extended Data Fig. 2c-d). Thus, using the combination of b and c (indicated by the red box) gives the most stringent corner score criteria for defining corner cells. Critically, we believe the use of a relatively conservative score, such that we identified cells that very strongly encode convex/concave geometric features, aids in the characterization of this previously undescribed functional ‘cell type’.

How were the criteria for BVCs decided? They identified BVCs as those cells with significant spatial information and whose field covered more than 70% of the nearest wall. Given the current lack of observed BVCs in mice, it is important that the authors justify their choice of criteria (which might not be best applied with the same criteria from rat studies).

Fig. R2. (a) Rate maps of two example BVCs from two different mice in geometrically distinct environments. Each line is a cell with its activity tracked across all the environments (6 days). Note that the rectangle and convex arenas here were smaller than those used in the paper. **(b)** Proportion of BVCs identified using different boundary coverage thresholds from 0.5 to 0.9 in the square and rectangle environments. Solid line: mean; Shaded area: SEM. (n = 10 mice). **(c)** Related to Extended Data Fig. 10c. Left: Proportion of overlap between BVCs and corner cells when BVCs were defined using varying boundary coverage thresholds. Red line indicates the threshold level for random overlap. Right: Proportion of overlap between BVCs and corner cells (convex corners) when BVCs were defined using varying boundary coverage thresholds

9. This is an important question. First, we would note that border cells have been observed in mice (Giocomo et al., 2011; Mallory et al., 2018). While this is a different functional cell type terminology, border cells can be interpreted as short-range boundary vector cells (Barry et al.,

2006; Lever et al., 2009). Thus, we do not have any reason to suggest that mice should not have boundary vector cells. Consistent with this idea, in the current subiculum dataset, we reliably identified BVCs in multiple different geometrically shaped environments in every mouse that we imaged. In addition to the BVC examples we showed in Extended Data Fig. 10, two more examples of BVCs in the subiculum are shown here (Fig. R2a). These two neurons were from two different mice and showed clear and stable boundary coding across a series of environments with distinct geometries (Fig. R2a).

However, we also acknowledge the importance of establishing criteria to define boundary vector cells (BVCs) in mice, not only for the observations presented in the current manuscript but also for future research. As the reviewer suggested, we varied our boundary coverage threshold from 50% to 90% to investigate how this affected the classification of BVCs (See Fig. R2b). In addition to the boundary coverage criteria, we also introduced a within-session stability threshold of 0.3 for defining BVCs. At a 50% boundary coverage threshold, the proportion of BVCs was $12.1\% \pm 1.0\%$ of all recorded neurons in the square environment. Similarly, at a 90% threshold, there was still a significant proportion of neurons classified as BVCs in the square ($7.1\% \pm 0.6\%$) and rectangle environments ($5.5\% \pm 0.7\%$) (Fig. R2b).

Importantly, the primary conclusion in the current manuscript regarding BVCs is that BVCs and corner cells represent distinct neural populations in the subiculum. This conclusion holds across different BVC coverage thresholds (Fig. R2c). Namely, at any given boundary coverage threshold, the overlap between BVCs and corner cells (blue line in Fig. R2c) was consistently lower than the threshold indicating significant overlap (red line in Fig. R2c).

I have a major concern with the use of the GLM method to determine whether corner cells are allocentric or egocentric, which is a large claim of the paper. If rate maps are used to define the corner score of cells, and only those cells that are identified with significant corner scores are used for the GLM analysis, I am worried there is an inherent bias to detect allocentric tuning in corner cells. The corner cell metric based on rate maps has strong allocentric priors baked in – the spatial binning of neurons is in an allocentric reference frame. Perhaps some subicular cells have egocentric tuning, but would not pass the corner cell criterion as-is, and therefore could bias the analysis to suggest all corner-selective activity in the subiculum is allocentric. This seems to be double-dipping. I suggest the authors use the GLM method itself as an alternative means to detect corner cells, and compare results across metrics. Alternatively, the authors could devise a corner-cell detection method based on an egocentric reference frame, and demonstrate whether there are any cells that pass such a criterion.

10. We thank the reviewer for their thoughtful comments. First, we agree with the reviewer that the identified corner cells are allocentric but that the method for classifying corner cells considers their coding primarily in an allocentric reference frame, so a more unbiased approach to potentially identify egocentric coding is appropriate. To consider this question, whether there are neurons in the subiculum that encode corners in an egocentric reference frame, we modified our LN models in a manner inspired by recent publications that identified egocentric boundary or center bearing cells (Alexander et al., 2020; LaChance and Taube, 2023; LaChance et al., 2019).

In our initial submission, the LN model contained allocentric position (P), head direction (H), running speed (S) and egocentric corner bearing as predictors (Model 1, Extended Data Fig. 8d). This model is still presented in the revised version of the manuscript as Model 1. However, this is complemented by additional LN models (Model 2, 3 and 4, described below) aimed at identifying egocentric coding in a more unbiased manner. First, in the revised version of the manuscript, we replaced the allocentric position in Model 1 with egocentric corner distance (D) to facilitate the identification of egocentric corner cells (Model 2, Extended Data Fig. 9a, copied below). Notably, using either Model 1 or Model 2 on data from the 40-cm square

environment, revealed only 2 neurons that exclusively encoded egocentric corner bearing (E). However, egocentric corner bearing can also be encoded in conjunction with other behavioral variables (e.g., DE, HE, DHE). Pooling neurons from all Model 2 groups that contained egocentric corner-bearing (E) in the square environment revealed that $6.24 \pm 1.20\%$ ($n = 10$ mice) of

Extended Data Fig. 9. (a) Schematic of the linear-non-linear Poisson (LN) model framework with behavioral variables including egocentric corner bearing (E), egocentric corner distance (D), allocentric head direction (H) and linear speed (S) (Model 2). (b) Proportion of subiculum neurons that were classified by Model 2 in the large square and convex-1 environments, respectively ($n = 10$ mice). Including neurons from all the model groups containing egocentric corner-bearing (E, highlighted in red) in the square environment, we found that 6.24 ± 0.6 ($n = 10$ mice) of the all recorded subiculum neurons encoded egocentric corner bearing (e.g. E, SE, HSE). Similarly, 3.1 ± 0.6 subiculum neurons encoded egocentric corner bearing for convex corners in the convex-1 environment. For the box plots, the vertical line indicates median, and the box indicates 25th and 75th percentiles. The whiskers extend to the most extreme data points without outliers (+). (c) Encoding for egocentric corner bearing, particularly in rotationally symmetric environments, could potentially be confounded by other correlated variables, such as egocentric wall bearing or egocentric center bearing. To rule out the possibility that the observed encoding of egocentric corner bearing in Model 2 was actually due to tuning to egocentric wall or center locations, we trained two separate LN models in which egocentric corner bearing and corner distance was replaced by egocentric wall bearing and wall distance (Model 3) or egocentric center bearing and center distance (Model 4). Since Models 2, 3 and 4 were trained and tested using the exact same data, we compared the model fitting of neurons with egocentric corner modulation in Model 2 to the fitting of the same neurons in Model 3 and Model 4. Neurons that exhibited a significantly better fit (higher increased correlation, $n = 10$ -fold) in Model 2 compared to Model 3 or Model 4 were considered as potential neurons encoding egocentric corner bearing. Finally, to rule out the possibility that egocentric corner coding could artifactually result from the conjunction of position and head direction, we also compared the neurons' fittings in Model 2 to the position and head direction groups (P, H, PH, PHS) in Model 1. (d) Left: Proportion of egocentric corner cells in the subiculum from the square, rectangle, right triangle, and convex-1 (bearing to convex corners) environments. Right: Egocentric corner cells primarily encode egocentric corner bearing in conjunction with other behavioral variables, most often with allocentric head direction. (e) Representative egocentric corner cells from the square, rectangle, right triangle, and convex-1 (bearing to convex corners) environments. Each column represents a neuron. The first row shows spike raster plots color coded with egocentric corner bearing on top of the animal's running trajectory (grey lines). S

all

recorded subiculum neurons encoded egocentric corner bearing (But, see next paragraph). Similarly, $3.1 \pm 0.6\%$ of all subiculum neurons encoded egocentric convex corner bearing in the convex-1 environment. Note that this analysis also showed that very few allocentric corner cells (as identified using the tuning curve method) overlapped with those cells modulated by egocentric corner bearing (6 out of 217 cells in the square environment from 10 mice).

However, encoding for egocentric corner bearing, particularly in rotationally symmetric environments, could potentially be confounded by other correlated variables, such as egocentric wall bearing (circular correlation with corner bearing = 0.43) or egocentric center bearing (circular correlation with corner bearing = -0.73)(LaChance et al., 2019). To rule out the possibility that the observed encoding for egocentric corner bearing in Model 2 was actually due to tuning to egocentric wall or center locations, we next trained two separate LN models in which egocentric corner bearing and corner distance was replaced by egocentric wall bearing and wall distance (Model 3, Extended Data Fig. 9c) or with egocentric center bearing and center distance (Model 4, Extended Data Fig. 9c). Since Models 2, 3 and 4 were trained and tested using the exact same data, we compared the model fitting of neurons with egocentric corner modulation in Model 2 to the fitting of the same neurons in Model 3 and Model 4. Neurons that exhibited a significantly better fit (higher increased correlation, $n = 10$ -fold) in Model 2 compared to Model 3 or 4 were considered as potential neurons encoding egocentric corner bearing. Finally, to rule out the possibility that egocentric corner coding could artifactually result from the conjunction of position and head direction (LaChance et al., 2019), we also compared the neurons' fittings in Model 2 to the position and head direction groups (P, H, PH, PHS) in Model 1 (Extended Data Fig. 8). After applying these criteria, we observed that $0.28 \pm 0.09\%$ of all neurons showed significant egocentric corner coding in the square environment, $0.75 \pm 0.24\%$ in the rectangle environment, and $0.14 \pm 0.05\%$ in the convex-1 environment (convex corners) (Extended Data Fig. 9d). Of note, we observed no overlap between egocentric corner cells and corner cells we identified using the tuning curve method (allocentric) in the square environment, and only two neurons overlapped in their classification in the rectangle and convex-1 environments.

Finally, to further disentangle the correlations among egocentric bearing variables in rectilinear environments, we repeated the same analysis (as described above) in the right triangle environment. In the right triangle, the circular correlation between corner and wall bearings decreases to 0.09, and the correlation between corner and center bearings shifted to -0.38. Correlations between egocentric distances also shifted by 0.2 to 0.4 towards zero. Despite the decrease in behavioral variable correlations, the number of neurons encoding egocentric corner bearing remained low in the right triangle ($0.68 \pm 0.50\%$ of all neurons, see Extended Data Fig. 9d). This result suggests that the number of egocentric corner cells in the subiculum is inherently low, even when the tuning to corner versus wall/center becomes sufficiently distinct. Taken together, our results demonstrate that a small number of neurons in the subiculum show egocentric corner modulation (see Extended Data Fig. 9e). These new results are now presented in Extended Data Fig. 9.

A general claim of the paper is that the subiculum represents the geometry of familiar environments. I feel the authors could do more with their large-scale population recordings to this effect. Namely, manifold embedding techniques can extract neural representations from such large-scale population recordings. The authors could present evidence that the subiculum in fact constructs geometric representation, using such procedures given the power that large population recordings afford with miniscope calcium imaging. Further, if the subiculum contains geometric information, then it should be possible to decode the geometry of environments across sessions. The authors should show if this is the case in animals that have been recorded across many geometries, across rooms.

11. We thank the reviewer for their thoughtful suggestions regarding manifold embedding and

1k-l. (k) Three dimensional (3D) embedding of the population activity of all recorded subiculum neurons in the triangle, square, and hexagon from a representative mouse. Each dot corresponds to the population state at one time point. Time points within 5 cm of the corners are color-coded, with the color graded to grey as a function of the distance away from the corner (as shown in the inset). For more examples, see Extended Data Fig. 5a-c. **(l)** Left: An example of coding the animal's quadrant location over time using a decoder trained on corner cell activity. Black line: true quadrant location; red dotted line: decoded quadrant location. Right: Quadrant decoding accuracy for the decoder trained on corner cell activity versus shuffle (mean \pm SEM: decoder vs. shuffle: 0.35 ± 0.02 vs. 0.26 ± 0.006 ; two-tailed Wilcoxon signed-rank test: $n = 9$ mice).

geometric decoding. Following these suggestions, we performed a two-step dimensionality reduction method based on a previous publication (Gardner et al., 2022). First, to improve robustness to noise, we performed a principal component analysis (PCA) on all the data recorded from the subiculum population for each mouse in a specific environment. Next, we chose the top 10 principal components to perform Uniform Manifold Approximation and Projection (UMAP), effectively reducing these 10 principal components to create a 3D visualization. As shown in Fig. 1k (copied below) and Extended Data Fig. 5a – c (copied below), each dot on the low-dimensional manifold represents the population state at one time point (a 67ms time bin). Time points within 5 cm of the corners are color-coded, with the color graded to grey as a function of the distance from the corner (as shown in the schematic). Across different mice, we consistently observed that corners were well represented on the low-dimensional neural manifold. The representation of each corner for a given environment was distinct from other corners and the rest of the space - in that we observed a grouping for the representation of each corner that was distinct from the rest of the representation. Furthermore, the sequential order of corners was effectively preserved in the low-dimensional neural manifold. For instance, in the hexagon, the orange corner was positioned next to the yellow corner but opposite of the green corner (Fig. 1k and Extended Data Fig. 5c).

On the other hand, corner representations also converged at a specific point on the manifold (as indicated by the black circles in Extended Data Figure 5a-c). This convergence suggests that subiculum neurons also generalize the concept of corners, in addition to representing their distinct locations. This generalization of corner coding is consistent, for example, with our observation that concave corner coding cells also encode a new corner when it is inserted into the environment (Figure 3) and encode concavity in oval environments (Figure 5). A prediction of this 'separated yet connected' corner representation is that corner cells only modestly encode the precise allocentric location of corners (e.g. the northwest versus the southwest corner), as they also more generally encode the concept that a corner exists. To test this prediction, we employed a quadrant decoding analysis (new Figure 1l), in which we trained a decoder solely using corner cells and then examined decoding accuracy for which quadrant of the square environment the mouse was located. While decoding performance significantly exceeded chance levels (consistent with modest encoding of the precise allocentric location of a given corner), the accuracy of the decoding was only moderate (~35%, Figure 1l), consistent with the idea that corner cells also generalize their coding to all corners.

Next, as suggested by the reviewer, we performed a new analysis in which we examined how well we could decode the geometry of an environment (i.e. identity) from all neurons we recorded from the subiculum. First, we trained the decoder on data from within a 4-cm radius around the center of the environment. We then examined the decoding accuracy of the

geometry across circular, triangular, square, and hexagonal arenas. We observed a decoding accuracy = 79%. While this decoding accuracy was higher than expected by chance, one possibility is that this decoding reflects neurons that 'remap' (i.e. move the spatial location where they are active) across the different geometries, a phenomenon that has been well-documented in previous studies of hippocampal place cells (Bostock et al., 2004; Colgin et al., 2010; Knierim et al., 1998; Leutgeb et al., 2005; Leutgeb et al., 2004; Lever et al., 2002; Muller and Kubie, 1987; O'Keefe and Conway, 1978; Wills et al., 2005). However, based on the notion that remapping across geometrically distinct environments is stochastic, one possibility is that the decoding accuracy should be the same if using data from other environmental locations. In contrast, when we performed the decoding analysis using the data from positions close to geometric features of the environment, such as boundaries and corners (matched to the data length for the decoder using data from the 4 cm radius around the center of the environment, Extended Data Fig. 5e), the decoder performed better (85%) than when we used the data from the center (away from geometric features). This improvement in decoding accuracy cannot be attributed to stochastic remapping of neurons. Instead, it supports the hypothesis that data near the geometric features of the environment carries more information about the overall environmental geometry. In summary, these results suggest that the population code of the subiculum effectively encodes the geometric information of the

environment. These new analyses are now presented in Figure 1 and Extended Data Fig 5 of

Extended Data Fig. 5. (a) Three-dimensional (3D) embedding of the population activity of recorded subiculum neurons in the triangle from four different mice. We applied a sequential dimensionality reduction method using PCA and UMAP to obtain this neural manifold embedding (Methods). Each dot represents the population state at one time point. Time points within 5 cm of the corners are color-coded, with the color graded to grey as a function of the distance away from the corner (top row). The example from mouse 1 is the same as Fig. 1k but from a rotated view. The black circle denotes the place that corner representations converged on each manifold. (b) same as (a), but in the square environment. (c) same as (a), but in the hexagon environment. (d) Left: Schematic of using the data from the environmental center (8 cm diameter) to decode the environmental identity (geometry). Right: A decoding example to predict the geometry of the environment. (e) Left: Schematic of using the data near (within 8 cm) a geometric feature of the environment (e.g., a corner) to decode the environmental identity (geometry). Right: A decoding example to predict the geometry of the environment. (f) Comparison of decoding accuracy in (d) and (e) (mean \pm SEM: center vs. corner: 0.79 ± 0.02 vs. 0.85

the revised version of the manuscript.

While the authors used relatively rigorous methods to detect corner cells, it is still unclear to me how many cells would be identified as corner cells by chance based on

the combined shuffling and corner distance measures. A shuffling-only criterion with significance considered at $p < 0.05$ might yield 5% of cells as corner cells, but it's unclear with the additional corner distance measure how this would affect chance levels of corner cell detection. I suggest the authors add a control analysis wherein they probe randomized (simulated) corner locations in each environment, and detect how many cells would be classified as corner cells if simulated "corners" were randomly distributed throughout the environment. It seems they attempted a similar procedure with simulated place cells in random locations, but I suggest they attempt the randomization of "corner locations" using their actual data.

12. The reviewer raises a very important point and we thank the reviewer for suggesting this control analysis. Following the reviewer's suggestion, first, we used the data from the circle, triangle, square, and hexagon environments and manually assigned evenly spaced points on the walls to serve as 'corners' (Extended Data Fig. 1l, copied below). We then re-performed all the procedures used to classify corner cells using these 'corners'. We found the numbers of cells classified as corner cells were extremely low in this condition, with their proportions below 0.5% of all recorded neurons (Extended Data Fig. 1l). We also tried to assign completely random locations as 'corners' in the environments and the results were essentially at zero (0.01% of all recorded neurons). In addition, we ran the same analysis using the data from the right triangle and trapezoid and found the proportion of cells classified as corner cells were less than 0.1% in both environments (Extended Data Fig. 1n).

Furthermore, as mentioned above, we examined a population of 5212 neurons in the hippocampal CA1 from 12 mice, imaged using miniscopes during exploration of a square box. In this hippocampal dataset, we observed 34 neurons in total from the 12 mice that passed the corner cell criteria. This is only $0.62 \pm 0.14\%$ (mean \pm SEM, $n = 12$) of the total recorded CA1 neurons (Extended Data Fig. 1m). Together, these results support the idea that

Extended Data Fig. 1l-n. (l) Proportion of neurons that passed the definition for a corner cell when corners of the environments were manually assigned to the walls. Red dots in the bottom schematic denote the locations that were assigned as 'corners'. (m) Left: Proportion of corner cells in CA1 ($n = 12$ mice). Right: Rate maps of two example CA1 corner cells. Peak spike rates (fr) and corner scores (c) for the cells are indicated at the bottom. (n) Same as (l), but for the right triangle and trapezoid environments

identified corner cells in the subiculum do not reflect chance levels

The sub-heading on line 160 is misleading. From my reading of the paper and respective figures (Extended data Fig 3), they did not observe any effect of darkness + whisker trimming on corner cell coding in the subiculum, but only on subicular place cells. Thus, they cannot claim that corner cells depend on visual and/or tactile senses given they have not demonstrated that combined darkness and whisker trimming impair corner coding, and I do not believe they have strong grounds to speculate on this (line 178).

13. We thank the reviewer for pointing this out. In the revised manuscript, in which we have applied the updated criteria for defining corner cells, recording in the dark and with trimmed

whiskers still had no effect on the number of corner cells. However, the peak spike rate of corner cells in darkness slightly decreased compared to the baseline condition. This effect was not observed in the whisker trimming condition. Thus, we have amended the language of this section to indicate that visual information, in part, contributes to corner coding in the subiculum. We also agree with the reviewer (and reviewer 2) that this is not a key component of the manuscript and we have also reduced some of the text related to this finding.

The authors suggest that concave and convex corner coding exist in two separate subpopulations in the subiculum. However, I do not see what this adds to their conclusion. If subicular neurons encode corners, irrespective of concavity/convexity, and their tunings parameters are highly specific, then you would expect these cells to be non-overlapping, and distributed with a salt-and-pepper topography. In the spirit of not over-complicating a growing list of functional cell types in the parahippocampal system, I suggest they scrap this added distinction, since it is not necessary to view these as functionally heterogeneous cell types. I do not see there would be a theoretical value in treating these concave and convex corner cells as distinct populations.

14. We thank the reviewer for this suggestion. We have removed unnecessary text stating concave and convex corner cells and now use corner cells instead, whenever possible. We did not intent to emphasize language that would increase the number of functionally-defined cell types, rather we intended to emphasize the visual appearance of the corner that the cells are encoding.

Minor comments:

In illustrations of the environment (e.g., Fig 1.) the authors should show where the visual cues were in the apparatus throughout all geometric manipulations relative to the recorded shape (described in methods, line 356-7).

We have added this information to our figures.

The authors should add one sentence stating why the additional polygon masking procedure was implemented for convex corner cell detection (line 537).

In the convex environments (e.g., convex-1), when the distance between the location of a field (e.g., a field at a concave corner in the convex-1 environment) and the environment center is greater than the distance between the center and the convex corner, the corner score becomes non-linear (see the illustration below). We have now added an explanation for using the polygon masking method in convex environments to the methods section, stating: "The reason for using the polygon mask is to avoid nonlinearity in corner score calculation in the convex environment, particularly, when the distance between the location of a field (e.g., a field at a concave corner in the convex-1 environment) and the environment center is greater than the distance between the center and the convex corner."

In the description of the anatomical clustering method (line 564), they hypothesize that if functionally-defined neuronal groups were anatomically clustered, then inter-group distance should be greater than intra-group distance. I suggest the authors make this clearer in the main text and accompanying figure, as this will help readers to interpret the lack of clustering shown.

We have added the relevant information into the main text and the corresponding figure legend.

The authors should specify that their deconvolution procedure was used to infer spikes, but not to detect spikes per se. A small change in language would suffice. For example, on line 60, they could instead state: “Calcium signals were extracted with CNMF and OASIS deconvolution, and subsequently binarized to estimate spikes for all cells.”

We have changed the language as the reviewer suggested.

On line 66, the authors state many cells exhibited place-cell like firing patterns that did not change significantly across environments. Given the coarseness of this spatial firing, and lack of remapping across environments, I suggest they rephrase this description to suggest that the firing across environments was “spatially modulated but not geometry-specific”.

We have changed the language as the reviewer suggested.

The authors should state approximate atlas coordinates where they imaged in the subiculum based on the approximate, combined lens implant and expression location (line 74).

We measured the effective image size (the area with detectable neurons) for each mouse and combined this information with histology. The anatomical region where neurons were recorded was approximately within a 450- μ m diameter circular area centered around AP: -3.40 mm and ML: +2 mm. Due to the limitations of 1-photon imaging, we could not accurately estimate the imaging depth, but we believe the recordings were primarily from the deep layer of the subiculum. This information has now been added to the Methods.

Referee #2 (Remarks to the Author):

Environmental geometry is an important, sometime overriding, feature that animals use to orient and navigate in an environment. The present study investigates the coding of convex and concave corners of an environment by recording the calcium activity of large populations of neurons in the subiculum of freely moving mice. The authors make a thorough case that a subset of subiculum neurons encode these corners, in addition to the known place and boundary-related firing already described in the subiculum. This is an important study and represents the most complete characterization of corner-coding properties in the larger hippocampal formation. The results have strong implications for how this system encodes spatial environments, adding to the list of cell types in this system such as place cells, grid cells, head direction cells, border cells, vector cells, and others that collectively allow the system to recognize different environments, localize themselves within an environment, and guide the animal through an environment as it navigates the terrain.

The paper is written clearly and the data are presented clearly and comprehensively. There are a number of concerns with the present version of the manuscript, however, mostly related to improper statistical procedures and questions about the robustness of some of the effects. These concerns must be addressed before the paper can be published.

We thank the reviewer for their constructive feedback. Below, we describe the new analyses, figures and text that we believe addresses the concerns the reviewer raised. Note that we have numbered the comments for ease of reference throughout our response to reviewer comments.

Major

1) The first problem is that the authors define a corner cell based on it exceeding the 95th percentile of a shuffled distribution, which is a standard procedure. However, in many cases, the percentage of cells that are classified as corner cells are at or below the predicted percentage based on chance. That is, using this shuffling test, one expects that ~ 5% of cells will pass the test by chance, and it is incumbent on the authors to demonstrate that the percentage of cells actually passing the criterion exceed the chance level by a statistically significant amount. In a number of cases, the percentage of neurons that pass their test is barely above, or below, 5% (lines 85, 113-114, 131, 204; Extended Data instances not included here). Although many examples in the paper are clearly corner cells (especially in the square and triangle environments), it is not clear whether these are representative or not. If they are representative, rather than best examples (this should be stated either way), then it appears that the shuffling procedure is not doing what the authors intend. (The hexagon corner cells are less convincing, as are the asymmetric environments.) It is possible that the percentages of corner cells are much less than those that pass the shuffling test because of the second criterion of field distance (line 82-83); if so this should be explained. The authors need to resolve this statistical problem to provide any confidence in the validity of the percentages of corner cells and devise a test to show that the percentage of corner cells in the various experiments exceeds that expected by chance. I recognize that this is difficult when the true prevalence of a cell type in the population is near or below the standard alpha level of 0.05, but it is a problem that needs to be addressed in some fashion.

15. This is a very important comment and we thank the reviewer for raising this point. In the revised manuscript, we addressed this issue by applying several methods and approaches to ensure that the corner cells we identified did not reflect chance levels. Below, the first four

points describe why the corner cell numbers are low and the fifth to seventh points describe analyses and datasets used to determine the level of corner cells expected by chance.

First, we would note that we aimed to use a relatively conservative score such that we identified cells that very strongly encode convex/concave geometric features. We felt this would aid in the characterization of this previously undescribed functional 'cell type'.

Second, the stringent score threshold in part reflects the fact that we calculated the shuffled corner scores without penalization (see also response 8 to reviewer 1). This is because shuffled rate maps often exhibit a greater number of fields than the number of corners, and thus applying the penalization lowers the 95th % score of the shuffled distribution (i.e. more neurons would actually be classified as corner cells) (Extended Data Fig. 2a-d). Thus, we did not use this penalization process when calculating shuffled corner scores, as we aimed to use a relatively conservative 95th% threshold. To clarify this in the revised manuscript, we plotted the distributions of corner scores and shuffled corner scores for all the subiculum neurons with and without the penalization (see Extended Data Fig. 2a-d). As shown in the Figure, calculating corner scores for each cell with penalty shifted the distribution towards left compared to without penalty (Extended Data Fig. 2a-b). In contrast, calculating shuffled corner scores for each cell without penalty shifted the distribution towards right compared to with penalty (Extended Data Fig. 2c-d). Thus, using the combination of Extended Data Fig. 2b and c (red box) gives the most stringent corner score criteria for defining corner cells. This resulted in a very low random appearance of corner cells (much lower than 5%, see point fifth below and Extended Data Fig. 1l-n, copied below).

Third, as the reviewer mentioned, we implemented a criterion on the distance between fields to classify corner cells (Extended Data Fig. 1j, copied below). Namely, the distance between any two fields (major fields, if the number of fields > number of corners) needed to be greater than half of the distance from the corners to the center of the arena (panels d and c in Extended Data Fig. 1f). This criterion was applied to avoid classifying neurons with multiple fields clustered around a single or a subset of corners as corner cells. The field distance criterion filtered out one-third ($33.5\% \pm 3.9\%$, $n = 9$ mice) of the corner cells that passed the shuffling threshold.

Fourth, as suggested by reviewer 1, we also added another criterion for identifying corner cells – within session stability. Extended Data Fig. 1k shows the within-session stability distribution of all the corner cells that passed the criteria from i to j in the triangle, square, and hexagon sessions before applying this stability criterion (n = 1018 cells from 9 mice). In the revised manuscript, corner cells need to have a within session (two halves) stability value of > 0.3, as

Extended Data Fig. 1f-k. **(f)** Left, the definition of the corner score for a given spatial field ($\text{cornerscore}_{\text{field}}$). $d1$: distance from the center of the arena to the field; $d2$: distance from the field to the nearest corner; dc : the mean distance from the corners to the center of the arena. Right, the distribution of $\text{cornerscore}_{\text{field}}$ in a square environment for a given field. Namely, this represents the corner score you would expect if a neuron was active in a given pixel of this plot. Note that the corner score can range from -1 (blue) to 1 (green). **(g)** The definition of the corner score for a given cell ($\text{cornerscore}_{\text{cell}}$, see Methods). **(h)** An example corner cell with corner score values for each field labeled in red, the final corner score for this cell is shown below. **(i)** Shuffling of $\text{cornerscore}_{\text{cell}}$ to determine a threshold for classifying a neuron as a corner cell. This example is from the same cell as in (h). **(j)** To be classified as a corner cell, the distance between any two fields (major fields, if the number of fields > number of corners) needed to be greater than half of the dc value, as indicated by the blue line in (f). **(k)** As an additional criterion, to be classified as a corner cell, within-session stability needed to be greater than 0.3 (Pearson's correlation between the two halves of the data). The distribution shows the within-session stability of all corner cells from the triangle, square, and hexagon sessions

determined by the random within session stability level using shuffled spikes.

Fifth, to determine the random level of cells that can pass all of our corner cells criteria, we used our subiculum data from the circle, triangle, square, and hexagon environments and manually assigned evenly spaced points on the walls to serve as 'corners' (Extended Data Fig. 1l, copied below). We then re-performed all the procedures used to classify corner cells using these 'corners'. We found the number of cells classified as corner cells were extremely low in this condition, with their proportions below 0.5% of all recorded subiculum neurons (Extended Data Fig. 1l). We also tried to assign completely random locations as 'corners' in the environments and the results were essentially at zero (0.01%). In addition, we ran the same analysis using the data from the right triangle and trapezoid and found the proportion of cells classified as corner cells were less than 0.1% in both environments (Extended Data Fig. 1n).

Sixth, we examined a population of 5212 neurons in the hippocampal CA1 from 12 mice, imaged using miniscopes during exploration of a square box. In this dataset, we only observed

34 neurons in total from the 12 mice that passed the corner cell criteria. This is only $0.62 \pm 0.14\%$ (mean \pm SEM, $n = 12$) of all recorded CA1 neurons (Extended Data Fig. 1m, copied below). This proportion is significantly lower than the square-classified corner cells in the subiculum (Mann-Whitney test: $p < 0.0001$). Together, we believe these analyses and data sets support the idea that the identified corner cells in the subiculum does not reflect chance. Moreover, this analysis specifically also reveals that the subiculum exhibits corner coding that is distinct from coding in the hippocampal CA1 region.

Seventh, we would note that the percentage of cells we identify as corner coding neurons is not dissimilar from the percentage of cells identified as border cells (less than 10% of neurons) (Solstad et al., 2008) or grid cells (in mice, $\sim 8\%$) (Mallory et al., 2018) in medial entorhinal cortex; functionally defined cell classes that have been foundational in understanding how the brain encodes an internal map of external space.

Finally, we have also added all the identified corner cells from a representative mouse in Extended Data Fig. 3 to give provide a clearer picture of the repertoire of tuning curves observed in this functionally defined cell population and replaced some corner cell examples (e.g., asymmetric environments) to reflect more representative examples.

Extended Data Fig. 1l-n. (l) Proportion of neurons that passed the definition for a corner cell when corners of the environments were manually assigned to the walls. Red dots in the bottom schematic denote the locations that were assigned as 'corners'. (m) Left: Proportion of corner cells in CA1 ($n = 12$ mice). Right: Rate maps of two example CA1 corner cells. Peak spike rates (fr) and corner scores (c) for the cells are indicated at the bottom. (n) Same as (l), but for the right triangle and trapezoid environments.

2) Similarly, the percentage of corner cells in the circle is much less than 5%, which would indicate that these fictive corner cells are less likely than chance to be identified in the circle; one would have expected that the number should be $\sim 5\%$. Were the arbitrary points at the edges of the circle or could they be anywhere? It would seem that they should be at the edges to match the real corner conditions.

16. The reviewer was correct that the points assigned as 'corners' in the circle environment were on the edge of the circle. In the revised manuscript, we have tried to assign either 3 or 4 points to the circle (Extended Data Fig. 1l and Fig. 1g, respectively) to serve as 'corners'. As explained above, we used a very stringent shuffle criteria and applied additional criteria to identify corner cells, we thus did not observe many cells that passed the corner cells criteria in the circle environment (Fig. 1g: $0.04 \pm 0.03\%$; Extended Data Fig. 1l: $0.0 \pm 0.0\%$).

3) In some analyses, the authors analyze at most 2 sessions/mouse and use the session as the unit of analysis, which is a welcome improvement over many studies that combine data over many sessions that record unknown numbers of repetitions of the same cells, and use the same repeated cells as the unit of analysis. However, this is still a problem, given that in the two sessions the authors are probably recording from a very large percentage of the same cells twice, making the two recording sessions not

independent samples and violating the statistical tests. The authors should either take an average of sessions for each mouse and use that as the measure, or choose one session per mouse based on unbiased criteria. It is unlikely to make any difference in the results or interpretation, but is the proper statistical methodology. For example, ED Figure 1K should replace Fig 1g, and similar throughout manuscript where necessary.

17. We agree with the reviewer that this is a more appropriate methodology. In response to the reviewer's suggestion, we've changed our data quantification approach. Instead of using individual sessions, we've shifted to using each mouse as the unit of analysis by averaging sessions within each mouse, as suggested by the reviewer. All averages were calculated across sessions within each mouse. We did not pool all the cells from multiple sessions together and then average them. For example, in Fig. 1g, the proportion of corner cells was determined by averaging the proportions of corner cells in session 1 (a single number) and session 2 (a single number). Similarly, in Fig. 1l, the decoding accuracy for each mouse was averaged using the mean decoding accuracy of session 1 (a single number) and session 2 (a single number).

Extended Data Fig. 2h is the only place we've retained the use of sessions for quantification. This is because identified corner cells were slightly different from session to session and we want the readers to see the contribution of corner cells to spatial decoding for each session. Nevertheless, it's important to note that we conducted thorough cross-validation, and both session and mouse-based approaches yielded consistent and identical results in our statistical analyses.

4) The statistical methodology used to compare the firing of corner cells near and far from the corners to the corresponding firing of non-corner cells is fundamentally flawed. A core assumption of the statistical tests is that the data are sampled randomly from the two populations being compared. In this case, the authors are identifying cells from a single sample that have a certain characteristic (corner-related firing) and then comparing that biased subsample to the remainder of the sample. It is similar to selecting a sample of tall people from a population and then performing a statistical test on whether these tall people are taller than the rest of the population. Any p values that come from such a test are statistically meaningless because of this violation of the assumption of random sampling. Thus, Figure 1h and all further results based on the same procedure (Fig 2k, 4d, ED Fig 2g,h) are invalid statistical tests and need to be removed from the manuscript (hopefully replaced by another analysis that is statistically valid).

18. To provide some background for the inclusion of these graphs, the goal of presenting the data in this manner was to validate that cells we identified as corner cells using the novel corner score did indeed show higher firing rates near corners. However, we also agree with the reviewer that some of these statistics were invalid, particularly in the cases of direct comparisons between corner cells and non-corner cells from the same session. With some exceptions (described below), we have removed this statistical test and present the data for illustration purposes only. However, for some of the comparisons, we used cells that were not corner cells in a given session, and compared them to other non-corner cells from the same

Fig. 2k. (k) Positional spike rates plotted relative to the distance to the nearest corner in the 15 cm wall square condition. Blue curve indicates neurons that were a corner cell in the 30 cm wall arena (green check mark) but not in the 15 cm wall arena (red cross). Grey curve indicates other non-corner cells in the 15 cm wall arena. Statistical tests were compared between the blue and grey curves within ~5 cm of the corners (head of the curve, averaged from the first 3 bins, 1 bin = 1.6 cm) and ~5 cm of the environmental center (tail of the curve, averaged from the last 3 bins) (two-tailed Wilcoxon signed-rank test: head of the curve: $p = 0.0002$; tail of the curve: $p = 0.0002$; $n = 9$ mice).

session, a type of comparison we believe to be statistically valid. For example, in Fig. 2k (copied above), we identified neurons that were classified as corner cells in the 30 cm wall square but were not corner cells in the 15 cm wall square. We then compared their activity in the 15 cm wall square with other non-corner cells in the 15 cm wall square. Importantly, both groups, represented by the blue and grey traces, were not corner cells in the 15 cm wall square. We used this comparison to infer whether neurons in the blue trace still show any corner-related activity (or not at all as other non-corner cells), even though they did not pass our classification criterion in the 15 cm wall square. A similar rationale applies for Fig. 2D (top two panels) and Extended Data Fig 4g-h.

5) Figure 1j suffers from the same problem, as the authors are preselecting the cells that have stable corner firing across sessions and then testing whether these cells have high correlation across sessions (if they are preselected accurately, then of course they will). There is no purpose in doing a statistical test on this biased sample.

19. We understand the reviewer's point. However, we only presented the values of the cross-session stability in the manuscript and did not perform any statistical test on Fig. 1j.

6) Lines 99-100: 2.5% of neurons were consistently classified as corner cells across all non-circle environments. What is the chance level? Does 2.5% statistically significantly exceed this chance level?

20. There were zero neurons classified as across session corner cells when we used the identified 'corner cells' from Extended Data Fig. 1l (copied above) to perform the same characterization. In the revised manuscript, as we applied more criteria for defining corner cells (e.g., within session stability), there were 1.7% neurons classified as across session corner cells. This proportion is significantly higher than zero (Wilcoxon Signed Rank test against zero: $p = 0.0039$).

7) Lines 130-131, 138-139: It is not appropriate to do parametric statistics comparing percentages across groups, as percentage data are not normally distributed. An arcsin transform is required, or the use of nonparametric tests.

21. We thank the reviewer for this suggestion. As we switched to using data averaged across sessions from each mouse instead of data from each session, some of the data that was previously normally distributed is no longer normally distributed. Therefore, we switched all of our statistical tests to nonparametric tests in the revised manuscript. Please note that we also performed parametric tests in parallel, and confirmed that all the comparisons using either parametric or nonparametric tests consistently led to the same conclusions.

8) In a number of places, statements are made that two groups (e.g., corner cells vs noncorner cells) have different responses to a particular manipulation. However, it is necessary to statistically compare these groups directly to each other to support this claim. It is not sufficient to merely show a significant effect for one group and a nonsignificant effect for the other. See Nieuwenhuis et al. (2011) doi:10.1038/nn.2886. Examples: Fig 5f, 5g, Extended Data Fig 2h, ED Fig 4j vs. 4k, ED Fig 4l vs. 4m. In each case, 2-way ANOVAs need to be run to test for main effects of the 2 factors and interactions.

22. We appreciate the reviewer's suggestion and agree with the reviewer that including a comparison between corner and non-corner cells in these statistics is important. The key metric we are interested in examining is whether a given neural population (e.g. corner cells) shows a change spike rate between two geometrically differing regions (high concavity/convexity and low concavity/convexity). This, using Fig. 5e-h as an example (copied

below), we changed this analysis such that we first calculated the difference between high and low concavity/convexity regions for each cell type and tested these differences against zero. The results indicated that corner cells exhibited values higher than zero, while non-corner cells did not. We then compared these differences between corner and non-corner cells and observed that corner cells had higher values than non-corner cells. We implemented a similar modification for the other comparisons mentioned by the reviewer, and our conclusions remained unchanged.

Extended Data Fig. 11-n. (e) Illustration showing the high versus low concavity regions in the oval arena that were used for quantification. (f) Spike rate differences between high and low concavity regions in the oval arena for both corner and non-corner cells. Corner cells were identified in the square environment. (two-tailed Wilcoxon signed-rank test against zero: corner cells: $p = 0.0039$; non-corner cells: $p = 0.16$; two-tailed Wilcoxon signed-rank test: corner cells vs. non-corner cells: $p = 0.0039$; $n = 9$ mice, data averaged from day 2 and day 3 for each mouse). (g) Illustration showing the high versus low convexity regions around the objects that were used for quantification. (h) Spike rate differences between high and low convexity regions around the objects for both corner and non-corner cells. Corner cells encoding convex corners were identified in the convex-1 environment. (two-tailed Wilcoxon signed-rank test against zero: corner cells: $p = 0.016$; non-corner cells: $p = 0.58$; two-tailed Wilcoxon signed-rank test: corner cells vs. non-corner cells: $p = 0.016$; $n = 7$ mice).

9) Lines 167-169. The analysis of ED Fig 3c,d does not address the question posed here. The statement on the previous sentence is that color and texture did not affect the activity of corner cells. ED Fig 3c,d show that the proportions of Corner cells and the peak rates of corner cells in each session on average do not change based on nonspatial changes. They do not show that individual cells are not affected (e.g, by firing rate changes, some corner cells losing their corner tuning while other cells gain corner tuning, etc.).

Extended Data Fig. 6a-f. (a) Schematic of a shuttle box composed of two compartments that differed in their visual and tactile cues. (b) Two example corner cells from two different mice recorded in the shuttle box shown in (a). Raster plot (left) indicates extracted spikes (red dots) on top of the animal's running trajectory (grey lines) and the spatial rate map (right) is color coded for maximum (red) and minimum (blue) values. (c) Proportion of neurons classified as corner cells in the grey vs. black compartments of the shuttle box (two-tailed Wilcoxon signed-rank test: $p = 0.91$; $n = 9$ mice). (d) Average corrected peak spike rates of corner cells at the corners in the grey vs. black compartments (two-tailed Wilcoxon signed-rank test: $p = 0.43$; $n = 9$ mice). Corner cells included in this quantification were defined as corner cells in both grey and black compartments. (e) Same as (d), but using corner cells that defined in the grey compartment (two-tailed Wilcoxon signed-rank test: $p = 0.07$). (f) Same as (d), but using corner cells that defined in the

23. We thank the reviewer for noting this. In the revised manuscript, we have included additional quantification that supports our conclusion (Extended Data Fig. 6d-f, copied above). First, we compared the spike rates between the grey and black compartments for neurons that were classified as corner cells in both compartments. Second, we compared the spike rates between the grey and black compartments for neurons that were classified as corner cells only in grey compartment. Finally, we also compared the spike rates between the grey and black compartments for neurons that were classified as corner cells only in black compartment. For all three comparisons, we did not observe any spike rate differences across the two compartments, suggesting corner cells are not sensitive to the colors and textures of the environment.

10) Lines 172-173: Similar concern as Item 7

24. We have changed these comparisons to a non-parametric test.

11) Lines 206-207: Given how rare the corner cells are, it is not surprising that the combination of these properties is even smaller. However, it may still be the case that corner and convex tuning appears in the same population of cells across conditions more than expected by chance. The authors need to perform some calculation and test to determine if these numbers truly reflect different populations or whether they just appear that way due to the low prevalence of these properties. Line 222, 276-77 may be overstatements if this result is not confirmed statistically. At present, I am not convinced the authors have demonstrated this.

25. We agree with the reviewer that this is a critical statistical calculation and is needed to support our conclusion. Thus, in the Extended Data Fig. 7a (copied below) of the revised manuscript, we have included comparisons between the overlap of corner cells that encode concave or convex corners and the overlap expected by chance. The left panel compares corner cells (both concave) in square and rectangle environments, the middle panel compares corner cells (concave vs. convex) in square and convex-1 environments, and the right panel compares corner cells (concave vs. convex) in rectangle and convex-1 environments (Extended Data Fig. 7a). The gray histogram illustrates the distribution of overlap expected by chance. This distribution was generated by randomly selecting the same number of neurons, as indicated above for each environment, 1000 times in each mouse ($n = 9$ mice). The black bar denotes the 95th percentile of each distribution, while the red bar indicates the actual measured overlap between corner cells (see also Fig. 4f copied below). Corner cells in the

Fig. 4f. (f) Venn diagram showing the overlap between classified corner cells encoding concave or convex corners. All numbers were normalized to the number of corner cells in square.

Extended Data Fig. 7a. (a) Related to Fig. 4f: Comparisons between the overlap of concave and convex corner cells and the overlap expected by chance. The left panel compares corner cells (both concave) in square and rectangle environments, the middle panel compares corner cells (concave vs. convex) in square and convex-1 environments, and the right panel compares corner cells (concave vs. convex) in rectangle and convex-1 environments. The gray histogram illustrates the distribution of overlap expected by chance. This distribution is generated by randomly selecting the same number of neurons, as indicated above for each environment, for 1000 times in each mouse ($n = 9$ mice). The black bar denotes the 95th percentile of each distribution, while the red bar indicates the actual measured overlap between corner cells. Corner cells in the square and rectangle showed an overlap that is higher than chance (first panel), while the

square and rectangle showed an overlap that is higher than chance (first panel), while the overlap between corner cells that encode concave or convex corners was minimal and below the chance level.

12) On a similar note, were any cells identified as conjunctive corner and place cells? Is it possible that a cell can have both properties, but criterion 2 of defining a corner cell excludes such cells?

26. The reviewer asks an interesting question. In fact, the majority of corner cells ($89.4 \pm 2.3\%$, $n = 9$ mice) carry significant spatial information, as they are active in a consistent and constrained spatial location in the environment, which would classify them as 'place cells' if one were to use only a spatial information criterion. However, spatial information is only sensitive to the shape of the distribution of spatially-binned firing rates. In other words, a high spatial information score would result from a cell that is active in 10% of bins and not active in 90% of bins - even if those 10% of bins were distributed across four corners or restricted to a single corner. Thus, corner cells should have high spatial information scores, as they have firing activity restricted to a small number of spatial bins (those near the corners). However, if one considers classifying place cells as cells that are active in one (or possibly two) restricted spatial locations, then almost no corner cells would fall under this category (see all corner cells recorded in a single mouse in new Extended Data Fig. 3).

13) It is not clear what is accomplished by the corrections of spike rates procedure described in lines 551-559, using the mean firing rate across all cells. What if a particular cell has a behavioral confound that is not reflected in the population mean activity? I don't understand how this analysis works and whether it does what the authors intend. How is this any different from doing an occupancy normalized rate map? Can the authors show a real or hypothetical example of a cell whose rate map is altered appreciably by this procedure?

27. The initial motivation for conducting this additional correction on top of the occupancy-normalized rate map arose from the inspection of spike rates in corner cells across different corners within an environment (triangle, square, and hexagon). Extended Data Fig. 4b (copied below) illustrates the spike rates of corner cells at each corner of the triangle, square, and hexagon, respectively, calculated using the occupancy-normalized rate maps of corner cells (Extended Data Fig. 4a-b). Using these rate maps, we observed a significant difference in the spike rates across different corners in the square (repeated measures ANOVA: $F(2.24, 12.92) = 5.76$, $p = 0.010$; $n = 9$ mice), but not in the triangle or hexagon.

Next, we examined the same quantification using the simulated rate map for each mouse (Extended Data Fig. 4c-d). As mentioned previously, the simulated rate map was generated using a simulated neuron that fires along the animal's trajectory, using the animal's own speed and the overall mean spike rate observed across all neurons of a given mouse. Unexpectedly, we observed that the difference in spike rates in the square persisted even in the simulated rate maps (repeated measures ANOVA: $F(2.56, 20.48) = 4.24$, $p = 0.022$; $n = 9$ mice). This suggests that this effect in the square was not directly related to the firing of corner cells but rather to the animals' behavior in the square environment.

Finally, we divided the occupancy-normalized rate map by the simulated rate map to obtain a corrected rate map (Extended Data Fig. 4e). Since purely behavior-related changes should be evident in both the original and simulated rate maps, this method corrected the observed spike rate difference of corner cells in the square environment (Extended Data Fig. 4f, repeated measures ANOVA: $p > 0.05$).

To ensure that this correction does not significantly alter our conclusions, we repeated our key results using the occupancy-normalized rate map instead of the corrected rate map (Fig. R3). We found that the two methods yielded almost identical results, and our conclusions remained consistent regardless of whether we used the occupancy-normalized rate map or the corrected rate map (Fig. R3, below).

Extended Data Fig. 4a-f. (a) Raster plot and the corresponding rate map of an example corner cell. The raster plot (top) indicates extracted spikes (red dots) on top of the animal's running trajectory (grey lines) and the spatial rate map (bottom) is color coded for maximum (red) and minimum (blue) values. (b) Spike rates of corner cells at each corner of the triangle, square, and hexagon, respectively, calculated using the rate maps of corner cells. Each line represents a mouse. There is a significant difference in the corner spike rates across different corners in the square (repeated measures ANOVA: $F(2.24, 12.92) = 5.76$, $p = 0.010$; $n = 9$ mice). ns: not significant. (c) Raster plot and the corresponding rate map of a simulated cell. The simulated rate map was generated using a simulated neuron that fires along the animal's trajectory using the animal's own speed at the overall mean spike rate observed across all neurons of a given mouse (Methods). (d) Same as (b), but calculated using simulated rate maps for each mouse. The difference in corner spike rates in the square persists even in the simulated rate maps (repeated measures ANOVA: $F(2.56, 20.48) = 4.24$, $p = 0.022$; $n = 9$ mice), indicating this effect is due to animals' behavior. (e) An example of corrected rate map. To obtain the corrected rate map, we divided the original rate map (i.e., a) by the simulated rate map (i.e., c). Therefore, the spike rates on the corrected rate map were automatically converted to fold changes relative to the simulated rate map. This method was used to correct for any measurements that might have been associated with the animal's movement or occupancy, as purely behavior-related changes should be evident in both the original and simulated rate maps. (f) Same as (b), but calculated using the corrected rate

Fig. R3. Data plotted using the occupancy normalized rate map instead of the corrected rate map in the manuscript. Upper left labels indicate the corresponding figure panels in the manuscript. Two-tailed Wilcoxon signed rank tests, * $p < 0.05$, ** $p < 0.01$, ns: not significant.

Minor:

14) Line 45: The subiculum receives input from both the medial and lateral entorhinal cortex, and thus the word “medial” should be deleted.

We have removed this word.

15) Words like “Surprisingly” and “Interestingly” at the start of sentences should be used sparingly, and preferably deleted entirely. Let the data speak for themselves and let the reader decide whether they are interesting or surprising.

We have removed these words entirely.

16) Line 302-307: Alternatively, the egocentric corner cells of LEC may combine to produce the allocentric corner cells of subiculum (similar to the model of Gofman et al. (2019) from the Derdikman lab of egocentric border cells in postsubiculum combining to form allocentric border cells in MEC and parasubiculum). In general, it is not clear that the subiculum is unique in having cells that can encode geometry (boundaries, corners, curvatures), as similar properties have been seen in CA1, LEC, and MEC as well. This does not negate the importance of this paper in terms of its thorough analysis of corner tuning in the system, but the evidence is not strong that these geometric coding

properties are specially associated with the subiculum in the hippocampal formation processing loops.

We appreciate the reviewer's careful consideration of the structure of our manuscript. In the revised version, we have re-framed a significant portion of our writing to better reflect our key findings and have also expanded the scope of the paper. Specifically, our work highlights that the subiculum explicitly encodes a fundamental geometric property that defines an environmental shape: concavity and convexity. Importantly, single-cell activity tracks the curvatures of this geometric feature. Consequently, this explicit encoding of concave and convex curvatures, in combination with the encoding of straight environmental boundaries, allows the brain to adapt to encoding and reconstructing any geometric shapes within an environment. This achievement is impossible to attain using only allocentric and egocentric boundary vector cells. Moreover, this encoding of curvature is fundamentally different from the coding of geometry as a whole in place cells and grid cells (see our response 2 to reviewer 1), and we have not come across similar reports of this curvature encoding in other brain regions. Thus, the subiculum appears to be a unique brain structure for encoding this geometric property and has the potential to effectively reconstruct the detailed geometric shape of an environment.

As detailed in responses 1-5, we have changed the title to: 'Subicular neurons encode concave and convex geometries', and made significant changes to the abstract, introduction and discussion of the manuscript in addition to new analyses (as detailed in response 11). We believe this new text and analyses have provided a richer conceptual framework in which to consider the unique significance of the information encoded by the subiculum neurons we describe.

17) Line 304 does not entirely accurately represent the corner-associated activity of LEC shown by Wang et al. (2023). 8/12 of the corner cells in that paper appeared to be egocentric in terms of CW vs CCW direction selectivity and about 7/12 were speed-tuned. Thus, not all corner cells in LEC were egocentric and speed-modulated.

We have edited the discussion regarding the description of Wang et al., 2023 to reflect the findings more precisely.

18) Although the authors recorded males and females and combined data (which is fine if there are no differences between groups), the authors should include a statement that they investigated whether there were sex differences and combined the data if there were no significant differences.

We have added "data from both males and females were combined for analysis, as we did not observe sex differences in this research" into the Methods.

19) Line 431: Were whiskers completely trimmed? Please provide some details for replication purposes.

We have added "facial whiskers were trimmed (not epilated) with scissors until no visible whiskers remained on the face" into the method.

20) Line 494: Please describe how a place field was defined and how its limits were determined.

Please see response 7 to reviewer 1 and Figure R1 for the justification of how a place field was defined on a given rate map. We have also added more detailed descriptions into the method section.

21) If possible, please provide some indication of where the recordings were from along the transverse axis and along the longitudinal axis of the subiculum. Also, can the authors estimate how deep into the layer the recordings were taken from? Were these all deep-layer cells?

We measured the effective image size (the area with detectable neurons) for each mouse and combined this information with histology. The anatomical region where neurons were recorded was approximately within a 450- μ m diameter circular area centered around AP: -3.40 mm and ML: +2 mm. Due to the limitations of 1-photon imaging, we could not accurately estimate the imaging depth, but we believe the recordings were primarily from the deep layer of the subiculum. This information has now been added to the Methods.

22) Line 1062: How is a 2D surface linearized?

The linearization of the 2D surface was a vertical concatenation of each column of spatial bins. However, this was just an intermediate step for the decoding process and we eventually mapped the position back onto the 2D space. We recognize the presentation of the decoding trace using the linearized position was confusing and have now changed it to the decoding trace of x and y positions in Extended Data Fig 2e (copied below).

Extended Data Fig. 2e. (e) An example of the true vs. decoded spatial x-y position using the full decoder (all recorded subiculum neurons).

23) ED Fig 5b is unclear. Are these real data, or a schematic? It is not clear how to read this figure.

We apologize that description of this figure panel was not clear. It is real behavioral data from a representative mouse. We have replotted this figure and edited the figure legend as follows: “Allocentric corner bearing showing real behavioral data in the square and convex-1 environments from one representative mouse. Each position is color coded for the allocentric bearing of the nearest corner relative to the animal at that location. Note the discrete shift in color coding represents a shift in which corner is the closest to the animal (e.g. the north west versus south west corner).”

References:

- Alexander, A.S., Carstensen, L.C., Hinman, J.R., Raudies, F., Chapman, G.W., and Hasselmo, M.E. (2020). Egocentric boundary vector tuning of the retrosplenial cortex. *Science Advances* 6.
- Barry, C., Hayman, R., Burgess, N., and Jeffery, K.J. (2007). Experience-dependent rescaling of entorhinal grids. *Nature Neuroscience* 10, 682-684.
- Barry, C., Lever, C., Hayman, R., Hartley, T., Burton, S., O'Keefe, J., Jeffery, K., and Burgess, N. (2006). The Boundary Vector Cell Model of Place Cell Firing and Spatial Memory. *Reviews in the Neurosciences* 17.
- Bostock, E., Muller, R.U., and Kubie, J.L. (2004). Experience - dependent modifications of hippocampal place cell firing. *Hippocampus* 1, 193-205.
- Colgin, L.L., Leutgeb, S., Jezek, K., Leutgeb, J.K., Moser, E.I., McNaughton, B.L., and Moser, M.-B. (2010). Attractor-Map Versus Autoassociation Based Attractor Dynamics in the Hippocampal Network. *Journal of Neurophysiology* 104, 35-50.
- Gardner, R.J., Hermansen, E., Pachitariu, M., Burak, Y., Baas, N.A., Dunn, B.A., Moser, M.-B., and Moser, E.I. (2022). Toroidal topology of population activity in grid cells. *Nature* 602, 123-128.
- Giocomo, Lisa M., Hussaini, Syed A., Zheng, F., Kandel, Eric R., Moser, M.-B., and Moser, Edvard I. (2011). Grid Cells Use HCN1 Channels for Spatial Scaling. *Cell* 147, 1159-1170.
- Knierim, J.J., Kudrimoti, H.S., and McNaughton, B.L. (1998). Interactions Between Idiothetic Cues and External Landmarks in the Control of Place Cells and Head Direction Cells. *Journal of Neurophysiology* 80, 425-446.
- Krupic, J., Bauza, M., Burton, S., Barry, C., and O'Keefe, J. (2015). Grid cell symmetry is shaped by environmental geometry. *Nature* 518, 232-235.
- LaChance, P.A., and Taube, J.S. (2023). Geometric determinants of the postrhinal egocentric spatial map. *Current Biology* 33, 1728-1743.e1727.
- LaChance, P.A., Todd, T.P., and Taube, J.S. (2019). A sense of space in postrhinal cortex. *Science* 365, eaax4192.
- Leutgeb, J.K., Leutgeb, S., Treves, A., Meyer, R., Barnes, C.A., McNaughton, B.L., Moser, M.B., and Moser, E.I. (2005). Progressive transformation of hippocampal neuronal representations in "morphed" environments. *Neuron* 48, 345-358.
- Leutgeb, S., Leutgeb, J.K., Treves, A., Moser, M.-B., and Moser, E.I. (2004). Distinct Ensemble Codes in Hippocampal Areas CA3 and CA1. *Science* 305, 1295-1298.
- Lever, C., Burton, S., Jeewajee, A., O'Keefe, J., and Burgess, N. (2009). Boundary Vector Cells in the Subiculum of the Hippocampal Formation. *Journal of Neuroscience* 29, 9771-9777.
- Lever, C., Wills, T., Cacucci, F., Burgess, N., and O'Keefe, J. (2002). Long-term plasticity in hippocampal place-cell representation of environmental geometry. *Nature* 416, 90-94.
- Mallory, C.S., Hardcastle, K., Bant, J.S., and Giocomo, L.M. (2018). Grid scale drives the scale and long-term stability of place maps. *Nature Neuroscience* 21, 270-282.
- Muir, G.M., and Bilkey, D.K. (2001). Instability in the Place Field Location of Hippocampal Place Cells after Lesions Centered on the Perirhinal Cortex. *The Journal of Neuroscience* 21, 4016-4025.
- Muller, R.U., and Kubie, J.L. (1987). The effects of changes in the environment on the spatial firing of hippocampal complex-spike cells. *The Journal of Neuroscience* 7, 1951-1968.
- Munn, R.G.K., Mallory, C.S., Hardcastle, K., Chetkovich, D.M., and Giocomo, L.M. (2020). Entorhinal velocity signals reflect environmental geometry. *Nature Neuroscience* 23, 239-251.
- Nagelhus, A., Andersson, S.O., Cogno, S.G., Moser, E.I., and Moser, M.-B. (2023). Object-centered population coding in CA1 of the hippocampus. *Neuron* 111, 2091-2104.e2014.
- O'Keefe, J., and Burgess, N. (1996). Geometric determinants of the place fields of hippocampal neurons. *Nature* 381, 425-428.
- O'Keefe, J., and Conway, D.H. (1978). Hippocampal place units in the freely moving rat: Why they fire where they fire. *Experimental Brain Research* 31.
- Solstad, T., Boccara, C.N., Kropff, E., Moser, M.-B., and Moser, E.I. (2008). Representation of Geometric Borders in the Entorhinal Cortex. *Science* 322, 1865-1868.

- Stensola, H., Stensola, T., Solstad, T., Frøland, K., Moser, M.-B., and Moser, E.I. (2012). The entorhinal grid map is discretized. *Nature* 492, 72-78.
- Sun, Y., and Giocomo, L.M. (2022). Neural circuit dynamics of drug-context associative learning in the mouse hippocampus. *Nature Communications* 13.
- Sun, Y., Jin, S., Lin, X., Chen, L., Qiao, X., Jiang, L., Zhou, P., Johnston, K.G., Golshani, P., Nie, Q., *et al.* (2019). CA1-projecting subiculum neurons facilitate object-place learning. *Nat Neurosci* 22, 1857-1870.
- Wills, T.J., Lever, C., Cacucci, F., Burgess, N., and O'Keefe, J. (2005). Attractor dynamics in the hippocampal representation of the local environment. *Science* 308, 873-876.
- Witter, M. (2006). Connections of the subiculum of the rat: Topography in relation to columnar and laminar organization. *Behavioural Brain Research* 174, 251-264.
- Witter, M., and Amaral, D. (2004). The hippocampal region. *The rat nervous system*, 637-703.

Reviewer Reports on the First Revision:

Referees' comments:

Referee #1 (Remarks to the Author):

The dedication of the authors to refine and enhance their study is commendable, and it is evident that a significant effort has been made to address my comments comprehensively. The revisions made throughout the manuscript, in the title, abstract, introduction, and discussion, reflect an effort to better highlight the research findings. The updated classification criteria for 'corner cells' and the incorporation of within-session stability strengthens the core arguments of the paper. The additional control analyses, notably those presented in the new supplementary material, provide important validation for the existence and distinctiveness of corner cells in the subiculum. Given the concern and focus of the 2nd reviewer about these statistics, I would make sure that they are satisfied with the authors' response. The supplemental data from the hippocampal CA1 region, which shows only a minimal presence of corner cells, helps to draw a contrast between the encoding capabilities of different brain regions, reinforcing the unique role of the subiculum. Though here I must note that there were no corner manipulations, so this additional dataset is not as thorough as one would hope. The introduction of manifold embedding is okay (though not very surprising) and new decoding analyses are good additions to address my previous comments. Overall, these techniques used are sound and contribute to the nuanced understanding of the neural representation of space in the subiculum.

Despite these strengths, I must again express that my reservations about the overarching impact and novelty of the findings remain. My overall enthusiasm for the finding that a small percentage (5%) of subicular neurons activate at corners is not very high. The discovery of 'corner cells', although an interesting addition to the lexicon of spatially responsive neurons, appears to be a subtle addition to the established framework of spatial encoding by hippocampal areas. The manuscript's detailed discussion on the explicit encoding of concave and convex curvatures, in addition to straight environmental boundaries, does propose a novel function of spatial encoding that is possibly unique to the subiculum. However, to elevate this finding to warrant publication at a major journal, there would need to be a clearer elucidation of how this encoding significantly alters or enhances our conceptual models of spatial navigation and cognition. The authors' efforts to situate 'corner cells' within the existing framework of spatial encoding are evident and appreciated. The updated statistical methodology, adoption of nonparametric testing, and individualized treatment of data points for statistical analysis collectively ensure that the conclusions are robust and statistically sound.

While the manuscript has certainly improved and the authors' commitment is to be acknowledged, I must convey, with all due respect and consideration for the authors' substantial efforts, that the leap to a groundbreaking discovery does not fully materialize in this study. The authors have revealed an interesting (and rare) cellular response to add to the many spatial cellular responses reported throughout the hippocampus and its related cortical areas. To me, this is incremental in nature, and the absence of a demonstrated significant paradigm shift or a fundamental challenge or support to existing theories compels me to express doubts about the manuscript's suitability at a top

journal.

Referee #2 (Remarks to the Author):

The authors have comprehensively and satisfactorily addressed all of my comments from the initial submission. I have no further concerns.

Referees' comments:

Referee #1 (Remarks to the Author):

The dedication of the authors to refine and enhance their study is commendable, and it is evident that a significant effort has been made to address my comments comprehensively. The revisions made throughout the manuscript, in the title, abstract, introduction, and discussion, reflect an effort to better highlight the research findings. The updated classification criteria for 'corner cells' and the incorporation of within-session stability strengthens the core arguments of the paper. The additional control analyses, notably those presented in the new supplementary material, provide important validation for the existence and distinctiveness of corner cells in the subiculum. Given the concern and focus of the 2nd reviewer about these statistics, I would make sure that they are satisfied with the authors' response. The supplemental data from the hippocampal CA1 region, which shows only a minimal presence of corner cells, helps to draw a contrast between the encoding capabilities of different brain regions, reinforcing the unique role of the subiculum. Though here I must note that there were no corner manipulations, so this additional dataset is not as thorough as one would hope. The introduction of manifold embedding is okay (though not very surprising) and new decoding analyses are good additions to address my previous comments. Overall, these techniques used are sound and contribute to the nuanced understanding of the neural representation of space in the subiculum.

We thank the reviewer for their constructive feedback and appreciate the reviewer's careful consideration of the work, which we believe led to an overall improved manuscript.

Despite these strengths, I must again express that my reservations about the overarching impact and novelty of the findings remain. My overall enthusiasm for the finding that a small percentage (5%) of subicular neurons activate at corners is not very high. The discovery of 'corner cells', although an interesting addition to the lexicon of spatially responsive neurons, appears to be a subtle addition to the established framework of spatial encoding by hippocampal areas. The manuscript's detailed discussion on the explicit encoding of concave and convex curvatures, in addition to straight environmental boundaries, does propose a novel function of spatial encoding that is possibly unique to the subiculum. However, to elevate this finding to warrant publication at a major journal, there would need to be a clearer elucidation of how this encoding significantly alters or enhances our conceptual models of spatial navigation and cognition. The authors' efforts to situate 'corner cells' within the existing framework of spatial encoding are evident and appreciated. The updated statistical methodology, adoption of nonparametric testing, and individualized treatment of data points for statistical analysis collectively ensure that the conclusions are robust and statistically sound. While the manuscript has certainly improved and the authors' commitment is to be acknowledged, I must convey, with all due respect and consideration for the authors' substantial efforts, that the leap to a groundbreaking discovery does not fully materialize in this study. The authors have revealed an interesting (and rare) cellular response to add to the many spatial cellular responses reported throughout the hippocampus and its related cortical areas. To me, this is incremental in nature, and the absence of a demonstrated significant paradigm shift or a fundamental challenge or support to existing theories compels me to express doubts about the manuscript's suitability at a top journal.

We appreciate the reviewer's comments. First, we would note that in the environment with the smallest angled corners (the triangle) the percentage is 7%, closely paralleling the percentages observed for other functionally defined cell types like border or grid cells (in mice) in the medial entorhinal cortex (Solstad et al., 2008; Mallory et al., 2018). These cell classes have been instrumental in advancing our understanding of how the brain builds an internal map of the external world. In addition, we would note that grid, border, head direction and other spatial cell 'classes' are functionally-defined and represent the clearest tuning curves of a more continuous coding scheme for an animal's position and orientation in the world (Hardcastle et al., Neuron, 2017). We expect the same is true for corner coding, an interpretation consistent with our observation that corner activity was highly consistent across different geometries when considering the neurons as a population rather than only single cells classified based on their corner score and that, at the population level, each corner for a given environment

was represented (as visualized by the low-dimensional neural manifold of all neural activity in the subiculum). However, to provide a rigorous investigation of the coding properties of this completely novel cell ‘class’, we chose to use a relatively strict threshold for functionally-defining corner cells.

Second, the discovery of neurons attuned to specific geometric features of the environment, notably convexity and concavity, underscores the brain’s capacity to integrate sensory information across modalities to construct an internal model of an external variable (in this case geometry), which is then presumably used to guide behavior. This finding is particularly significant considering that there are no dedicated sensory receptors for detecting convexity or concavity. Moreover, these geometric features have important relevance to the natural world, in which burrow locations or entrances are often defined by having high concavity or convexity. Hence, this work paves the way for new explorations into the brain’s algorithms for computing high-order representations that are associated with features that have strong behavioral relevance to an animal.

Finally, prior work shows compelling evidence that animal’s use geometric features of an environment to navigate. However, whether such geometric features were represented explicitly by the coding features of individual neurons or only constructed and observable at the population (or manifold) level, was previously completely unknown. Here, we provide a robust demonstration that single cells do indeed encode the convexity and concavity of environments. We believe this will open new avenues for models of how the brain supports navigation, with the finding of corner coding providing a potential mechanism to more intricately connect such a model to the complexity of navigation in the natural world.

In an effort to keep the manuscript concise, we do not elaborate on all of these points. However, to further emphasize the novelty of the work, we have added this text to the discussion:

“Furthermore, this corner coding generalized to a broader framework for coding environmental concavity and convexity. Such coding may have particular relevance to animals navigating in natural environments, in which features such as burrows or nesting sites are often high in concavity or convexity.”

“Understanding how corner specific patterns are generated could provide important insight into the algorithms the brain uses to construct a single cell code for geometric features and future work using targeted manipulations in the hippocampus may help resolve this question⁴⁷.”

Referee #2 (Remarks to the Author):

The authors have comprehensively and satisfactorily addressed all of my comments from the initial submission. I have no further concerns.

We thank the reviewer for their prior comments, which were highly constructive and we believe led to an improved manuscript.